### ARTICLES
## OPEN

# *Bifidobacterium* species associated with breastfeeding produce aromatic lactic acids in the infant gut

Martin F. Laursen [1,8], Mikiyasu Sakanaka[1,2], Nicole von Burg[3], Urs Mörbe [3], Daniel Andersen[4], Janne Marie Moll [4], Ceyda T. Pekmez[5], Aymeric Rivollier[3], Kim F. Michaelsen [5], Christian Mølgaard[5], Mads Vendelbo Lind[5], Lars O. Dragsted[5], Takane Katayama [2,6], Henrik L. Frandsen[1], Anne Marie Vinggaard[1], Martin I. Bahl [1], Susanne Brix [4], William Agace[3,7], Tine R. Licht[1 ✉] and Henrik M. Roager [1,5,8 ✉]

Breastfeeding profoundly shapes the infant gut microbiota, which is critical for early life immune development, and the gut microbiota can impact host physiology in various ways, such as through the production of metabolites. However, few breastmilk-dependent microbial metabolites mediating host–microbiota interactions are currently known. Here, we demonstrate that breastmilk-promoted *Bifidobacterium* species convert aromatic amino acids (tryptophan, phenylalanine and tyrosine) into their respective aromatic lactic acids (indolelactic acid, phenyllactic acid and 4-hydroxyphenyllactic acid) via a previously unrecognized aromatic lactate dehydrogenase (ALDH). The ability of *Bifidobacterium* species to convert aromatic amino acids to their lactic acid derivatives was confirmed using monocolonized mice. Longitudinal profiling of the faecal microbiota composition and metabolome of Danish infants (n = 25), from birth until 6 months of age, showed that faecal concentrations of aromatic lactic acids are correlated positively with the abundance of human milk oligosaccharide-degrading *Bifidobacterium* species containing the ALDH, including *Bifidobacterium longum*, *B. breve* and *B. bifidum*. We further demonstrate that faecal concentrations of *Bifidobacterium*-derived indolelactic acid are associated with the capacity of these samples to activate in vitro the aryl hydrocarbon receptor (AhR), a receptor important for controlling intestinal homoeostasis and immune responses. Finally, we show that indolelactic acid modulates ex vivo immune responses of human CD4[+] T cells and monocytes in a dose-dependent manner by acting as an agonist of both the AhR and hydroxycarboxylic acid receptor 3 (HCA₃). Our findings reveal that breastmilk-promoted *Bifidobacterium* species produce aromatic lactic acids in the gut of infants and suggest that these microbial metabolites may impact immune function in early life.

Human breastmilk is a well-adapted nutritional supply for the infant[1]. Breastfeeding provides the infant with important short-term protection against infections and may also provide long-term metabolic and immunological benefits[1–3]. These benefits may partly be mediated through the gut microbiota, since breastfeeding is the strongest determinant of gut microbiota composition and function during infancy[4–6]. Human breastmilk contains human milk oligosaccharides (HMOs), which are complex, highly abundant sugars serving as substrates for specific microbes including certain species of *Bifidobacterium*[7]. This co-evolution between bifidobacteria and the host, mediated by HMOs, to a large extent directs the colonization of the gut in early life, which has critical impact on the immune system[8]. Depletion of specific microbes, including *Bifidobacterium*, in early life has been associated with increased risk of allergy and asthma development in childhood[9,10] and has been suggested to compromise immune function and lead to increased susceptibility to infectious disease[11,12]. Despite *Bifidobacterium* dominating the gut of breastfed infants and

being widely acknowledged as beneficial, mechanistic insights into the contribution of these bacteria and their metabolites to immune function and development are limited and have mainly focused on short-chain fatty acids[12,13]. Recent studies show that microbial aromatic amino acid metabolites including tryptophan-derived indoles[14] via activation of the aryl hydrocarbon receptor (AhR)[15,16] can fortify the intestinal barrier[17,18], protect against pathogenic infections[15,19] and influence host metabolism[17,20,21], making this group of microbial metabolites of particular interest in the context of early life.

Here, we show that breastmilk-promoted *Bifidobacterium* species, via a previously unrecognized aromatic lactate dehydrogenase (ALDH), produce aromatic lactic acids including indolelactic acid (ILA) in substantial amounts in the infant gut and that faecal concentrations of this metabolite are correlated with the capacity of infant faeces to activate AhR. We furthermore demonstrate that ILA via AhR and hydroxycarboxylic acid receptor 3 (HCA₃)-dependent pathways impact immune functions ex vivo, suggesting that

[1]National Food Institute, Technical University of Denmark, Kgs. Lyngby, Denmark. [2]Faculty of Bioresources and Environmental Sciences, Ishikawa Prefectural University, Ishikawa, Japan. [3]Mucosal Immunology Group, Department of Health Technology, Technical University of Denmark, Kgs. Lyngby, Denmark. [4]Department of Biotechnology and Biomedicine, Technical University of Denmark, Kgs. Lyngby, Denmark. [5]Department of Nutrition, Exercise and Sports, University of Copenhagen, Frederiksberg C, Denmark. [6]Graduate School of Biostudies, Kyoto University, Kyoto, Japan. [7]Immunology Section, BMC D14, Department of Experimental Medicine, Lund University, Lund, Sweden. [8]These authors contributed equally: Martin F. Laursen, Henrik M. Roager. ✉e-mail: trli@food.dtu.dk; hero@nexs.ku.dk

1367

breastmilk-promoted *Bifidobacterium* via production of aromatic lactic acids impact the immune system in early life.

## Results

**_Bifidobacterium_ associate with aromatic lactic acids in the infant gut.** To explore interactions between breastfeeding status, gut microbial composition and metabolism of aromatic amino acids in early life, we used 16S ribosomal RNA amplicon sequencing to infer gut microbiota composition and a targeted ultraperformance liquid chromatography–mass spectrometry (UPLC–MS) metabolomics approach to quantify 19 aromatic amino acids and derivatives thereof (Supplementary Tables 1 and 2) in faecal samples from 59 healthy Danish infants from the SKOT I cohort[22]. The SKOT I infants included were born full term, $9.1 \pm 0.3$ (mean $\pm$ s.d.) months of age at sampling and 40.7% were still partially breastfed (Supplementary Data 1a,b). After stratification of the 9-month-old infants on the basis of breastfeeding status (partially breastfed versus weaned), principal coordinates analysis (PCoA) of weighted UniFrac distances showed a significant separation across the first PC-axis ($r^2 = 0.093$, $P < 0.001$, Adonis test; Fig. 1a), which mirrored an increasing gradient in relative abundance of *Bifidobacterium* ($r^2 = 0.397$, $P < 0.001$, Adonis test; Fig. 1b). Other metadata (age, gender, mode of delivery, current formula intake and age of introduction to solid foods) did not explain gut microbiota variation to the same degree as breastfeeding status ($r^2 < 0.05$, $P > 0.03$, Adonis tests; Supplementary Data 1c) and no bacterial genera differed significantly according to these parameters (FDR-corrected $P > 0.1$, Mann–Whitney $U$-tests; Supplementary Data 1d). Principal component analysis (PCA) of faecal aromatic amino acid metabolite concentrations (Supplementary Data 1e) also suggested a minor separation by breastfeeding status, which was largely driven by three aromatic lactic acids, 4-hydroxyphenyllactic acid (4-OH-PLA), phenyllactic acid (PLA) and indolelactic acid (ILA) (Fig. 1c). Correlation analysis revealed that *Bifidobacterium*, but no other bacterial genera, were significantly associated with faecal concentrations of all three aromatic lactic acids (4-OH-PLA, PLA and ILA), in addition to indolealdehyde (IAld) (Fig. 1d and Supplementary Data 1f). The *Bifidobacterium* species (Extended Data Fig. 1a and Supplementary Data 1g) enriched in the breastfed infants, *B. longum, B. bifidum* and *B. breve*, were positively associated with the faecal concentrations of aromatic lactic acids (4-OH-PLA, PLA and ILA) and IAld (cluster 1 in Fig. 1e) but negatively associated with the faecal concentrations of aromatic propionic acids, aromatic amino acids and, to a lesser degree, with aromatic acetic acids (cluster 2 in Fig. 1e). In contrast, postweaning type *Bifidobacterium* species, including *B. adolescentis, B. animalis/pseudolongum* and *B. catenulatum* group[23,24], were not significantly associated with aromatic lactic acids nor breastfeeding status (Fig. 1e). These associations were in agreement with the observation that the concentrations of the three aromatic lactic acids were higher in the faeces of breastfed than weaned infants (Extended Data Fig. 1b). Furthermore, the abundances of the three aromatic lactic acids in infant urine (Supplementary Figs. 1–3) showed similar positive associations with relative abundances of breastmilk-promoted *Bifidobacterium* species (Extended Data Fig. 1c). In addition, faecal and urinary levels of ILA were positively correlated ($\rho = 0.68$, $P < 0.0001$), showing that faecal levels of this metabolite are reflected systemically. Consistently, urine abundance of ILA, but not of PLA and 4-OH-PLA, were significantly higher in breastfed compared to weaned infants (Extended Data Fig. 1b). Together, this suggests that specific *Bifidobacterium* species produce aromatic lactic acids in the infant gut (Fig. 1f).

**_Bifidobacterium_ species produce aromatic lactic acids in vitro.** To confirm the ability of *Bifidobacterium* species detected in infants to produce aromatic lactic acids, *Bifidobacterium* type strains were grown anaerobically in a medium containing all three aromatic

amino acids with either glucose or HMOs as sole carbohydrate sources. Analyses of culture supernatants revealed that ILA, PLA and 4-OH-PLA were produced mainly by *B. bifidum, B. breve, B. longum* ssp. *longum, B. longum* ssp. *infantis* and *B. scardovii* (Fig. 2a), in accordance with the associations observed in the 9-month-old infants (Fig. 1e). Other *Bifidobacterium* species, namely *B. adolescentis, B. animalis* ssp. *lactis, B. animalis* ssp. *animalis, B. dentium, B. catenulatum, B. pseudocatenulatum* and *B. pseudolongum* ssp. *pseudolongum* produced only low amounts of these metabolites (Fig. 2a). The ability of *Bifidobacterium* species to produce high levels of the aromatic lactic acids was generally convergent with the ability to use HMOs as a carbohydrate source (Fig. 2a), suggesting a link between breastmilk-promoted bifidobacteria and production of aromatic lactic acids. None of the downstream products of the aromatic lactic acids (Fig. 1f) was detected in any of the culture supernatants.

**Identification of a responsible ALDH.** Since it has been reported that a lactate dehydrogenase (LDH) in *Lactobacillus* species can convert phenylpyruvic acid to PLA[25], we hypothesized that a corresponding enzyme would be present in *Bifidobacterium* species. Alignment and phylogenetic analysis of all genes annotated as *ldh* in the *Bifidobacterium* type strains included in this study, revealed four clusters (Fig. 2b). Whereas all *Bifidobacterium* genomes contain the *ldh* (here designated as type 1 *ldh*) responsible for conversion of pyruvic acid to lactic acid in the bifidobacterial fructose 6-phosphate shunt[26,27], some species have an extra *ldh*, here designated as type 2, type 3 and type 4, respectively. In agreement with the in vitro fermentations (Fig. 2a), all prominent aromatic lactic acid-producing *Bifidobacterium* species contain the type 4 *ldh*, suggesting that this could encode a previously unrecognized ALDH. A further analysis of all available whole-genome sequenced *Bifidobacterium* strains showed that the type 4 *ldh* is universally present in *B. longum, B. bifidum, B. breve* and *B. scardovii* strains (Supplementary Table 3). Interestingly, genomic analysis of the *Bifidobacterium* type strains revealed that the type 4 *ldh* gene is part of a genetic element containing an amino acid transaminase gene (suspected to be responsible for converting the aromatic amino acids into aromatic pyruvic acids) and a haloacid dehalogenase gene (of unknown importance) (Supplementary Fig. 4), which has been indicated to constitute an operon in *B. breve*[28]. Cloning of the type 4 *ldh* gene from the type strain of *B. longum* ssp. *infantis* (DSM20088) into a vector transformed into *Escherichia coli* revealed that the expression of the type 4 *ldh* gene indeed resulted in the appearance of PLA, 4-OH-PLA and ILA in the culture supernatant (Fig. 2c). To verify the type 4 *ldh*-dependent production of aromatic lactic acids in *Bifidobacterium* species, we generated a type 4 *ldh* insertional mutant strain by homologous recombination in *B. longum* ssp. *longum* 105-A (Supplementary Fig. 5), a genetically tractable strain[29,30] containing the type 4 *ldh* (Supplementary Fig. 6a). The type 4 LDH amino acid sequence of the 105-A strain had >98% identity to the homologues in type strains of *B. longum* ssp. *longum* and *B. longum* ssp. *infantis* and >91% identity to *B. bifidum, B. breve* and *B. scardovii* (Supplementary Fig. 6b) but no non-bifidobacterial homologues were found by BLAST analysis (amino acid sequence identity cutoff 60%). Cultivation of the wild type (WT), the type 4 *ldh* mutant strain and a complemented type 4 *ldh* mutant strain in a medium containing the three aromatic amino acids confirmed that type 4 *ldh* disruption did not impair growth in a rich medium (Fig. 2d). ILA, PLA and 4-OH-PLA accumulated in the supernatant of the WT and of the complemented type 4 *ldh* mutant strains but not in the type 4 *ldh* mutant (Fig. 2e). Importantly, the type 4 *ldh* mutant was not significantly compromised in its ability to convert pyruvic acid to lactic acid (Fig. 2e), supporting the distinct role of type 4 *ldh* in converting aromatic pyruvic acids. Further, to demonstrate in vivo production of the indicated aromatic lactic acids, we monocolonized germ-free mice with either the WT or the type

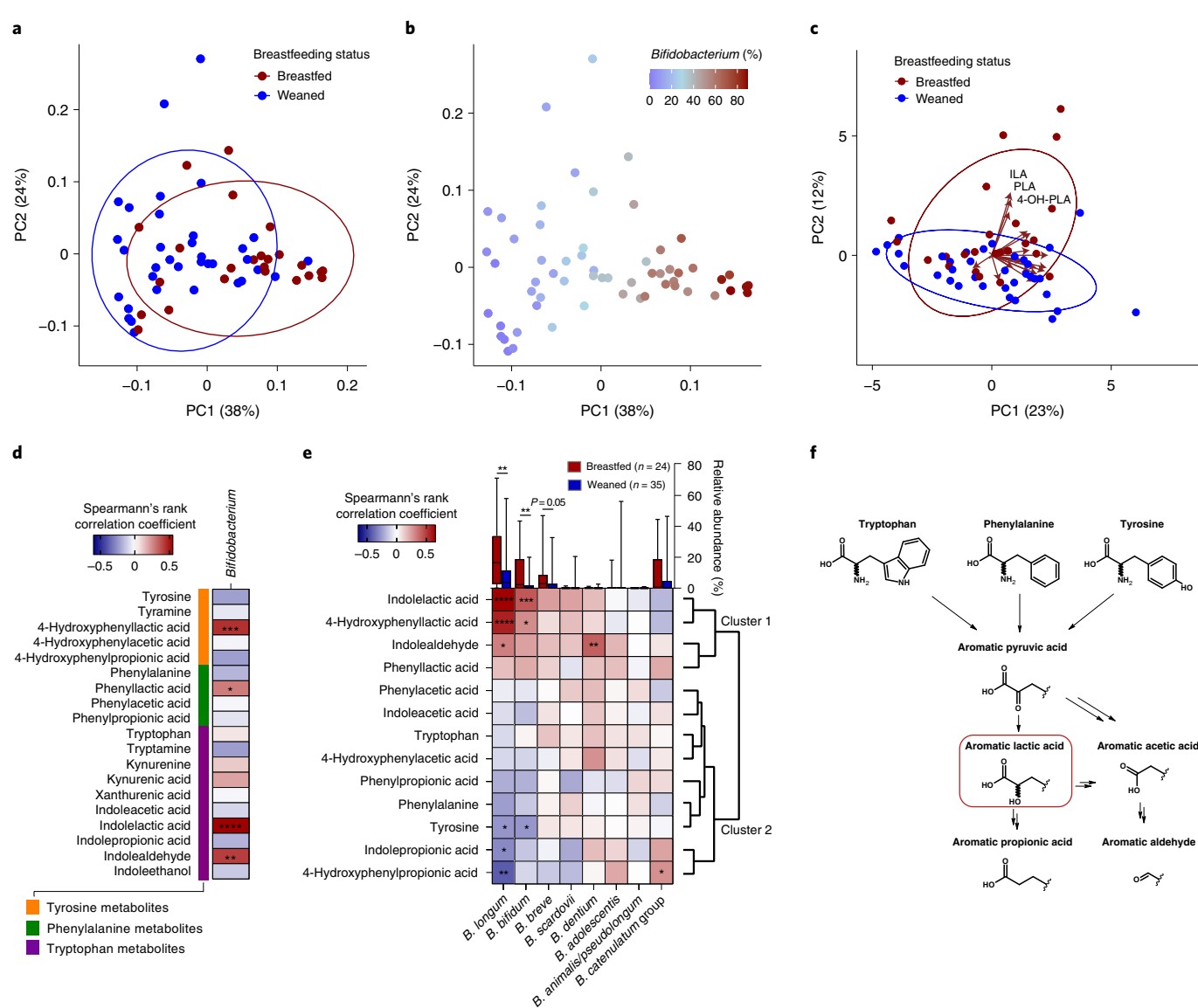

**Fig. 1 | Breastfeeding associates with faecal microbiota composition and aromatic amino acid metabolites in 9-month-old infants. a,b**, PCoA plots of weighted UniFrac distances based on OTUs from faecal samples of 9-month-old infants participating in the SKOT cohort (*n* = 59). **a**, Samples are coloured according to breastfeeding status, with ellipses indicating 80% confidence interval (CI) of datapoints for partially breastfed (red, *n* = 24) and weaned (blue, *n* = 35) infants. **b**, Samples are coloured according to relative abundance of the genus *Bifidobacterium*. **c**, PCA plot of concentrations (nmol g⁻¹ faeces) of aromatic amino acids and their derivatives in SKOT faecal samples, coloured according to breastfeeding status, with ellipses indicating 80% CI of datapoints for breastfed (red, *n* = 24) and weaned (blue, *n* = 35) infants. Loadings (correlations between variables and the principal components) are shown with arrows, with annotations of the aromatic amino acids ILA, 4-OH-PLA and PLA shown. **d**, Heatmap illustrating Spearman's rank correlation coefficients (two-sided tests) between the relative abundance of *Bifidobacterium* and concentrations of aromatic amino acids and their derivatives in SKOT faecal samples (*n* = 59). **e**, Heatmap illustrating hierarchical clustering (dendrogram on the right side) of Spearman's rank correlation coefficients (two-sided tests) between the relative abundance of the different *Bifidobacterium* species and selected microbial-derived aromatic amino acid catabolites in SKOT faecal samples (*n* = 59). Box and whiskers plots are showing relative abundance (line, median; box, interquartile range (IQR); whiskers, minimum to maximum) of the *Bifidobacterium* species, stratified according to breastfeeding status, with statistical significance evaluated by two-sided Mann–Whitney *U*-test. **f**, The pathway of aromatic amino acid catabolism by gut microbes (modified from Smith and Macfarlane[110], Smith and Macfarlane[111] and Zelante et al.[15]). For all panels, asterisks indicate statistical significance: *$P < 0.05$, **$P < 0.01$, ***$P < 0.001$ and ****$P < 0.0001$.

4 *ldh* mutant strain and found a 20–60-fold increase in their concentrations in WT versus type 4 *ldh* mutant monocolonized mice (Extended Data Fig. 2). Purification and characterization of the recombinant type 4 LDH enzyme revealed that it had a mass of 33.9 kDa (Supplementary Fig. 7a), while the native molecular mass was estimated to be 71.9 kDa by size exclusion chromatography, indicating dimer formation in solution (Supplementary Fig. 7b). Lack of added metal ions or addition of ethylenediaminetetraacetic

acid (EDTA) did not reduce enzymatic activity, the optimal pH was 8.0–8.5 and the enzyme was most stable at 37 °C (Supplementary Fig. 7c–e). Heterotrophic effects were neither observed for fructose 1,6-bisphosphate (an allosteric effector for type 1 LDH) nor for several intermediates for aromatic amino acid synthesis[26,27] (Supplementary Fig. 8). However, we found that phosphate served as a positive effector, suggesting that type 4 LDH is an intracellular enzyme (Supplementary Fig. 9a,b). Assay performed at the different

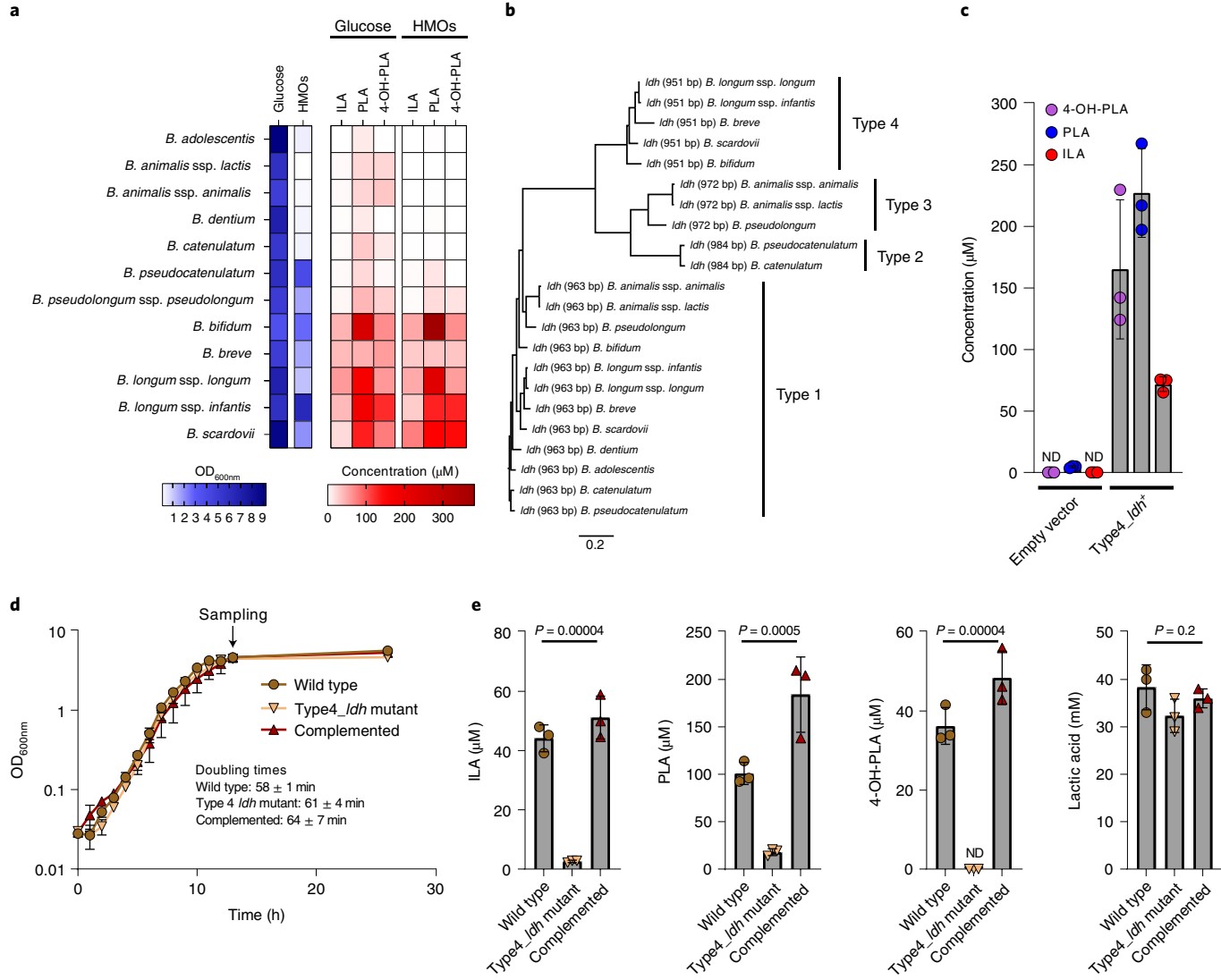

**Fig. 2 | *Bifidobacterium* species produce aromatic lactic acids via an ALDH. a**, In vitro production of ILA, PLA and 4-OH-PLA by *Bifidobacterium* species type strains in modified MRS medium (MRSc) with 2% (w/v) glucose or a mix of HMOs as sole carbohydrate source. For the type strains of *B. adolescentis*, *B. animalis* ssp. *animalis*, *B. animalis* ssp. *lactis*, *B. dentium* and *B. catenulatum*, no or very poor growth (OD$_{600nm}$ < 0.4) was observed with HMOs as the carbohydrate source. Means of three biological replicates are shown. **b**, Neighbour-joining phylogenetic tree of all genes in the *Bifidobacterium* species type strains annotated as LDH-encoding genes (*ldh*). The four clusters are designated types 1–4. **c**, Production of ILA, PLA and 4-OH-PLA by *E. coli* LMG194 cells transformed with an inducible vector lacking (empty vector) or containing the type 4 *ldh* (Type4_*ldh*+) from *B. longum* ssp. *infantis* DSM 20088 in LB medium 5 h post-induction of gene expression by addition of L-arabinose and supplementation with the aromatic pyruvic acids (indolepyruvic acid, phenylpyruvic acid and 4-hydroxyphenylpyruvic acid). Bars show mean ± s.d. of three biological replicates. **d**, Growth curves of *B. longum* ssp. *longum* 105-A (WT), its isogenic insertional type 4 *ldh* mutant (Type4_*ldh* mutant) and the type 4 *ldh* mutant strain complemented with the type 4 *ldh* gene (complemented). Curves show mean ± s.d. of three biological replicates and doubling times reported as mean ± s.d.. **e**, Production of ILA, PLA, 4-OH-PLA and lactic acid by wild type, Type4_*ldh* mutant and the complemented strain in early stationary phase cultures as indicated in **d**. Bars show mean ± s.d. of three biological replicates. Statistical significance was evaluated by one-way ANOVA. ND, not detected.

phosphate concentrations revealed the type 4 LDH is a *K*-type allosteric enzyme (Supplementary Fig. 9b). The catalytic rate ($k_{cat}$) was moderate to high for the aromatic pyruvic acid substrates but very low for pyruvic acid (Fig. 3), in accordance with the non-impaired lactic acid production observed for the type 4 *ldh* mutant (Fig. 2e). Production of ILA, PLA and 4-OH-PLA from the respective aromatic pyruvic acid substrates was verified by high-performance liquid chromatography (HPLC) (Supplementary Fig. 9c). The enzyme showed highest affinity (lowest $K_{0.5}$) for indolepyruvic acid but highest catalytic rate for 4-hydroxyphenylpyruvic acid in the presence of 100 mM phosphate (Fig. 3). However, the catalytic efficiency ($k_{cat}/K_{0.5}$) was highest for indolepyruvic acid (194 s$^{-1}$ mM$^{-1}$),

followed by 4-hydroxyphenylpyruvic acid (16 s$^{-1}$ mM$^{-1}$) and phenylpyruvic acid (11 s$^{-1}$ mM$^{-1}$), suggesting preference for indolepyruvic acid. The observed Hill coefficient ($n_H$ = 1.0–1.4) for all substrates indicate weak positive cooperativity under the conditions tested. Collectively, these results show that the type 4 *ldh* gene (from now on denoted *aldh*) encodes an ALDH responsible for the production of ILA, PLA and 4-OH-PLA in *Bifidobacterium* species associated with breastfeeding.

**Bifidobacterium species govern aromatic lactic acid profiles during early infancy.** To study the dynamics of *Bifidobacterium* species establishment and aromatic lactic acids in early infancy, we

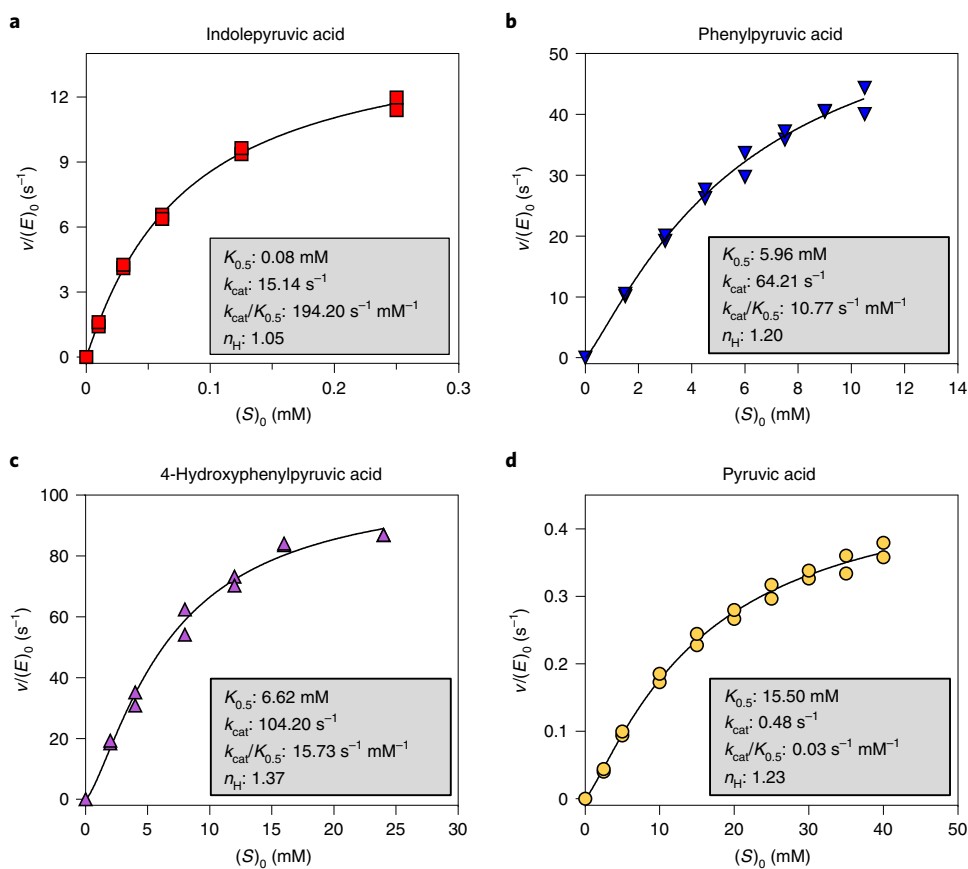

**Fig. 3 | Kinetic characterization of the bifidobacterial ALDH (type 4 LDH). a–d**, Substrate saturation curves of the type 4 LDH obtained for indolepyruvic acid (**a**), phenylpyruvic acid (**b**), 4-hydroxyphenylpyruvic acid (**c**) and pyruvic acid (**d**). Kinetic parameters, which were calculated by curve-fitting two independent experimental data to the Hill equation, are shown in the insets. The reaction was carried out in the presence of 100 mM phosphate.

established the Copenhagen Infant Gut (CIG) cohort including 25 healthy breastfed or mixed-fed infants, sampled every 2–4 weeks from birth until the age of 6 months (Supplementary Data 2a,b) for microbiome profiling (Supplementary Data 2c and Extended Data Fig. 3) and targeted metabolite quantification including aromatic lactic acids (Supplementary Table 4 and Supplementary Data 2d). The subject-specific gut microbiota profiles revealed a highly individualized species composition (Extended Data Fig. 3c and Extended Data Fig. 4a). As expected from a cohort mainly containing vaginally born, breastfed infants, the gut microbiota was highly dominated by *Bifidobacterium* (average of 64.2%) and among the top ten dominating taxa, *B. longum* (38.5%), *B. breve* (9.1%), *B. bifidum* (7.9%), *B. catenulatum* group (6.4%) and *B. dentium* (1.7%) were found (Extended Data Fig. 3a,b), with the remaining *Bifidobacterium* species being assigned to *B. scardovii* (0.24%), *B. adolescentis* (0.15%) and *B. animalis/pseudolongum* (0.10%) (Supplementary Data 2e). PCoA using Bray–Curtis dissimilarities, revealed a separation of the communities across samples based on relative abundance of the five dominating *Bifidobacterium* species, *B. longum*, *B. bifidum*, *B. breve*, *B. catenulatum* group and *B. dentium* (Fig. 4a and Extended Data Fig. 4b–f). Community abundance of *B. longum*, *B. bifidum* and *B. breve* but not *B. catenulatum* group and *B. dentium* (Fig. 4a) matched the measured faecal concentrations of the aromatic lactic acids (Fig. 4b). On the basis of quantitative PCR (qPCR) estimated total bacterial load of all samples, we calculated absolute abundances of each bacterial taxon in the 16S rRNA amplicon dataset and defined infant-type *Bifidobacterium* species as the summarized abundance of *B. longum*, *B. bifidum*, *B. breve* and *B. scardovii*. We observed a significant increase in the

absolute abundance of infant-type *Bifidobacterium* species from birth to around 6 months of age and this occurred concurrently with a progressive increase in the faecal concentrations of ILA, PLA and 4-OH-PLA and a progressive decrease in faecal abundances of HMO residuals (Extended Data Fig. 5). We confirmed by linear mixed models[31] adjusting for subject and age that the absolute abundances of the infant-type *Bifidobacterium* species were positively associated with faecal levels of ILA, PLA and 4-OH-PLA and additionally negatively associated with abundances of HMOs residuals in faeces (Fig. 4c). Among all bacterial taxa detected, *B. longum*, *B. bifidum* and *B. breve* were most strongly associated with faecal levels of ILA, PLA and 4-OH-PLA (Supplementary Data 2g). These associations were also evident within individuals when using repeated measures correlations[32] (Extended Data Fig. 6) and across individuals at each sampling point using Spearman's rank correlations (Extended Data Fig. 7). Furthermore, re-analysing the microbiome data at the amplicon sequence variant (ASV) level showed very similar results (Extended Data Fig. 8a). Finally, qPCR targeting *B. longum* ssp. *longum*, *B. longum* ssp. *infantis*, *B. breve* and *B. bifidum* confirmed the associations to aromatic lactic acids and HMOs (Extended Data Fig. 8b). Notably, we found that both subspecies of *B. longum* were associated with the aromatic lactic acids but mainly *B. longum* ssp. *infantis* and *B. bifidum* were associated with the HMO residuals in faeces (Extended Data Fig. 8b). To further corroborate our findings regarding the relevance in early life and impact of breastfeeding, we mined a published metagenomic dataset from faecal samples from a cohort of 98 Swedish mother–infants pairs[5] for bifidobacterial metagenome-assembled genomes (MAGs) containing the *aldh* gene. This analysis revealed a

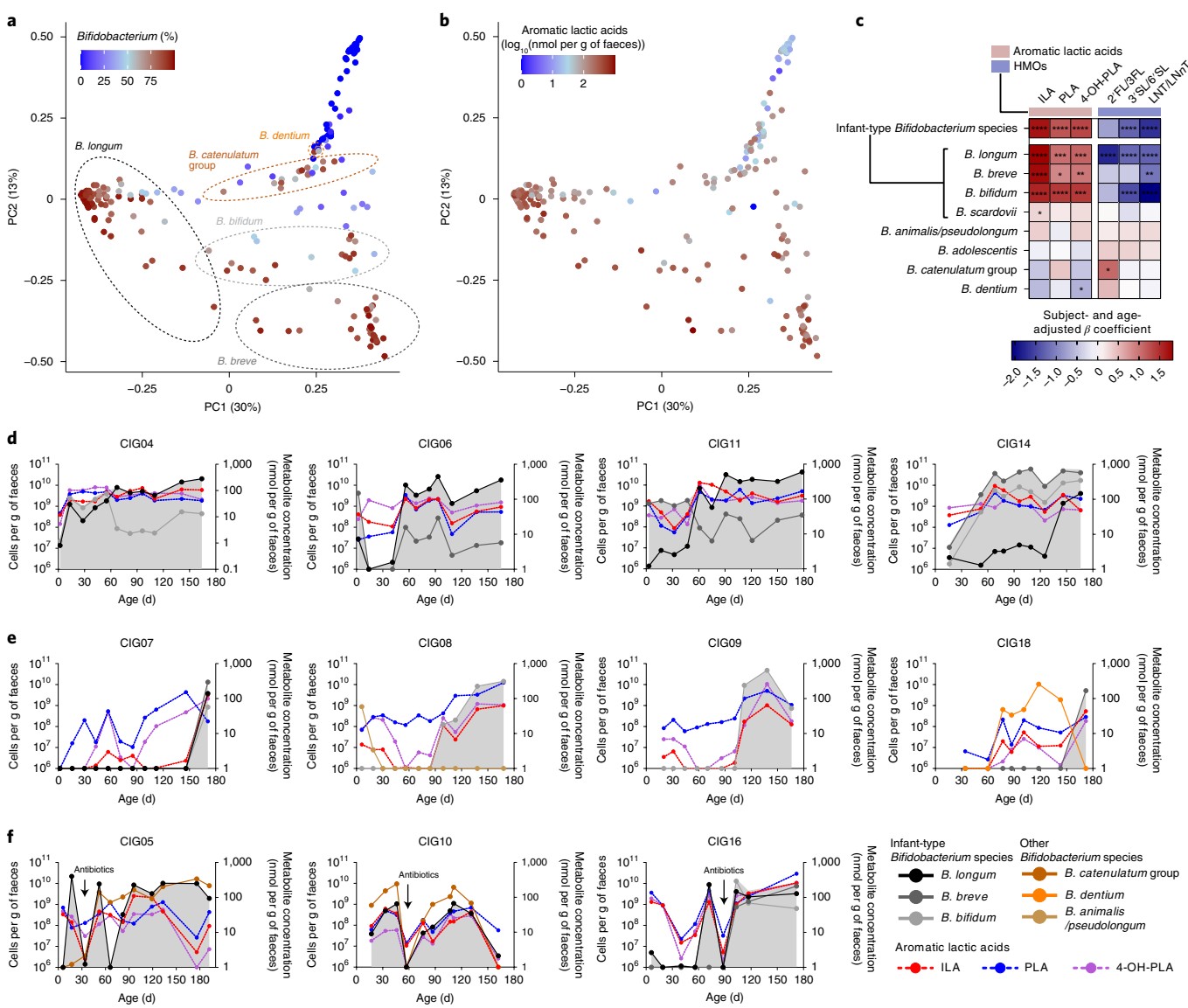

**Fig. 4 | Infant-type *Bifidobacterium* species determine faecal aromatic lactic acid concentrations during early infancy. a,b**, PCoA plots of Bray–Curtis dissimilarities (*n* = 234 (i)), coloured according to relative abundance of *Bifidobacterium* (**a**) or log₁₀-transformed concentration (nmol per g of faeces) of aromatic lactic acids (sum of ILA, PLA and 4-OH-PLA) (**b**) in faeces of infants participating in the CIG cohort. Dashed lined circles include communities dominated (relative abundance >50%) by *B. longum*, *B. bifidum*, *B. breve*, *B. catenulatum* group or *B. dentium* (*B. adolescentis*, *B. scardovii* and *B. animalis/pseudolongum* never dominated any of the communities; Extended Data Figs. 3 and 4). (i) Six samples were omitted from the analyses due to low read counts (<8,000), and one sample was omitted due to no available metabolomics data. **c**, Heatmap illustrating linear mixed-model coefficients (two-sided test, adjusted for subject and age) between the absolute abundance of *Bifidobacterium* species (cells per g of faeces) and faecal concentrations (nmol per g of faeces) of aromatic lactic acids (ILA, PLA and 4-OH-PLA, *n* = 240 (ii)) or faecal relative abundances of HMOs (2′FL/3FL, 2′/3-*O*-fucosyllactose; 3′SL/6′SL, 3′/6′-*O*-sialyllactose; LNT/LN*n*T, lacto-*N*-tetraose/lacto-*N*-*neo*tetraose; *n* = 228 (iii)) in the CIG cohort. Infant-type *Bifidobacterium* species is the sum of absolute abundances of *B. longum*, *B. breve*, *B. bifidum* and *B. scardovii*. Statistical significance was evaluated by FDR-corrected *P* values indicated by asterisks with **P* < 0.05, ***P* < 0.01, ****P* < 0.001 and *****P* < 0.0001. (ii) One sample was omitted due to no available metabolomics data. (iii) Twelve samples were omitted due to the infants no longer being breastfed and one due to no available metabolomics data. **d–f**, Absolute abundance of *Bifidobacterium* species (average relative abundance >1% of total community) and concentrations of ILA, PLA and 4-OH-PLA in faeces of selected individuals from the CIG cohort. Values of bacterial counts <10⁶ cells per g of faeces and metabolite concentrations <1 nmol per g of faeces are not shown. Summarized absolute abundance of infant-type *Bifidobacterium* species is indicated with grey background shading. **d**, Breastfed infants colonized early with infant-type *Bifidobacterium* species and with concurrent high concentrations of ILA, PLA and 4-OH-PLA through the first 6 months of life. **e**, Infants colonized late with infant-type *Bifidobacterium* species and with concurrent low concentrations of ILA, PLA and 4-OH-PLA. **f**, Infants with recorded oral antibiotics intake during the first 6 months of life. Similar dynamics of the remaining infants can be seen in Extended Data Fig. 10.

significantly higher abundance of *aldh*-containing MAGs in exclusively breastfed (compared to mixed- or formula-fed) infants at 4 months and in partially breastfed (compared to weaned) infants at 12 months of age (Extended Data Fig. 9). In addition, we found

very low abundance of *aldh*-containing MAGs in the mothers and a significant decline of these MAGs in infants after introduction to solid foods (4 versus 12 months of age). We have thus established a link between breastfeeding, degradation of HMOs, abundance

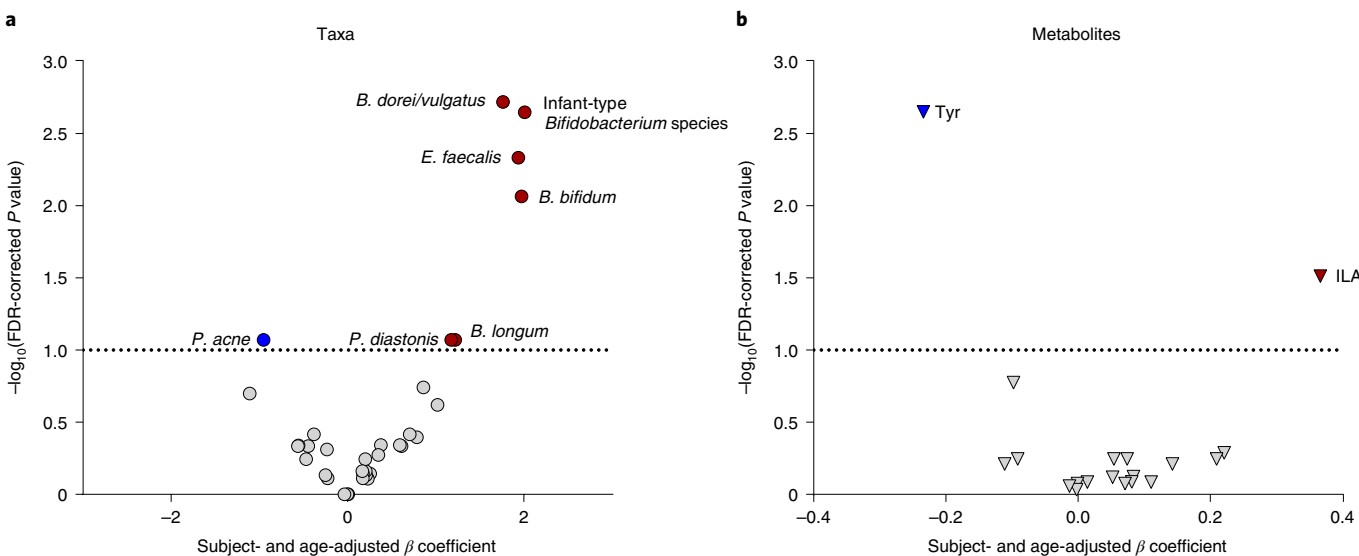

**Fig. 5 | Faecal abundances of infant-type *Bifidobacterium* species and ILA associate with AhR activity. a,b,** Scatter plot of subject- and age-adjusted linear mixed-model $\beta$ coefficients (versus FDR-corrected $P$ values, two-sided tests) obtained from associations between AhR activity of faecal water ($n=111$) from selected CIG infants (Fig. 4d–f, $n=11$) in a reporter cell line assay and absolute abundance of faecal bacterial taxa (relative abundance >0.1%, $n=40$, circles) (**a**) or quantities of aromatic amino acid metabolites ($n=19$, triangles) (**b**) measured in the same samples. Coloured circles/triangles mark taxa/ metabolite measures that are significantly positively (red) or negatively (blue) associated with AhR activity within an FDR-corrected $P$ of 0.1 (dashed line). ILA, indolelactic acid; Tyr, tyrosine.

of specific *Bifidobacterium* species and concentrations of aromatic lactic acids in early infancy.

Examination of the *Bifidobacterium* and aromatic lactic acid dynamics in each of the 25 infants during the first 6 months of life (Fig. 4d–f, Extended Data Fig. 10 and Supplementary Fig. 10) revealed that breastfed infants that were early colonized by infant-type *Bifidobacterium* species ($n=17$) consistently showed high concentrations of aromatic lactic acids in faeces (Fig. 4d and Extended Data Fig. 10a). In contrast, infants with delayed infant-type *Bifidobacterium* species colonization ($n=5$) showed considerably lower concentrations of the aromatic lactic acids, in particular of ILA, despite breastfeeding (Fig. 4e and Extended Data Fig. 10b). We noticed among the latter, infants CIG08 and CIG09 were twins, born late preterm and dominated by an operational taxonomic unit (OTU) assigned to *Clostridium neonatale* (Extended Data Fig. 3c and Supplementary Data 2c) in accordance with previous reports on *C. neonatale* overgrowth[33] and delayed *Bifidobacterium* colonization[34–37] in preterm infants. Infant CIG07 who also showed delayed colonization with infant-type *Bifidobacterium*, was mixed-fed throughout the whole period and predominantly colonized with *E. coli* and *Clostridium* species (Extended Data Fig. 3c). Infant CIG18 had relatively low faecal concentrations of aromatic lactic acids until age 172 days, when *B. breve* replaced *B. dentium* (Fig. 4e), consistent with the fact that *B. dentium* lacks the *aldh* gene while *B. breve* contains it (Fig. 2b and Supplementary Table 3). Finally, in the three infants treated with antibiotics during our study, *Bifidobacterium* species abundances were temporarily decreased simultaneously with reduced concentrations of the aromatic lactic acids (Fig. 4f). Together, these results demonstrate that HMO-using infant-type *Bifidobacterium* species determine the abundance of aromatic lactic acids in the infant gut. Yet, the impact of early/late *Bifidobacterium* colonization, preterm delivery, exposure to antibiotics and formula supplementation with respect to bifidobacterial aromatic lactic acid production warrants further investigation.

**Indolelactate modulates immune responses via AhR and HCA3.**
The tryptophan-derived metabolite ILA was consistently measured

in the faeces of breastfed infants at 0–6 months (Supplementary Table 4) and 9 months of age (Supplementary Table 2). Microbial tryptophan catabolites have been found to contribute to intestinal and systemic homoeostasis, in particular by their ability to bind the AhR[14]. Furthermore, aromatic lactic acids have been found to activate HCA3 (ref. [38]), which is involved in the regulation of immune function and energy homoeostasis[39,40]. In accordance with previous reports[15,16], we observed modest but significant dose-dependent increases in agonistic activity of ILA in both rat and human AhR reporter gene cell lines (Supplementary Fig. 11). Furthermore, all three aromatic amino lactic acids, and especially ILA, showed very potent and dose-dependent agonistic activity towards the HCA3 in a reporter cell line assay (Supplementary Fig. 12), in agreement with previous reports[38,40]. To investigate the relationship between gut microbiota, aromatic amino acid metabolites and AhR signalling, the AhR activity induced by sterile-filtered faecal water from selected CIG infants (Fig. 4d–f) was associated with the most abundant bacterial taxa (Fig. 5a) and all quantified aromatic amino acid metabolites ($n=19$) in the same samples (Fig. 5b). This revealed that, among other taxa, in particular the infant-type *Bifidobacterium* species were positively associated with AhR activity across individuals using linear mixed models adjusted for subject and age (Fig. 5a) as well as within individuals using repeated measures correlations (Supplementary Fig. 13a). Of all the aromatic amino acid metabolites measured, only faecal concentrations of ILA were significantly positively associated with AhR activity (Fig. 5b and Supplementary Fig. 13b).

Since ILA on absorption in the gut is circulated in the body[18], we next asked whether ILA affects immune function via AhR and HCA3. Since the human AhR has adapted to sense microbial tryptophan catabolites[41] and only humans and other hominids contain HCA3 (ref. [38]), we isolated immune cells from human blood and assessed the impact of ILA on their function. Specifically, we cultured isolated human CD4+ T cells under $T_{H}17$-polarizing conditions and assessed IL-22 production on exposure to ILA. Interestingly, ILA induced the production of IL-22, an effector cytokine produced by $T_{H}17$ cells after AhR stimulation[42–44], in a dose-dependent manner (Fig. 6a). Conversely, the addition of AhR antagonist CH-223191

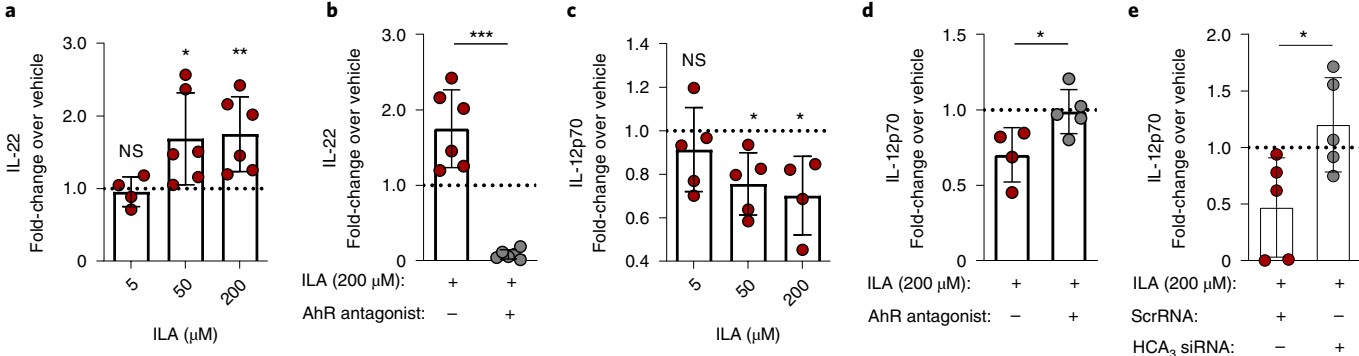

**Fig. 6 | ILA affects human immune responses via AhR and HCA$_3$. a**, Fold change in IL-22 production in purified human CD4$^+$ T cells cultured for 3 d under T$_H$17-polarizing conditions in the presence of ILA at 5, 50 and 200 μM as compared to vehicle (DMSO control). ILA$_{5μm}$ versus vehicle, $P = 0.54$; ILA$_{50μm}$ versus vehicle, $P = 0.031$; ILA$_{200μm}$ versus vehicle, $P = 0.0098$. **b**, Fold changes over vehicle (DMSO control) in IL-22 production in human purified CD4$^+$ T cells cultured under T$_H$17-polarizing conditions in the presence of 200 μM ILA with and without the AhR antagonist CH-223191. ILA$_{200μm}$ versus ILA$_{200μm}$ + AhR$_{ant}$, $P = 0.0005$. **c**, Fold change in IL-12p70 production in purified human monocytes stimulated with LPS and IFN-γ in the presence of ILA at 5, 50 and 200 μM as compared to vehicle (DMSO control). ILA$_{5μm}$ versus vehicle, $P = 0.27$; ILA$_{50μm}$ versus vehicle, $P = 0.041$; ILA$_{200μm}$ versus vehicle, $P = 0.031$. **d**, Fold changes over vehicle (DMSO control) in IL-12p70 production in purified human monocytes stimulated with LPS and IFN-γ in the presence of 200 μM ILA with and without the AhR antagonist CH-223191. ILA$_{200μm}$ versus ILA$_{200μm}$ + AhR$_{ant}$, $P = 0.033$. **e**, Fold changes over vehicle (DMSO control) in IL-12p70 production in purified human monocytes stimulated with LPS and IFN-γ in the presence of 200 μM ILA and ScrRNA or HCA$_3$ siRNA. ILA$_{200μm}$ + ScrRNA versus ILA$_{200μm}$ + HCA$_3$ siRNA, $P = 0.027$. For all panels, bars show mean ± s.d.. Each dot represents data from an individual donor ($n = 4$–6), and, for both T cells and monocytes, the measurements are derived from two-to-three independent experiments. Statistical significance was evaluated by two-sided paired (**a** and **c**) or two-sided unpaired (**b**, **d** and **e**) $t$-tests (with Welch's correction for **b** and asterisks indicating *$P < 0.05$, **$P < 0.01$, ***$P < 0.001$. For **a** and **c**, the absolute cytokine values were compared between ILA- versus vehicle-treated cells. For **b**, **d** and **e**, the fold changes over vehicle (ratios) were compared between ILA- and ILA + antagonist/siRNA-treated cells. Dashed lines indicate a fold change over vehicle of 1.

inhibited IL-22 production, further corroborating that ILA acts through AhR to induce IL-22 production (Fig. 6b). We also isolated monocytes from human blood, where both AhR (ref. [45]) and HCA$_3$ (ref. [46]) are expressed, stimulated the cells with *E. coli* lipopolysaccharide (LPS) and interferon-gamma (IFN-γ) to induce pro-inflammatory conditions, and assessed IL-12p70 production on ILA exposure. ILA reduced pro-inflammatory IL-12p70 production in a dose-dependent manner (Fig. 6c). Addition of CH-223191 blocked the ILA-induced inhibition of IL-12p70 production, confirming that ILA also acts through AhR in human monocytes (Fig. 6d). Furthermore, ILA-induced inhibition of IL-12p70 was prevented, when using knockdown of HCA$_3$ by small interfering RNA (siRNA), supporting that ILA also acts as an anti-inflammatory agent via HCA$_3$ in human monocytes (Fig. 6e). Thus, ILA affects human immune responses via AhR and HCA$_3$-dependent pathways, suggesting that *Bifidobacterium*-derived ILA is a highly relevant AhR and HCA$_3$ agonist that may impact immune responses in early life.

## Discussion

The importance of intestinal commensal bacteria in regulation of the intestinal barrier function and immune development during infancy is well established[47,48]. Yet, specifically the symbiotic role of the breastmilk-promoted *Bifidobacterium* species, which are highly abundant in many breastfed infants, remains largely unknown. Here, we identified an ALDH, which catalyses the last step of the conversion of aromatic amino acids into their respective aromatic lactic acids in the infant gut. This ALDH was different in terms of amino acid sequence compared to the ALDH previously identified in *Clostridiales* species (*C. sporogenes*, *C. cadaveris* and *P. anaerobius*)[18]. Furthermore, these species are not prevalent nor abundant in the infant gut and have previously been found to convert the aromatic lactic acids into aromatic propionic acids as end products[18]. We show that only the infant-type *Bifidobacterium* species, adapted to breastfeeding by their HMO-transport and degradation genes providing them with a colonization advantage in infant gut[13,49–52],

contain the ALDH gene. This fits the observation that *Bifidobacterium* species commonly isolated from the infant gut in vitro produce relative higher levels of ILA compared with adult- or animal-associated *Bifidobacterium* species[53]. Our enzymatic assays showed strong adaptation of ALDH towards indolepyruvic acid, resulting preferentially in the formation of ILA. Importantly, our data suggest that the production of the AhR agonist ILA by breastmilk-promoted *Bifidobacterium* is a key determinant of AhR-dependent signalling in the gut during infancy. Previous studies have found that ILA decreases inflammation in intestinal cells through activation of AhR[54,55]. Here, we show ex vivo that ILA impacts human primary immune cells via AhR- and HCA$_3$-dependent pathways. The observed dose- and AhR-dependent stimulation of IL-22 secretion by ILA may reflect a mechanism by which infant-type *Bifidobacterium* species impact intestinal homoeostasis in early life, as IL-22 for example provides protection against gastrointestinal pathogens[56–58] and promotes mucus production[57,59] and epithelial function[60]. Further, the AhR- and HCA$_3$-dependent inhibitory effect of ILA on IL-12p70 secretion by monocytes may constitute a means by which infant-type *Bifidobacterium* species contribute to the regulation of the pro-inflammatory responses to LPS derived from Enterobacteriaceae species that also often colonize the neonatal/infant gut. While the functional implications of aromatic lactic acids remain to be fully characterized, the phenomenon observed here is probably of fundamental importance, since HCA$_3$, which is only expressed in humans and other hominids[38], is involved in the regulation of immune functions and energy homoeostasis[39,40]. Furthermore, AhR signalling is involved in protection against gastrointestinal pathogens[15] and plays a key role in enhancement of intestinal barrier function[61,62], immune development[19,63–65], attenuation of induced colitis[66], autoimmunity[67–69] and metabolic syndrome[61]. In addition, ILA and PLA have been shown in vitro to have direct antibacterial[70,71] and antifungal properties[72,73]. Therefore, our findings provide a rationale for further investigation of the implications of aromatic lactic acids in infant health and immune development.

## Methods

**Human study populations and metadata.** *SKOT cohort.* The discovery cohort consisted of a random subset of 59 healthy infants (30 male, 29 female) of the observational SKOT I cohort[22]. No statistical method was used to predetermine sample size. The infants were originally recruited from Copenhagen and Frederiksberg regions by random selection from the National Danish Civil Registry[74]. Inclusion criteria were single birth and full-term delivery, absence of chronic illness and age of 9 months ± 2 weeks at inclusion. Mode of delivery, gender, age at sampling, use of medication, breast- and formula-feeding prevalence, as well as exclusive and total breastfeeding duration and age of introduction to solid foods were recorded by parental questionnaires (Supplementary Data 1a,b). Anthropometrics, full dietary assessment and other relevant metadata have been published previously[4,75]. Faecal samples were obtained at 9 months ± 2 weeks of age and were stored at –80 °C until DNA extraction, as described previously[4]. Urine samples were collected by the use of cotton balls placed in the infants' disposable nappies from which the urine was squeezed into a sterile tube and stored at –80 °C. In cases of faeces in the nappy, the urine sample was discarded. The study protocol was approved by the Committees on Biomedical Research Ethics for the Capital Region of Denmark (H-KF-2007-0003) and The Data Protection Agency (2002-54-0938, 2007-54-026) approved the study. Informed consent was obtained from all parents of infants participating in the SKOT I study. The parents did not receive any compensation.

*CIG cohort.* The validation cohort, CIG, consisted of 25 healthy infants (12 male, 13 female), mostly vaginally born (23/25) and full-term (23/25) delivered. No statistical method was used to predetermine sample size. However, on the basis of the original observations using the SKOT cohort, we estimated that 25 infants with multiple time points would be sufficient to demonstrate the dynamics between *Bifidobacterium* species and aromatic lactic acids. Infants in CIG were recruited through social media and limited to the Copenhagen region. Parents collected faecal samples approximately every second week, starting from the first week of life until 6 months of age (within weeks 0, 2, 4, 6, 8, 10, 12, 16, 20 and 24), ending with a total of 269 samples. Parents were instructed to collect faecal samples from nappies into sterile faeces collection tubes (Sarstedt) and immediately store them at –18 °C in a home freezer until transportation to the Technical University of Denmark where the samples were stored at –80 °C until sample preparation. Gender, preterm versus full-term birth, mode of delivery, infant/maternal antibiotics, feeding patterns (breastmilk versus formula) and introduction to solid foods were recorded (Supplementary Data 2a,b). The Data Protection Agency (18/02459) approved the study. The Committees on Biomedical Research Ethics for the Capital Region of Denmark confirmed that the CIG study was not notifiable according to the Act on Research Ethics Review of Health Research Projects (paragraph 1, subsection 4), as the study only concerned the faecal microbial composition and activity and not the health of the children. Informed consent was obtained from all parents of infants participating in the CIG study. In addition, parents of twins gave informed consent to publish data from the twins although the parents themselves would be able to identify their children using indirect identifiers. The parents did not receive any compensation.

**Gut microbiota analysis.** *16S rRNA gene amplicon sequencing.* Sample preparation and sequencing were performed as previously described[4] using a subset of 59 faecal samples originating from infants participating in the SKOT I cohort and 241 faecal samples from 25 infants participating in the CIG cohort (data from a total of 28 samples were missing due to insufficient sample material ($n = 1$), insufficient DNA extraction/lack of PCR product ($n = 20$), very low number of sequencing reads ($n = 6$) or resemblance of community to sequenced blank buffer DNA extraction negative controls ($n = 1$)). Briefly, DNA was extracted from 250 mg of faeces or blank buffer negative controls (PowerLyzer PowerSoil DNA isolation kit, MoBio 12855-100) and the V3 region of the 16S rRNA gene was amplified (30 s at 98 °C, 24–30 cycles of 15 s at 98 °C and 30 s at 72 °C, followed by 5 min at 72 °C) using non-degenerate universal barcoded primers including sequencing adaptors (Supplementary Table 1)[76] and then sequenced with the Ion OneTouch and Ion PGM platform with a 318-Chip v.2. Sequences from SKOT and CIG were analysed separately. Briefly, they were demultiplexed according to barcode and trimmed as previously described[76,77] in CLC Genomic Workbench (v.8.5 CLCbio, Qiagen). Quality filtering (-fastq_filter, $MAX\_EE_{(SKOT)} = 2.0$, $MAX\_EE_{(CIG)} = 1.0$), dereplication, OTU clustering (-cluster_otus, minsize 4), chimaera filtering (-uchime_ref, RDP v.9 database), mapping of reads to OTUs (-usearch_global, id 97%) and generation of OTU tables (python, uc2otutab.py) were done according to the UPARSE pipeline[78]. In QIIME (ref. [79]), OTU tables ($n_{OTUs(SKOT)} = 545$, $n_{OTUs(CIG)} = 478$) were filtered to include only OTUs with abundance across all samples above 0.005% of the total OTU counts ($n_{OTUs(SKOT)} = 258$, $n_{OTUs(CIG)} = 145$), ending up with 21,781 ± 13,110 (mean ± s.d.) reads for SKOT I and 40,156 ± 17,614 (mean ± s.d.) reads for CIG. OTU relative abundances within samples were then estimated by total sum scaling. Taxonomy was assigned to the OTUs using the rdp classifier with confidence threshold 0.5 (ref. [80]) and the GreenGenes database v.13.8 (ref. [81]). Estimating species composition in the CIG cohort, the OTUs detected with identical taxonomy were collapsed and using a cutoff of average relative abundance of 0.1%, only 39 bacterial species/taxa remained, representing

97.5% of total community (Supplementary Data 2c and Extended Data Fig. 3c). On the basis of PyNAST alignment of representative OTU sequences from each cohort separately, a phylogenetic tree was created with FastTree, as described previously[77]. Alpha diversity (Shannon index) and beta diversity (weighted and unweighted UniFrac distances, abundance weighted and binary Bray–Curtis were calculated in QIIME, with the sequencing depth rarefied to 2,000 (SKOT)/8,000 (CIG) sequences per sample. To investigate *Bifidobacterium* species composition, OTU sequences classified as *Bifidobacterium* according to the GreenGenes database v.13.8 were filtered to remove low-abundant OTUs (cutoff 0.1% of total *Bifidobacterium*) and the taxonomy of these resulting OTUs ($n_{OTUs(SKOT)} = 23$, $n_{OTUs(CIG)} = 8$) was confirmed by BLAST (ref. [82]) search against the 16S rRNA gene sequence database at the National Center for Biotechnology Information (NCBI). The top BLAST hit indicated species annotation (Supplementary Data 1g and 2e). OTUs were collapsed into *Bifidobacterium* species (*B. longum, B. bifidum, B. breve, B. catenulatum* group, *B. adolescentis, B. scardovii, B. dentium* and *B. animalis/pseudolongum*) on the basis of the top BLAST hit (Supplementary Data 1g and 2e). In addition, to validate the findings from the OTU analysis, we performed ASV analysis on the CIG cohort samples using the DADA2 pipeline v.1.14 (ref. [83]) with the demultiplexed and trimmed reads and the same cutoffs as for the OTU analysis ($MAX\_EE = 1$, ASVs filtered to include only those with abundance across all samples above 0.005% of the total ASV counts), resulting in a total of 211 ASVs and 13 ASVs assigned to *Bifidobacterium* (using the RDP database v.18) above the abundance cutoff of 0.1% of the total *Bifidobacterium* population (Supplementary Data 2f). As for OTUs, the taxonomy for individual *Bifidobacterium* ASVs was confirmed by BLAST search against the 16S rRNA gene sequence database at NCBI. Infant-type *Bifidobacterium* species were defined as the summarized abundance of *B. longum, B. bifidum, B. breve* and *B. scardovii.* CIG individuals were grouped on the basis of colonization with infant-type *Bifidobacterium* species, into those with early colonization (colonized within first month reaching average relative abundance >40% during first 6 months, $n = 17$) and late colonization (not detectable or on average <0.5% of total community within the first 3 months of life, $n = 5$), as well as those associated with antibiotics (at least one episode of recorded oral antibiotics during the first 6 months of life, $n = 3$).

*Quantitative PCR.* Total bacterial load (universal primers) and absolute abundances of *B. longum* ssp. *longum, B. longum* ssp. *infantis* (subspecies-specific primers), *B. bifidum* and *B. breve* (species-specific primers) were estimated by qPCR, using the primers listed in Supplementary Table 5. Each reaction was performed (in triplicates) with 5 μl of PCR-grade water, 1.5 μl of forward and reverse primer, 10 μl of SYBR Green I Master 2X (LightCycler 480 SYBR Green I Master, Roche) and 2 μl of template DNA, in a total volume of 20 μl. Standard curves were generated from tenfold serial dilutions of linearized plasmid (containing $10^8$–$10^0$ gene copies per μl), constructed by cloning a PCR-amplified 199-base pair (bp) fragment of the 16S rRNA gene (V3 region) of *E. coli* (ATCC 25922) or a 307-bp fragment of the *Blon0915* gene[84] of *B. longum* ssp. *infantis* (DSM 20088) or a 301-bp fragment of the *BL0274* gene[85] of *B. longum* ssp. *longum* (DSM 20219) into a pCR4-Blunt-TOPO (Invitrogen) or pCRII-Blunt-TOPO vector (Invitrogen). For *B. bifidum* (16S rRNA gene) and *B. breve* (groEL gene), tenfold serial dilutions of DNA (containing $10^7$–$10^0$ gene copies per μl) extracted from pure cultures of the type strains (DSM 20456 and DSM 20213) were used for standard curves. Plates were run on the LightCycler 480 Instrument II (Roche) with the programme including 5 min of pre-incubation at 95 °C, followed by 45 cycles with 15 s at 95 °C, 15 s at 50–72 °C and 15 s at 72 °C and a subsequent melting curve analysis including 5 min at 95 °C, 1 min at 65 °C and continuous temperature increase (ramp rate 0.11 °C s⁻¹) until 98 °C. Data were analysed with the LightCycler 480 Software (v.1.5) (Roche). Bacterial load data (using the universal primers) were used to estimate absolute abundances of each microbial taxa by multiplying with relative abundances derived from 16S rRNA gene amplicon sequencing.

***Bifidobacterium* strains and growth experiments.** *Aromatic lactic acid production by Bifidobacterium type strains. Bifidobacterium* type strains (Supplementary Table 6) were cultivated on MRSc (MRS containing 2% (w/v) glucose and supplemented with 0.05% (w/v) L-cysteine) agar plates for 48 h at 37 °C anaerobically. Single colonies were dissolved in 5.0 ml of prereduced MRSc broth and incubated for 24 h at 37 °C anaerobically with shake. The overnight (ON) cultures were washed (10,000g, room temperature, 5 min) and resuspended in sterile 0.9% NaCl water, diluted 1:20 (in triplicates) in prereduced MRSc or MRSc + HMOs (MRS without glucose but supplemented with 2.0% (w/v) HMO mixture and 0.05% (w/v) L-cysteine) and re-incubated at 37 °C anaerobically for 72 h, after which optical density $OD_{600nm}$ was measured and the culture supernatants (16,000g, 5 min, 4 °C) were analysed by UPLC–MS. The individual HMOs were kindly donated by Glycom A/S; 2′-*O*-fucosyllactose (2′FL), 3-*O*-fucosyllactose (3FL), lacto-*N*-tetraose (LNT), lacto-*N*-neotetraose (LN*n*T), 6′-*O*-sialyllactose (6′SL), 3′-*O*-sialyllactose (3′SL), together representing the three types of structures found in human breastmilk (fucosylated, sialylated and neutral core). On the basis of the HMO composition in breastmilk[86,87], these were mixed in a ratio of 53% 2′FL, 18% 3FL, 13% LNT, 5% LN*n*T, 7% 6′SL and 4% 3′SL in sterile water to obtain a representative HMO mix used in the in vitro experiments at 2% (w/v).

*Growth experiment with B. longum ssp. longum 105-A strains.* *B. longum* ssp. *longum* 105-A (JCM 31944)[30] was obtained from Japan Collection of Microorganisms (RIKEN BioResource Research Center). *B. longum* ssp. *longum* 105-A strains (WT, insertional mutant (type 4 *ldh*::pMSK127) and plasmid-complemented mutant (type 4 *ldh*::pMSK127/ pMSK128 (P*xfp*-type 4 *ldh*)); Supplementary Table 6) were cultivated on MRSc or MRSc-Chl (MRSc supplemented with 2.5 μg ml⁻¹ of chloramphenicol) agar plates for 48 h at 37 °C anaerobically. Single colonies were dissolved in 5.4 ml of MRSc or MRSc-Chl broth, tenfold serially diluted and incubated for 15 h at 37 °C anaerobically with shake. The most diluted culture (exponential phase) was washed in same medium (10,000*g*, room temperature, 5 min) and resuspended in MRSc or MRSc-Chl broth to yield OD$_{600nm}$ = 1 and subsequently diluted 1:40 in prewarmed and reduced MRSc or MRSc-Chl broth (in triplicates), before incubation at 37 °C, anaerobically with shake. The cultures were sampled (500 μl) every hour for OD$_{600nm}$ measurements and the culture supernatants (16,000*g*, 4 °C, 5 min) from early (13 h) stationary phase were analysed by UPLC–MS for aromatic amino metabolites and by GC–MS for lactate.

**Identification of *aldh* gene/operon in *Bifidobacterium* strains, metagenomic data and homology searches.** From the full genome sequences (available at NCBI Genome, https://www.ncbi.nlm.nih.gov/genome/) of *Bifidobacterium* type strains included in this study (Supplementary Table 6) all genes annotated as LDHs were aligned (gap cost 10, gap extension cost 1) and subsequently a phylogenetic tree (Algorithm = Neighbor-Joining, Distance measure = Jukes-Cantor, 100 bootstrap replications) was constructed in CLC Main Workbench (v.7.6.3, CLCbio, Qiagen). The tree was visualized by use of the FigTree software v.1.4.3 (http://tree.bio.ed.ac.uk/software/figtree/). For identification of *aldh* (type 4 *ldh*) in *Bifidobacterium* strains, all complete human gut-associated *Bifidobacterium* genomes (*n* = 127) including plasmids were retrieved from NCBI Genome and *aldh* genes were identified using NCBI tBLASTn with default settings and a cutoff of 70% identity and 70% query coverage. Aligned genomic nucleotide sequences were translated and verified to match LDHs using reciprocal BLASTx against NCBI's non-redundant database. In addition, the ALDH amino acid sequence (translated from the *aldh* nucleotide sequence) of *B. longum* ssp. *longum* 105-A was aligned (gap cost 10, gap extension cost 1) with the ALDH amino acid sequences of the *B. longum* ssp. *longum*, *B. longum* ssp. *infantis*, *B. bifidum*, *B. breve* and *B. scardovii* type strains and pairwise amino acid identity percentages were calculated in CLC Main Workbench. Potential non-bifidobacterial ALDH homologues were searched for by BLASTp analysis of the 105-A amino acid sequence against the non-redundant protein sequence database and the Swiss-Prot database using a cutoff of 60% amino acid sequence identity. Further, comparison of *aldh* gene cluster/operon in 12 *Bifidobacterium* type strains (Supplementary Fig. 4) was conducted by pairwise alignments in MBGD (Microbial Genome Database for Comparative Analysis; http://mbgd.genome.ad.jp/). The amino acid sequences of the gene cluster from *B. pseudolongum* ssp. *pseudolongum* type strain was collected from NCBI database (https://www.ncbi.nlm.nih.gov/genome/) and was used for comparison with that from *B. animalis* ssp. *animalis* type strain. Using 193 infant samples collected at 4 and 12 months of age with data on feeding practice available and data from 98 mothers[5], we used IGGsearch and IGGdb v.1.0.0 (ref. [88]) to identify *Bifidobacterium* MAGs. MAGs were included in the analysis if they passed the following criteria: --min-reads-gene=2 --min-perc-genes=40 --min-sp-quality=75. For each *Bifidobacterium* MAG identified, we used the representative genome to search for *aldh* genes. The *aldh* genes were identified using NCBI tBLASTn with default settings and a cutoff of 70% identity and 70% query coverage.

**Recombinant expression of *aldh* (type 4 *ldh*).** *Chemically competent cells for recombinant expression.* *E. coli* LMG194 ON culture (200 μl) was inoculated into 5 ml of Luria-Bertani (LB) medium and incubated at 37 °C, 250 r.p.m. until OD$_{600nm}$ = 0.5, at which time the culture was centrifuged for 5 min at 10,000*g* at 4 °C and supernatant discarded. Cell pellet was resuspended in ice-cold 1.8 ml of 10 mM MgSO₄ (Sigma, M2643) and centrifuged for 2 min at 5,000*g* at 0 °C. Supernatant was discarded and cell pellet resuspended in 1.8 ml of ice-cold 50 mM CaCl₂ (Merck, 1.02083.0250), incubated on ice for 20 min and centrifuged for 2 min at 5,000*g* at 0 °C. Cell pellet was resuspended in 0.2 ml of ice-cold 100 mM CaCl₂, 10 mM MgSO₄ and placed on ice until transformation.

*Cloning and recombinant expression.* Genomic DNA was extracted (PowerLyzer PowerSoil DNA isolation kit, MoBio 12855-100) from colony material of *B. longum* ssp. *infantis* DSM 20088ᵀ (ATCC 15697). To amplify the type 4 *ldh* gene, 50 ng of template DNA was mixed with 5 μl of 10× PCR buffer, 0.5 μl (50 mM) of dNTP mix, 1 μl (10 μM) of forward primer (ldh4_F, 5′-ACCATGGTCACTATGAACCG-3′), 1 μl (10 μM) of reverse primer (ldh4_R, 5′-AATCACAGCAGCCCCTTG-3′) and 1 μl (1 U μl⁻¹) of Platinum Taq DNA polymerase (Invitrogen, 10966-018) in a 50 μl total reaction volume. The PCR programme included 2 min at 94 °C, 35 cycles of 30 s at 94 °C, 30 s at 55 °C, 60 s at 72 °C, followed by a final extension 10 min at 72 °C. The PCR product was purified (MinElute PCR purification kit, Qiagen, 28004) and 4 μl was mixed with 1 μl of salt solution (1.2 M NaCl, 0.06 M MgCl₂) and 1 μl of pBAD-TOPO plasmid (Invitrogen, K4300-01) and incubated for 5 min at room temperature.

A total of 2 μl of the cloning mixture was transformed into 50 μl of One Shot TOP10 Competent Cells (Invitrogen, K4300-01) by gentle mix, incubation 15 min on ice and heat-shock for 30 s at 42 °C. A total of 250 μl of S.O.C. medium (Invitrogen, K4300-01) was added and incubated at 37 °C for 1 h at 200 r.p.m. and subsequently spread on LB-AMP (LB supplemented with 20 μg ml⁻¹ Ampicillin (Sigma, A9518)) agar plates and incubated at 37 °C ON. Transformants were picked and clean streaked on LB-AMP agar plates, incubated at 37 °C ON and afterwards single colonies of each transformant was inoculated into 5 ml of LB-AMP broth and incubated at 37 °C for 15 h at 250 r.p.m. Plasmid DNA was isolated (QIAprep Spin Miniprep Kit, Qiagen, 27104) from each transformant and subsequently 5 μl of plasmid DNA (80–100 ng μl⁻¹) was mixed with 5 μl (5 pmol μl⁻¹) pBAD forward (5′-ATGCCATAGCATTTTTATCC-3′) or reverse (5′-GATTTAATCTGTATCAGG-3′) sequencing primers (5 pmol μl⁻¹) and shipped for sequencing at GATC (GATC-biotech). To remove the leader peptide in pBAD-TOPO, 10 μl of plasmid (0.1 μg) with correct insert was cut with FastDigest NcoI (Thermo Scientific, FD0563) for 10 min at 37 °C and the enzyme inactivated 15 min at 65 °C. Plasmid was ligated using 1 μl (1 U μl⁻¹) T4 DNA Ligase (Invitrogen, 15224-017) for 5 min at room temperature and subsequently 2 μl of plasmid was transformed into 100 μl of chemically competent *E. coli* LMG194 cells by incubation on ice for 30 min, followed by heat-shock at 43 °C for 3 min and incubation on ice for 2 min. A total of 900 μl of LB medium was added and cells were incubated at 37 °C for 1 h at 250 r.p.m., before plating on LB-AMP agar plates and incubation at 37 °C ON. Transformants were picked, clean streaked and plasmid DNA isolated and sequenced as described above. A transformant with correct insert was selected for recombinant expression of the type 4 *ldh* gene; 2 ml of LB-AMP broth was inoculated with a single recombinant colony or the non-transformed *E. coli* LMG194 (negative control) and grown at 37 °C ON at 250 r.p.m. In 3× triplicates, 100 μl of the ON cultures (2×3× 100 μl of transformant culture + 1×3× 100 μl of non-transformed *E. coli* LMG194 culture) were diluted 100-fold into 9.9 ml of prewarmed LB-AMP/LB broth and grown at 37 °C, 250 r.p.m. until OD$_{600nm}$ ≈ 0.5, at which 9 ml of culture was added to 1 ml of mix of indolepyruvic acid, phenylpyruvic acid and 4-hydroxyphenylpyruvic acid (1 mg ml⁻¹ each). The cultures were sampled (time zero) and subsequently 100 μl of 20% L-arabinose (or 100 μl of sterile water; control for induction) was added to induce gene expression and the cultures were re-incubated at 37 °C, 250 r.p.m., before sampling at 1 and 5 h post-induction for OD$_{600nm}$ measurements and assessment of production of aromatic lactic acids. For the latter, samples were centrifuged at 16,000*g* for 5 min at 4 °C and supernatants were stored at –20 °C for UPLC–MS analyses.

**Construction of *aldh* (type 4 *ldh*) insertional mutant and *aldh* complemented strain.** *Transformation of B. longum ssp. longum 105-A.* *B. longum* ssp. *longum* 105-A cells were grown to exponential phase at 37 °C in Gifu anaerobic liquid medium (Nissui Pharmaceutical, catalogue no. 05422), harvested by centrifugation and washed twice with ice-cold 1 mM ammonium citrate buffer containing 50 mM sucrose (pH 6.0). The cells were concentrated 200 times with the same buffer and used for electroporation with settings of 10 kV cm⁻¹, 25 μF and 200 Ω. After recovery culturing in Gifu anaerobic liquid medium at 37 °C for 3 h, the cells were spread onto Gifu anaerobic agar containing antibiotics (30 μg ml⁻¹ spectinomycin and/or 2.5 μg ml⁻¹ chloramphenicol) for selection.

*Insertional mutant construction and plasmid complementation.* The type 4 *ldh* gene (BL105A_0985) of *B. longum* ssp. *longum* 105-A was disrupted by a plasmid-mediated single crossover event as described previously[89]. The plasmid used for disruption was constructed using the In-Fusion cloning kit (Clontech Laboratories, catalogue no. 639649). *E. coli* DH5α was used as a host for genetic manipulation. In brief, the internal region of the *ldh* gene (position 142–638 of the nucleotide sequence of BL105A_0985 (ref. [90]; Supplementary Fig. 5) was amplified by PCR using a primer pair Pr-580/581 (Supplementary Table 7) and ligated with the BamHI-digested pBS423 fragment carrying pUC *ori* and a spectinomycin-resistance gene[29]. The resulting plasmid pMSK127 was introduced into *B. longum* ssp. *longum* 105-A by electroporation to be integrated into type 4 *ldh* locus by single crossover recombination (type 4 *ldh*::pMSK127). Type 4 *ldh* disruption was confirmed by genomic PCR with a primer pair (Pr-543/546) designed to anneal outside of the gene (Supplementary Fig. 5 and Supplementary Table 7). The amplified fragment was also sequenced to ensure the correct recombination event. Complementation plasmid pMSK128 was constructed by ligating PCR-amplified *xfp* (xylulose 5-phosphate/fructose 6-phosphate phosphoketolase) promoter region (P*xfp*) and the type 4 *ldh* coding region with PstI- and SalI-digested pBFS38 (ref. [91]) using the In-Fusion cloning kit, by which type 4 *ldh* was placed under the control of P*xfp*. Primer pairs of Pr-598/Pr-599 and Pr-600/Pr-601 were used for amplifying P*xfp* from pBFS48 (ref. [91]) and the type 4 *ldh* gene from the *B. longum* ssp. *longum* 105-A genome, respectively (Supplementary Table 7). The resulting plasmid was electroporated into type 4 *ldh*::pMSK127 to give type 4 *ldh*::pMSK127/ pMSK128 (P*xfp*-type4_*ldh*) (Supplementary Fig. 5).

**Biochemical characterization of ALDH (type 4 LDH).** *Recombinant expression and purification.* Type 4 LDH (BL105A_0985) was recombinantly expressed as a non-tagged form. The gene was amplified by PCR using the genomic DNA

of *B. longum* ssp. *longum* 105-A as a template and a primer pair of Pr-617 (5′-GGTGGTGGTGCTCGAGTCACAGCAGCCCCTCGCAG-3′) and Pr-635 (5′-AAGGAGATATACATATGGTCACTATGAACCGC-3′). Underlined bases indicate 15 bp for In-Fusion cloning (Clontech). The amplified DNA fragment was inserted into the NdeI and XhoI site of pET23b(+) (Novagen) using an In-Fusion HD cloning kit (Clontech). The resulting plasmid was introduced into *E. coli* BL21 (DE3) Δ*lacZ* carrying pRARE2 (ref. [89]) and the transformant was cultured in LB medium supplemented with ampicillin (100 μg ml$^{-1}$) and chloramphenicol (7.5 μg ml$^{-1}$). When OD$_{600nm}$ reached 0.5, isopropyl β-D-thiogalactopyranoside was added at a final concentration of 0.02 mM to induce the protein expression. The culture was incubated for 4 d at 18 °C, harvested by centrifugation and resuspended in 50 mM potassium phosphate buffer (KPB; pH 7.0) supplemented with 1 mM 2-mercaptoethanol (2-ME) and 200 μM phenylmethane sulfonyl fluoride. Following cell disruption by sonication, the cleared lysate was saturated with ammonium sulfate (40–60%). The resulting precipitate was dissolved, dialysed against 20 mM KPB (pH 7.0) containing 1 mM 2-ME and concentrated by Amicon Ultra 10 K centrifugal device (Merck Millipore). The sample was then loaded onto an Affigel blue column (Bio-Rad) preequilibrated with 20 mM KPB (pH 7.0) containing 1 mM 2-ME and eluted by the same buffer containing 1 M NaCl. The protein was further purified by a Mono 5/50 (GE Healthcare; a linear gradient of 0–1 M NaCl in 20 mM Tris-HCl (pH 8.0) containing 1 mM 2-ME) and Superdex 200 Increase 10/300 GL column (GE Healthcare; 10 mM KPB (pH 7.0) containing 50 mM NaCl and 1 mM 2-ME). Protein concentration was determined by measuring the absorbance at 280 nm based on a theoretical extinction coefficient of 26,470 M$^{-1}$ cm$^{-1}$.

*Enzyme assay.* The standard reaction mixture contained 100 mM KPB (pH 8.0), 1 mM 2-ME, 0.1 mM β-NADH and the substrate. The reaction was initiated by adding the enzyme and the mixture was incubated at 37 °C for an appropriate time, in which the linearity of the reaction rate was observed. The substrate concentrations were varied between 0.01 and 0.25 mM for indolepyruvic acid 1.5 and 10.5 mM for phenylpyruvic acid, 2 and 24 mM for 4-hydroxyphenylpyruvic acid and 2.5 and 40 mM for pyruvic acid. The enzyme was used at the concentrations of 0.22 nM for indolepyruvic acid, 1.47 nM for phenylpyruvic acid, 0.12 nM for 4-hydroxyphenylpyruvic acid and 88.50 nM for pyruvic acid. The reducing reactions of phenylpyruvic acid and pyruvic acid were continuously monitored by measuring the decrease of the absorbance at 340 nm (NADH consumption). When 4-hydroxyphenylpyruvic acid and 4-indolepyruvic acid were used as the substrates, the reaction products 4-OH-PLA and ILA were quantified by HPLC after the termination of the reactions by adding 5% (w/v) trichloroacetic acid. HPLC analysis was performed using a Waters e2695 separation module (Waters) equipped with a LiChrospher 100 RP-18 column (250 × 4 mm, φ = 5 μm; Merck Millipore) at 50 °C. Following equilibration with a mixture of 10% solvent A (50% methanol, 0.05% trifluoroacetic acid) and 90% solvent B (0.05% trifluoroacetic acid) at a flow rate of 1 ml min$^{-1}$, the concentration of solvent A was linearly increased to 100% for 25 min and maintained at 100% for additional 15 min. The 4-OH-PLA and ILA were detected by a Waters 2475 Fluorescence Detector with $\lambda_{ex}$ 277 nm and $\lambda_{em}$ 301 nm and $\lambda_{ex}$ 282 nm and $\lambda_{em}$ 349 nm, respectively. The standard curves were created using the known concentrations of both compounds. Experiments were performed at least in duplicate. Physicochemical property of the enzyme was examined by using 1 mM phenylpyruvic acid as a substrate. The effects of metal ions (0.1 mM each) on the enzyme activity was examined using 50 mM MES (2-(*N*-morpholino) ethanesulfonic acid) buffer (pH 7.0). EDTA was added at the final concentration of 0.1, 0.5 or 1 mM. The optimal pH was determined using 50 mM KPB (pH 6.0–8.5) and TAPS (*N*-Tris(hydroxymethyl)methyl-3-aminopropanesulfonic acid) buffer (pH 8.0–9.0). The thermostability was evaluated by the residual activities after incubating the enzyme (1.0 mg ml$^{-1}$ in 10 mM KPB (pH 7.0) containing 50 mM NaCl and 1 mM 2-ME) at the indicated temperatures for 30 min before the assay. Fructose 1,6-bisphosphate, shikimate-3-phosphate, D-erythrose-4-phosphate and phosphoenolpyruvic acid were added to the reaction mixtures at the concentrations of 0.1 and 1 mM to examine their heterotropic effects. KPB, TAPS buffer or HEPES (4-(2-hydroxyethyl)-1-piperazineethanesulfonic acid) buffer (pH 8.0 each) containing 1 and 4 mM phenylpyruvic acid as a substrate were used. The effect of phosphate ion was analysed by adding various concentration of KPB (pH 8.0) into 10 mM HEPES buffer (pH 8.0). All experiments were conducted at least in duplicate. In the subsequent kinetic analysis, we used phosphate ion at the concentration of 100 mM because (1) no saturation was obtained for phosphate under the tested conditions (Supplementary Fig. 9a), (2) the intracellular phosphate concentration in Gram-positive bacteria is known to be 130 mM at maximum[92] and (3) the strong homotrophic effect of the substrate phenylpyruvic acid was observed only in the presence of 10 mM phosphate ion.

**In vivo monocolonization experiments.** Germ-free (GF) Swiss Webster mice (Tac:SW, originally obtained from Taconic Biosciences) were bred and housed within GF isolators (Scanbur) in type II Makrolon cages (Techniplast) with bedding, nesting material, hiding place and a wooden block at the National Food Institute, Technical University of Denmark. The mice were fed an irradiated standard Altromin 1314 chow (Brogaarden) and the environment was maintained

on a 12 h light/12 h dark cycle at a constant temperature of 22 ± 1 °C, with air humidity of 55 ± 5% relative humidity and change of air 50 times per hour. The GF condition of the mice before inoculation of bacteria was confirmed by plating of faecal sample suspensions on blood agar plates (Statens Serum Institut) incubated both aerobically and anaerobically. In two separate experiments, pregnant GF mice were randomized to be colonized with either *B. longum* 105-A WT (*n* = 4) or *aldh* (type 4 *ldh*) mutant (*n* = 5) by a single oral gavage (200 μl, ≈5 × 10$^7$ c.f.u. per dose) 1 week before giving birth. The monocolonized offspring (*n*$_{wildtype}$ = 21, 12 males and 9 females; *n*$_{aldhmutant}$ = 29, 18 males and 11 females) were euthanized at 4 weeks of age by cervical dislocation and dissected to collect caecal contents. Successful colonization with *B. longum* and absence of contamination in monocolonized offspring was confirmed by cultivation of caecal content on MRSc and blood agar plates incubated both aerobically and anaerobically. Aromatic lactic acids were quantified from caecal content. All mouse experiments were approved by the Danish Animal Experiments Inspectorate (license no. 2015-15-0201-00553) and carried out in accordance with existing Danish guidelines for experimental animal welfare.

**Metabolomics.** *Chemicals.* Authentic standards of the aromatic amino acids and derivatives (Supplementary Table 1) were obtained from Sigma-Aldrich, whereas isotope-labelled aromatic amino acids used as internal standards (L-phenylalanine (ring-d5, 98%), L-tyrosine (ring-d4, 98%), L-tryptophan (indole-d5, 98%) and indoleacetic acid (2,2-d2, 96%)) of the highest purity grade available were obtained from Cambridge Isotope Laboratories.

*Extraction of metabolites from faecal samples.* Faecal samples (100–500 mg) from the SKOT (*n* = 59) and the CIG cohort (*n* = 267, data from two samples missing due to insufficient sample material (*n* = 1) and problems detecting the internal standards (*n* = 1)) were diluted 1:2 with sterile MQ water, vortexed for 10 s and centrifuged at 16,000*g*, 4 °C for 5 min. Subsequently, the supernatant liquor was transferred to a new tube and centrifuged again at 16,000*g*, 4 °C for 10 min. Finally, an aliquot of 150–300 μl was stored at –20 °C. All samples were later thawed at 4 °C, centrifuged at 16,000*g*, 4 °C for 5 min, and diluted in a total volume of 80 μl of water corresponding to a 1:5 dilution of the faecal sample. To each sample, 20 μl of internal standard mix (4 μg ml$^{-1}$) and 240 μl of acetonitrile were added. The tubes were vortexed for 10 s and left at –20 °C for 10 min to precipitate the proteins. The tubes were then centrifuged at 16,000*g*, 4 °C for 10 min, and each supernatant (320 μl) was transferred to a new tube, which was dried with nitrogen gas. Subsequently, the residues were reconstituted in 80 μl of water (equalling a 1:5 dilution of the faecal sample with internal standards having a concentration of 1 μg ml$^{-1}$), vortexed for 10 s, centrifuged at 16,000*g*, 4 °C for 5 min and transferred to an LC vial, which was stored at −20 °C until analysis.

*Extraction of metabolites from urine samples.* Urine samples (*n* = 49) from the SKOT cohort were thawed in a refrigerator and all procedures during the sample preparation were carried out at 0–4 °C using an ice bath. The subjects were randomized between analytical batches by placing all the samples from the each subject in the same 96-well plate. The run order of the samples was randomized within the analytical batch. Urine samples were centrifuged at 3,000*g* for 2 min at 4 °C. A total of 150 μl of each urine sample were added to separate wells and diluted with 150 μl of diluent (MQ water: formic acid (99.9:0.1, v/v) / internal standard mixture (100 μg ml$^{-1}$) (90:10, v/v). A blank sample (diluent), standard mixture of external standard containing 44 biologically relevant metabolites (metabolomics standard)[93] and pooled sample containing equal amounts of each sample (20 μl) were added to spare wells as quality control samples. The plates were stored at −80 °C until the analysis. Immediately before analysis, the plates were thawed and mixed by vortex stirring for 10 min.

*Extraction of metabolites from in vitro fermentation samples.* Supernatants from in vitro fermentations were thawed at 4 °C, centrifuged at 16,000*g*, 4 °C for 10 min, before 80 μl was transferred to a new tube. To each sample, 20 μl of internal standard (40 μg ml$^{-1}$) and 300 μl of acetonitrile were added. The tubes were vortexed for 10 s and left at –20 °C for 10 min to precipitate the proteins. Following, the tubes were centrifuged at 16,000*g*, 4 °C for 10 min before 50 μl of each sample was diluted with 50 μl of sterile water and transferred to an LC vial (equalling a 1:10 dilution of the sample with internal standards having a concentration of 1 μg ml$^{-1}$).

*Metabolic profiling of faecal, caecal and in vitro samples using UPLC–MS.* Aromatic amino acids and derivatives (Supplementary Table 1) of faecal and in vitro samples were quantified by a semiquantitative UPLC–MS method[94]. In brief, samples were analysed in random order. For the analysis of the CIG faecal samples, a pooled quality control (QC) sample was injected for every ten samples. In all cases, five standard mix solutions (0.1, 0.5, 1, 2 and 4 μg ml$^{-1}$) were analysed once for every ten samples to obtain a standard curve for every ten samples. For each sample, a volume of 2 μl was injected into a ultraperformance liquid chromatography quadrupole time-of-flight mass spectrometry (UPLC-QTOF-MS) system consisting of Dionex Ultimate 3000 RS liquid chromatograph (Thermo Scientific) coupled to a Bruker maXis time-of-flight mass spectrometer equipped with an electrospray interphase (Bruker Daltonics) operating in positive mode.

The analytes were separated on a Poroshell 120 SB-C18 column with a dimension of $2.1 \times 100$ mm and $2.7$ μm particle size (Agilent Technologies) as previously published[94]. Aromatic amino acids and derivatives were detected by selected ions and semiquantified by isotopic internal standards with similar molecular structures as listed in Supplementary Table 1. The recoveries of the internal standards varied but were, relative to each other, in general rather consistent (Supplementary Fig. 14) emphasizing that while the absolute concentrations may not be accurate due to lack of isotope-labelled internal standards for each single analyte, the relative metabolite concentrations across samples were robust with the applied LC-MS method. Data were processed using QuantAnalysis v.2.2 (Bruker Daltonics) and bracket calibration curves for every ten lumen samples were obtained for each metabolite. The calibration curves were established by plotting the peak area ratios of all of the analytes with respect to the internal standard against the concentrations of the calibration standards. The calibration curves were fitted to a quadratic regression.

For untargeted metabolomics, the raw UPLC–MS data, obtained by analysis of the CIG faecal samples in positive ionization mode, were converted to mzXML files using Bruker Compass DataAnalysis 4.2 software (Bruker Daltonics) and preprocessed as previously reported[95] using the R packpage XCMS (v.1.38.0; ref. [96]). Noise filtering settings included that features should be detected in minimum 50% of the samples. A data table was generated comprising mass-to-charge ($m/z$), retention time and intensity (peak area) for each feature in the every sample. The data were normalized to the total intensity and log-transformed. Subsequently, features with a coefficient of variation >0.3 in the QC samples and features with a retention time <0.5 min were excluded from the data. Parent ion masses of HMO compounds of interest (2′FL/3FL, LNT/LN$n$T, 3′SL/6′SL) were searched in the cleaned dataset with 0.02 Da $m/z$ and 0.02 min retention time tolerance. Subsequently, the identities of the features of interest were confirmed at level 1 (ref. [97]) by tandem mass spectrometry and comparison to authentic standards (Supplementary Table 8). Of notice, HMO isomers could not be distinguished with the method applied due to identical retention times.

*Metabolic profiling of urine samples using UPLC–MS.* The samples were analysed by UPLC-QTOF-MS equipped with an electrospray ionization (ESI) (Waters Corporation). Reverse phase HSS T3 $C_{18}$ column ($2.1 \times 100$ mm, $1.8$ μm) coupled with a precolumn (VanGuard HSS T3 C18 column ($2.1 \times 5$ mm, $1.8$ μm)) were used for chromatographic separation. A total of 5 μl of each well was injected into the mobile phase A (0.1% formic acid in MQ water), mobile phase B (10% 1 M ammonium acetate in methanol), mobile phase C (methanol) and mobile phase D (isopropanol). Mobile phase gradient during the run time of 10 min was as follows: start condition (100% A), 0.75 min (100% A), 6 min (100% C), 6.5 min (70% B, 30% D), 8 min (70% B, 30% D), 8.1 min (70% B, 30% D), 9 min (100% A) and 10 min (100% A). The flow rate gradient was as follows: start condition (0.4 ml min⁻¹), 0.75 min (0.4 ml min⁻¹), 6 min (0.5 ml min⁻¹), 6.5 min (0.5 ml min⁻¹), 8 min (0.6 ml min⁻¹), 8.10 min (0.4 ml min⁻¹), 9 min (0.4 ml min⁻¹) and 10 min (0.4 ml min⁻¹). ESI was operated in negative mode with 3.0 kV capillary probe voltage. The cone voltage and the collision energy were set at 30 kV and 5 eV, respectively. Ion source and desolvation gas (nitrogen) temperature were 120 and 400 °C while sampling cone and desolvation gas flow rates were 50 and 1,000 l h⁻¹. Scan time set as 0.08 s with 0.02 s interscan time for both modes. Data were acquired in centroid mode with mass range between 50 and 1,500 Da. Leucine-enkephalin (500 ng ml⁻¹) was infused as the lock-spray agent to calibrate the mass accuracy every 5 s with 1 s scan time. Quality control samples were used to evaluate possible contamination, monitoring the changes in mass accuracy, retention time and instrumental sensitivity drifts[93,98].

The raw data were converted to netCDF format using DataBridge Software v.3.5 (Waters) and imported into MZmine v.2.28 (ref. [99]). A subset of samples was used to optimize the preprocessing parameters for the positive and negative mode data separately. Optimized preprocessing parameters are listed in Supplementary Table 9. Data preprocessing was used with the following steps: mass detection, chromatogram builder, chromatogram deconvolution, deisotoping, peak alignment and gap filling. After the preprocessing, each detected peak was represented by a feature defined with a retention time, $m/z$ and peak area.

The data matrix was imported into MATLAB R2015b (MathWorks). Features that were present in the blanks, were very early and late eluting (retention time < 0.30 and retention time > 9.46 min), potential isotopes, duplicates as well as features with masses indicating multiple charges were removed from the dataset using an in-house algorithm. The data were normalized using unit length normalization to correct the variation in urine concentration. Parent ion masses of the aromatic lactic acids (ILA, PLA and 4-OH-PLA) were searched in the cleaned dataset with 0.02 Da $m/z$ and 0.02 s retention time tolerance. A linear regression model was used feature-wise to correct for batch differences and instrumental sensitivity drifts[100]. The aromatic lactic acids were confirmed at level 1 (ref. [97]) by comparison to authentic standards and by tandem mass spectrometry using the same experimental conditions (Supplementary Figs. 1–3).

*Lactic acid production by B. longum ssp. longum 105-A strains using GC–MS.* The lactic acid production of the *B. longum* ssp. *longum* 105-A WT, type 4 *ldh* mutant and type 4 *ldh* complemented strains were assessed in supernatants obtained after 13 h of growth (early stationary phase) by gas chromatography–mass spectrometry (GC–MS) on methyl chloroformate derivatization using a slightly modified version of the protocol previously described[101]. All samples were analysed in a randomized order. Analysis was performed using GC (7890B, Agilent Technologies) coupled with a quadrupole detector (59977B, Agilent Technologies). The system was controlled by ChemStation (Agilent Technologies). Raw data was converted to netCDF format using Chemstation, before the data was imported and processed in Matlab R2014b (Mathworks) using the PARADISe software[102].

**Rat AhR reporter gene assay.** Rat hepatoma cells (H4IIE) stably transfected with a luciferase reporter gene under the control of AhR (pGudLuc1.1) were used. The cells were kindly provided by M. S. Denison (University of California, Davis). The assay was conducted as previously described[103], where cells were incubated for ~22 h in Minimum Essential Medium (MEM) α with 1% fetal bovine serum (FBS) and 1% penicillin/streptomycin/fungizone. Chemical exposure was performed for 24 h and successively luminescence was measured. Cell viability was analysed by measuring ATP levels with the CellTiter-Glo Luminescent Assay according to the manufacturer's instruction (Promega). As a positive control, 2,3,7,8-tetra chlorodibenzo-p-dioxin was used. Three experiments in triplicates were conducted with five twofold dilutions of ILA and IAld ranging from 12.5 to 200 μM with a constant vehicle concentration in all wells. Further, sterile-filtered faecal water (10 mg faeces per ml MQ water) obtained from all samples ($n = 119$) of 11 selected CIG infants (Fig. 4d–f) were run in technical triplicates in the assay. Only mild toxicity that did not correlate with AhR-induced luminescence signal was observed for some faecal water samples.

**Human AhR reporter gene assay.** ILA and IAld (positive control)[15] were tested for activation of the human AhR. AhR Reporter Cells from Indigo Biosciences (catalogue no. IB06001) that include a luciferase reporter gene functionally linked to an AhR-responsive promoter were used. The assay was run according to the instructions of the manufacturer (technical manual v.6.0) with the reference agonist MeBIO as the positive control. Three experiments in triplicates were conducted with five twofold dilutions of ILA and IAld ranging from 12.5 to 200 μM with a constant vehicle concentration in all wells. No cytotoxicity was observed for any of the tests as determined by a resazurin toxicity assay.

**Human HCA₃ receptor assay.** The aromatic lactic acids (ILA, PLA and 4-OH-PLA) were tested for activation of the HCA₃ receptor, which is a Gα$_i$-coupled receptor (GPCR). The cAMP Hunter eXpress GPR109B CHO-K1 GPCR Assay for chemiluminescence detection of cAMP was used (DiscoveRx Corporation, catalogue no. 95-0141E2CP2M). Following ligand stimulation of cells overexpressing the HCA₃ receptor, the functional status of the receptor was monitored by measuring cellular cAMP levels using a homogeneous, competitive immunoassay based on Enzyme Fragment Complementation technology. The assay was run in agonist mode in a 96-well plate format according to the instructions of the manufacturer (DiscoveRx Corporation) in the presence of 15 μM forskolin. Eleven threefold dilutions of ILA ranging from 0.03 to 1,574 μM and of PLA and OH-PLA ranging from 0.02 to 1,000 μM were tested twice in duplicates.

**Ex vivo stimulation of human immune cells.** Human buffy coats were acquired from the Copenhagen University Hospital (Rigshospitalet) from healthy anonymous donors. Use of the buffy coat material from healthy anonymous donors was approved by the Blood bank at Rigshospitalet, Copenhagen, under the jurisdiction of Region H. Prior written informed consent was obtained according to the Declaration of Helsinki. Blood samples were handled in accordance with guidelines put forward in the 'Transfusion Medicine Standards' by the Danish Society for Clinical Immunology (www.dski.dk).

**Isolation, cell culture and stimulation of T cells.** Peripheral blood mononuclear cells (PBMCs) were isolated from whole blood by density centrifugation on Lymphoprep and cryopreserved at −150 °C in FBS with 10% DMSO until the day of cell culture. For cultivation, PBMCs were thawed and CD4⁺ T cells isolated using EasySep Human CD4⁺ T Cell Isolation Kit (Stemcell, 17952) following the manufacturers protocol. In short, ~$2.5 \times 10^7$ PBMCs were incubated for 5 min at room temperature in 500 μl of IMDM-medium containing 50 μl of CD4⁺ T cell isolation cocktail, followed by the addition of 50 μl of RapidSpheres. Subsequently, the volume was topped up to 2.5 ml with IMDM-medium, the cells placed in an EasySep magnet (Stemcell) and incubated at room temperature for 3 min. The pure CD4⁺ T cell fraction was obtained by pouring the enriched non-bound cell fraction into a new tube. Enriched CD4⁺ T cells were cultured in T$_H$17-polarizing culture medium (IMDM supplemented with 10% FCS, 20 mM HEPES (pH 7.4), 50 μM 2-mercaptoethanol, 2 mM L-glutamine and penicillin–streptomycin (10,000 U ml⁻¹), 30 ng ml⁻¹ IL-6, 10 ng ml⁻¹ IL-1β, 0.5 ng ml⁻¹ TGFβ-1, 10 ng ml⁻¹ IL-23, 25 μl ml⁻¹ ImmunoCult Human CD3/CD28 T cell activator for 3 d at 37 °C and 5% CO₂ in Falcon polystyrene 48-well plates (Thermo Fisher, 10059110). Each culture condition contained 0.2% DMSO with or without the indicated amounts of ILA and/or the AhR-inhibitor CH-223191. After 3 d of culture, supernatants were collected for ELISA and frozen down until further use. The ELISA to detect IL-22

was performed in technical duplicates using the ELISA MAX Deluxe Set Human IL-22 kit (Biolegend, 434504) following the supplied manufacturer's protocol. In short, a Nunc MaxiSorb flat-bottom 96-well plate (Thermo Fischer, 44-2404-21) was coated for 12 h at 4 °C with IL-22 coating antibody followed by four rounds of washing with PBS + 0.05% Tween-20. The washed plate was blocked with supplied assay diluent A buffer for 1 h at room temperature and 400 r.p.m., washed four more times with PBS + 0.05% Tween-20 and incubated with cell culture supernatants for 2 h at room temperature and 400 r.p.m. Serially diluted standard controls and a blank control were included as a reference. To detect bound IL-22, the plate was washed four times with PBS + 0.05% Tween-20 and incubated with IL-22 detection antibody for 1 h at room temperature and 400 r.p.m. After four further washing steps with PBS + 0.05% Tween-20, Avidin-HRP was added for 30 min at room temperature and 400 r.p.m. To detect HRP activity, the plate was washed five times with PBS + 0.05% Tween-20 followed by an incubation with Solution F substrate solution in the dark at room temperature. HRP activity was stopped after 20 min using 1 M $H_2SO_4$ and the optical density recorded (absorption at 450 nm) using a PowerWave HT Microplate Spectrophotometer (BioTek Instruments). Values below limit of detection (16 pg ml$^{-1}$) of the kit were set to LOD/2. Sources and identifiers of all reagents used are given in Supplementary Table 10.

**Isolation, cell culture and stimulation of monocytes.** PBMCs were isolated by Ficoll-Paque (GE Healthcare) density centrifugation. Monocytes were isolated to >92% purity using the Pan Monocyte Isolation kit (Miltenyi Biotec). Cells were stained with diluted (1:25) CD14-PE-Cy7 (eBiosciences) and diluted (1:25) CD16-FITC (Biolegend) to determine the purity of monocytes (Supplementary Fig. 15) by flow cytometry (BD FACS Canto II). Monocytes were cultured in culture medium (RPMI 1640 (Lonza) containing 2 mM L-glutamine (Lonza), 10% heat-inactivated fetal bovine serum (Lonza), 1% penicillin/streptomycin (Lonza), 50 μM 2-ME (Sigma-Aldrich)) in a humidified 37 °C, 5% $CO_2$ incubator. ILA (Sigma I5508), dissolved in a maximum of 0.1% DMSO, was added to the cells, with the final concentrations of ILA 5, 50 or 200 μM, respectively. Vehicle (0.1% DMSO) was added as a control. LPS (TLR4 ligand, *E. coli* O26:B6, Sigma L2654) at 100 ng ml$^{-1}$ (final concentration) and IFN-γ (RD285-IF-100) at 10 ng ml$^{-1}$ (final concentration) were then added to the cells and they were stimulated for 18 h. To determine the contribution of AhR, monocytes were pretreated for 1 h before addition of above compounds with the AhR antagonist CH-223191 (Sigma C8124) at 10 μM. For HCA$_3$ silencing, where no specific antagonist is available, we performed reverse transfection using Lipofectamine RNAiMAX in Opti-MEM (Life Technologies) added 10 nM of scrambled siRNA (ScRNA) (Thermo Fisher 4390846) or HCA$_3$-specific siRNA (Thermo Fisher 4427037), reaching a maximum knockdown of 79% on the basis of qPCR validation (HCA$_3$ primer (Thermo Fisher 4448892) versus GAPDH) as compared to ScRNA targeted cells. Monocytes were pretreated with the siRNA constructs for 24 h before stimulation as above. Viability of cells was >97% as analysed by flow cytometry (BD FACS Canto II) using SYTOX AAD. Supernatants of stimulated cells were harvested after 18 h and kept at −80 °C until analysis. IL-12p70 was quantified using Meso Scale Discovery kits as previously detailed[104].

**Statistics and reproducibility.** All experiments were performed with full factorial (biological and technical) replication. Data collection and analysis were not performed blind to the conditions of the experiments. No data were excluded from the analyses, except in the CIG cohort, six samples were omitted from the PCoA due to low read counts (<8,000), for correlation analyses between *Bifidobacterium* species and HMO residuals in faeces, 12 samples with no reported breastfeeding were excluded. One donor was excluded from the 200 μM indolelactate (ILA) stimulation of monocytes, since something went wrong during the stimulation. Statistical analyses were performed using QIIME v.1.9 (ref. [79]), R v.3.1 (ref. [105]) and GraphPad Prism v.8.1 (GraphPad Software). If data were normally distributed (evaluated by visual inspection and D'Agostino–Pearson test), parametric statistical tests were used, whereas non-parametric tests were used with non-normally distributed data. OTU distance/dissimilarity matrices were generated from OTU tables with rarefied read counts (2,000 sequences per sample for SKOT and 8,000 sequences per sample for CIG) and ordination (PCoA, beta_diversity. py script, default settings) and statistical tests (ADONIS and PERMDISP tests, compare_categories.py script, permutations = 999, default settings) of OTU distance/dissimilarity matrices were performed in QIIME. PCoA plots were illustrated (ggplot function) in R using the ggplot2 (v.3.3.3) package[106]. PCA of metabolite concentrations was performed in R using the prcomp function with zero-centring and autoscaling and illustrated using the ggbiplot function[107] within ggplot2. Spearman's rank correlations were performed in GraphPad Prism, whereas repeated measures correlation analyses and linear mixed models were performed in R using the packages rmcorr (v.0.4.3)[32] (rmcorr function with default settings) and maaslin2 (v.1.0.0)[31], respectively. In MaAsLin2 linear mixed models (Maaslin2 function) subject and age were included as random effects and the individual faecal aromatic lactic acids or HMOs as fixed effects, with default settings, except that total sum scaling normalization was not performed when including absolute abundances of bacterial taxa. The faecal HMO data were log$_{10}$-transformed. Heatmaps and hierarchical clustering of correlation coefficient were generated

in R using the heatmap.2 function (default settings) within the gplots (v.3.1.1) package[108] and visualized in GraphPad Prism. Longitudinal metabolite and taxonomic abundance were modelled using LOESS regression and associations between taxa, metabolites and in vitro AhR activity (log$_{10}$-transformed) were modelled using linear regression in R using the ggplot function (method = 'loess' or 'lm') within the ggplot2 package[106]. For enzyme kinetics, the parameters ($k_{cat}$, $K_{0.5}$ and Hill coefficient $n_H$) were calculated by curve-fitting the experimental data to the Hill equation, using GraphPad Prism. Two-tailed paired or unpaired Student's *t*-test or two-tailed non-parametric Mann–Whitney *U*-test were performed when comparing two groups. For comparison of more than two groups, statistical significance was evaluated by one-way analysis of variance (ANOVA) or the non-parametric Kruskal–Wallis test. $P < 0.05$ were considered statistically significant. When applicable, $P$ values were corrected for multiple testing by the Benjamini–Hochberg false discovery rate (FDR)[109] using a cutoff of 0.1.

**Reporting Summary.** Further information on research design is available in the Nature Research Reporting Summary linked to this article.

## Data availability

All 16S rRNA gene amplicon sequencing data were deposited in the Sequence Read Archive (SRA) under the BioProjects PRJNA273694 (SKOT) and PRJNA554596 (CIG). The following databases were used: GreenGenes 16S rRNA database (https://greengenes.secondgenome.com/), RDP database (https://rdp.cme.msu.edu/), 16S rRNA gene sequence database at NCBI (https://ncbiinsights.ncbi.nlm.nih.gov/2020/02/21/rrna-databases/), non-redundant protein sequence database at NCBI (https://www.ncbi.nlm.nih.gov/protein/), Swiss-Prot database (https://www.uniprot.org/), MBGD (Microbial Genome Database for Comparative Analysis; http://mbgd.genome.ad.jp/) and the genome database at NCBI (https://www.ncbi.nlm.nih.gov/genome/). Metabolomics data (concentrations of aromatic amino acid metabolites) from SKOT and CIG cohorts are available in Supplementary Data 1e and 2d. Source data are provided with this paper.

## Code availability

No custom code was used in the analyses. R scripts are available on request.

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

## Acknowledgements

We thank the children and families participating in the SKOT I study, which was supported by the Danish Directorate for Food, Fisheries and Agribusiness (grant no. 3304-FSE-06-0503), as well as the children and parents participating in the CIG cohort. Furthermore, we thank Aarstiderne A/S for providing a small gift for the CIG participants. We thank M. Danner Dalgaard at the Technical University of Denmark in-house facility (DTU Multi-Assay Core, DMAC) for performing the 16S rRNA gene sequencing, MS-Omics for performing the lactate analyses, Glycom A/S for kindly donating the HMOs and A. Schnipper for her efforts in supporting the work. We also thank S. Fukiya and A. Yokota (Research Faculty of Agriculture, Hokkaido University) for providing *Bifidobacterium* gene manipulation tools and for technical suggestions, Y. Sugiyama (Graduate School of Agriculture, Kyoto University) for helpful discussions on HPLC analysis, S. Maeda (Faculty of Bioresources and Environmental Sciences, Ishikawa Prefectural University) for technical support on enzyme purification, technician L. Buus Rosholm (Department of Biotechnology and Biomedicine, Technical University of Denmark) for performing the human monocyte experiments and technician B. Møller Plesning (National Food Institute, Technical University of Denmark) for running the AhR and HCAR-3 assays. We thank laboratory animal caretakers M. Danielsen, E. Erna Navntoft and K. Rene Worm (National Food institute, Technical University of Denmark). This work was supported by Augustinus Fonden (grant no. 17-2003 to H.M.R.); Hørslev Fonden (grant no. 203866 to H.M.R.); Beckett Fonden (grant no. 17-2-0551 to H.M.R.); Aase og Ejnar Danielsens Fond (grant no. 10-002019 to H.M.R.); the Innovation Fund Denmark (grant no. 11-116163/0603-00487B; Center for Gut, Grain and Greens to T.R.L.); JSPS-KAKENHI (18K14379 to M.S., 19K22277 to T.K.); JSPS Overseas Research Fellowships (201860637 to M.S.) and the Institute for Fermentation, Osaka (to M.S. and T.K.); 'Diet-induced arrangement of the gut microbiome for improvement of cardiometabolic health' (DINAMIC) under the Joint Programming Initiative, 'A Healthy Diet for a Healthy Life' (JPI-HDHL), supported by the Innovation Fund Denmark (grant no. 5195-00001B to L.O.D.); and 'Biomarkers for infant fat mass development and nutrition' under the ERA-HDHL joint transnational programme 'Biomarkers for nutrition and health', supported by the Innovation Fund Denmark (grant no. 4203-00005B to S.B.).

## Author contributions

H.M.R. and M.F.L. conceived and designed the study. M.F.L. prepared the samples for sequencing/qPCR and analysed the sequencing/qPCR data. H.M.R. prepared the samples for faecal/caecal and in vitro metabolome analyses and performed together with H.L.F. the targeted and untargeted metabolomics experiments. C.T.P. and L.O.D. performed the urine metabolomics. M.F.L. and M.S. performed the in vitro growth experiments, cloning and recombinant expression and mutant construction experiments. M.F.L. and H.M.R. performed the animal experiment with monocolonized mice. M.S. and T.K. performed enzyme kinetics. K.F.M. and C.M. designed the SKOT I study and M.V.L. managed the data. H.M.R. and M.F.L. designed the CIG cohort, recruited the study participants and managed the data. A.M.V. performed the AhR and HCA₃ reporter assays. J.M.M. performed bioinformatics analysis of strain variation and MAGs. U.M. and S.B. performed ex vivo immune experiments. N.B., D.A. and A.R. contributed to the interpretation of the results. S.B., W.A., T.K., M.I.B. and T.R.L. contributed with expert supervision. H.M.R. and M.F.L. led the work, undertook the integrative data analyses and drafted the manuscript. All authors contributed to and approved the final manuscript.

## Competing interests

The authors declare no competing interests.

## Additional information

**Extended data** is available for this paper at https://doi.org/10.1038/s41564-021-00970-4.

**Correspondence and requests for materials** should be addressed to Tine R. Licht or Henrik M. Roager.

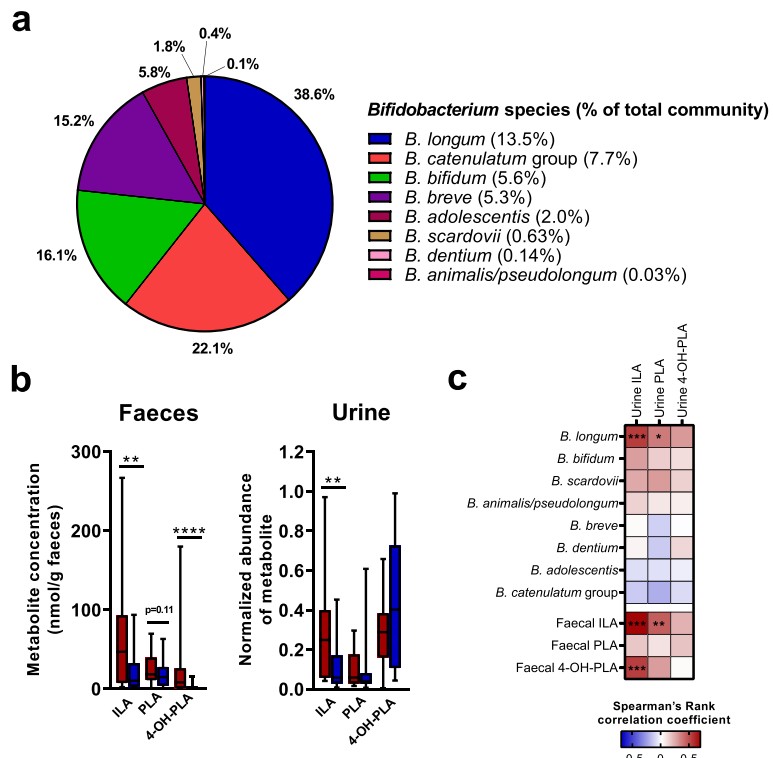

**Extended Data Fig. 1 |** *Bifidobacterium* **species composition and aromatic lactic acids in partially breastfed and weaned infants aged 9 months from the SKOT cohort. a**, Pie chart of the average *Bifidobacterium* species composition given as percent of total *Bifidobacterium* abundance (n = 59 infants). Legend includes the average percent of each species compared to the total faecal microbiota community (See also Supplementary Data 1g). **b**, Left panel: Box and whiskers plot (line: median, box: IQR, whiskers: min-max) of faecal abundance of indolelactic acid (ILA), phenyllactic acid (PLA) and 4-hydroxyphenyllactic acid (4-OH-PLA) in partially breastfed (n = 24, red) and weaned (n = 35, blue) infants. Right panel: Box and whiskers plot (line: median, box: IQR, whiskers: min-max) of urine abundance of ILA, PLA and 4-OH-PLA in partially breastfed (n = 19, red) and weaned (n = 30, blue) infants. Statistical significance was evaluated by two-sided Mann–Whitney U-test. **c**, Heatmap of Spearman's Rank correlation coefficients (two-sided tests) between the relative abundances of ILA, PLA and 4-OH-PLA measured in urine and the faecal relative abundance of *Bifidobacterium* species or faecal concentrations of ILA, PLA and 4-OH-PLA of the same infants (n = 49 infants). For all panels asterisks indicate statistical significance: *p < 0.05, **p < 0.01, ***p < 0.001 and ****p < 0.0001.

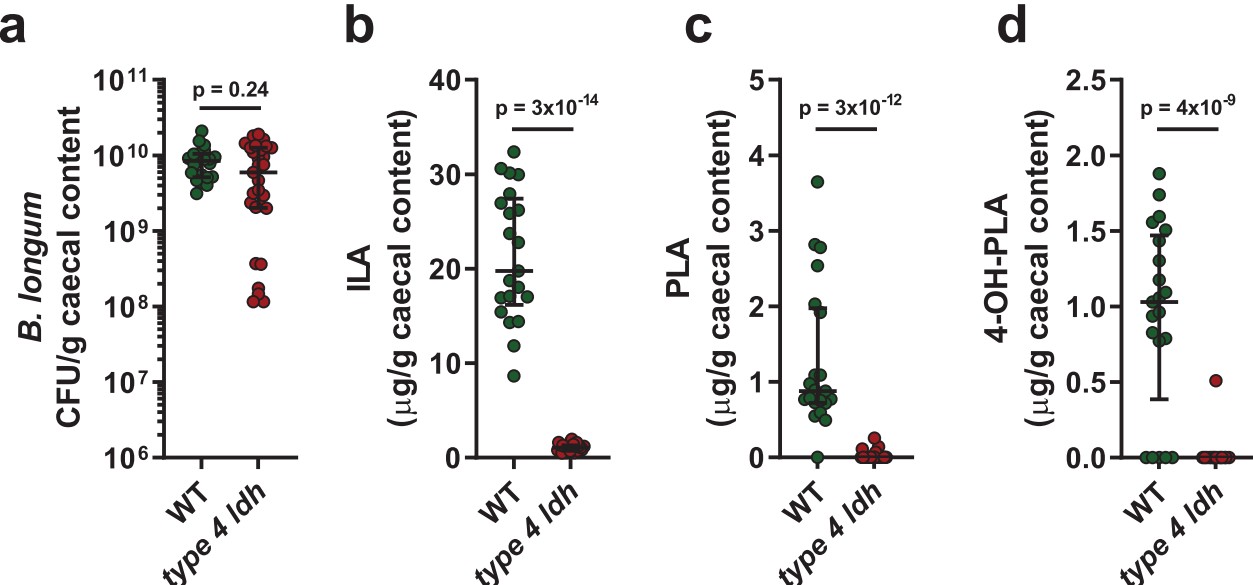

**Extended Data Fig. 2 |** *In vivo* **production of aromatic lactic acids in previously germ-free mice colonized with either** *B. longum* **105-A WT or type 4** *ldh* **mutant. a**, CFU counts of *B. longum* 105-A WT or type 4 *ldh* mutant from caecal content of mice monocolonized with either the WT (n = 21) or type 4 *ldh* mutant strain (n = 29). Bars and error bars indicate median ± IQR. **b–d**, caecal concentrations of the aromatic lactic acids (indolelactic acid (ILA), phenyllactic acid (PLA) and 4-hydroxyphenyllactic acid (4-OH-PLA)) in mice monocolonized with either the WT (n = 21) or type 4 *ldh* mutant strain (n = 29). Line and error bars indicate median ± IQR. For all panels statistical significance was evaluated by two-sided Mann–Whitney U-tests.

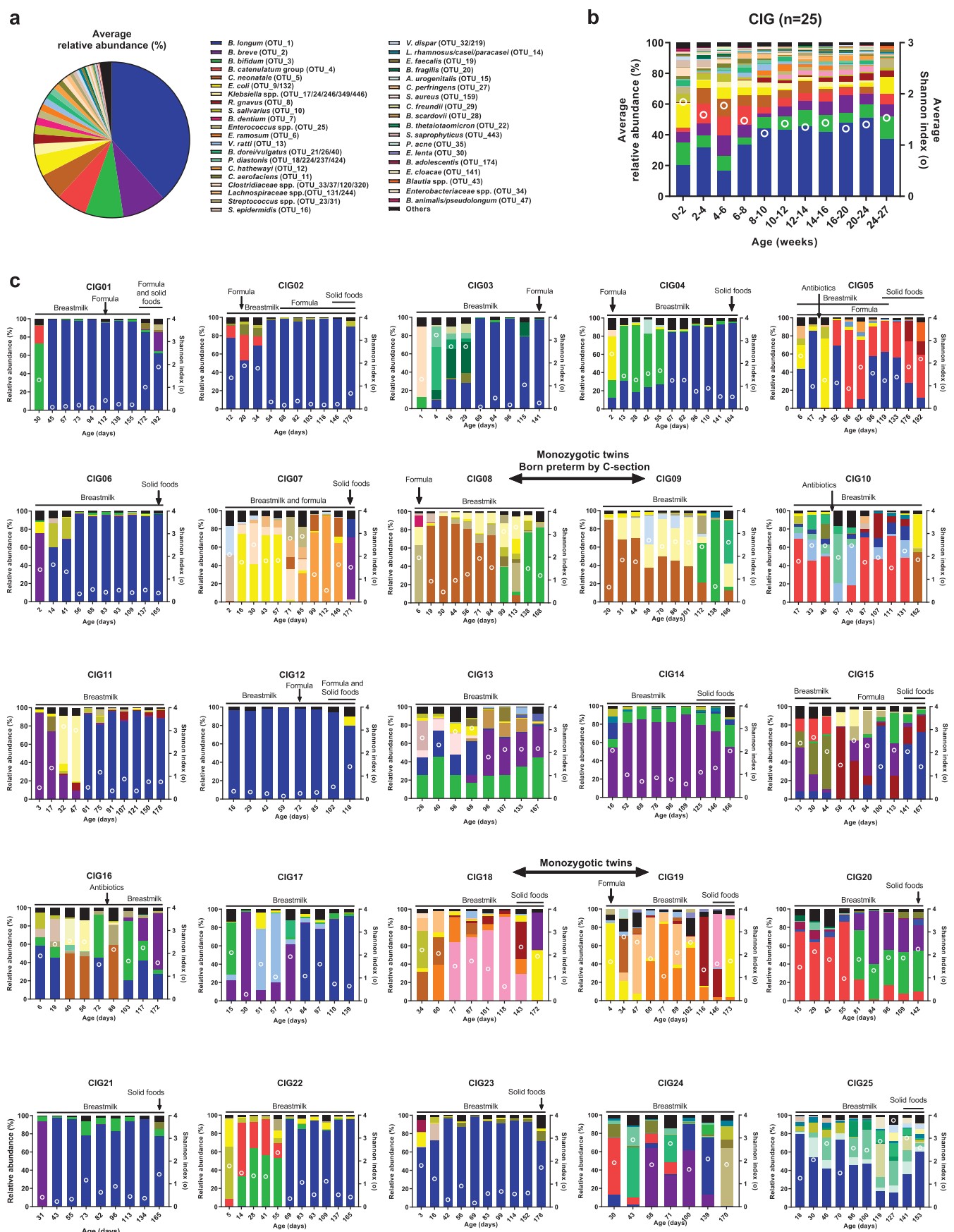

**Extended Data Fig. 3 | See next page for caption.**

**Extended Data Fig. 3 | Faecal microbiota composition of the CIG infants. a**, Average relative abundance of the dominant faecal microbial taxa (average relative abundance >0.1%) across all samples (comprising 97.5% of all microbial taxa detected). **b**, Temporal development of the average gut microbiota composition and Shannon diversity index (marked with circles) across all individuals. **c**, Intra-individual temporal development of gut microbiota composition and Shannon diversity index (marked with circles). Dietary patterns and consumption of antibiotics are indicated for each individual. If nothing else is indicated, infants were singletons, vaginally born at term.

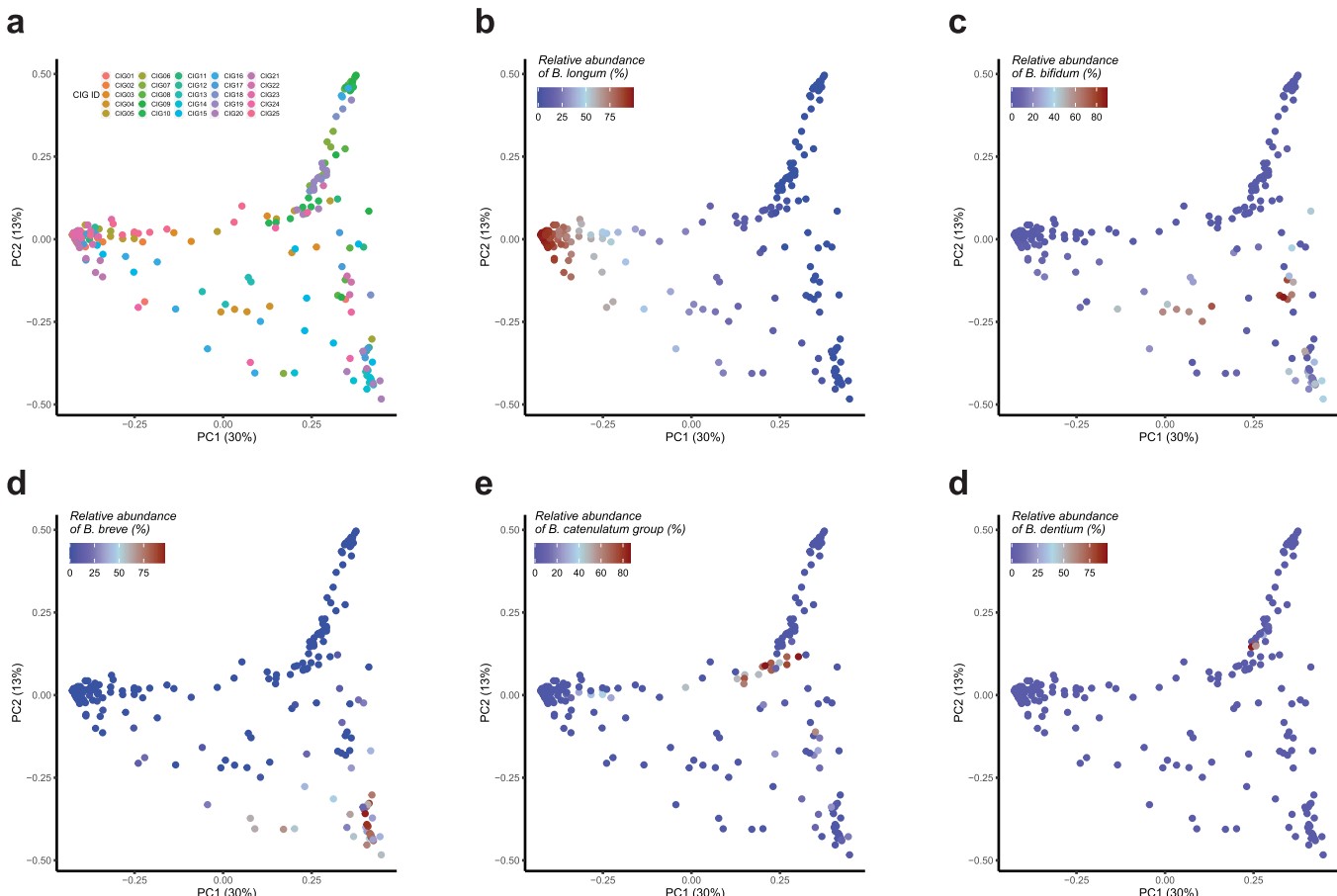

**Extended Data Fig. 4 | Beta diversity in CIG cohort. a-f,** Principal coordinates analysis (PCoA) plots of Bray–Curtis dissimilarities, based on all OTUs detected in CIG faecal samples (n = 234), coloured according to **a**, subject and **b-f**, relative abundances of *Bifidobacterium longum, Bifidobacterium bifidum, Bifidobacterium breve, Bifidobacterium catenulatum* group or *Bifidobacterium dentium*, respectively.

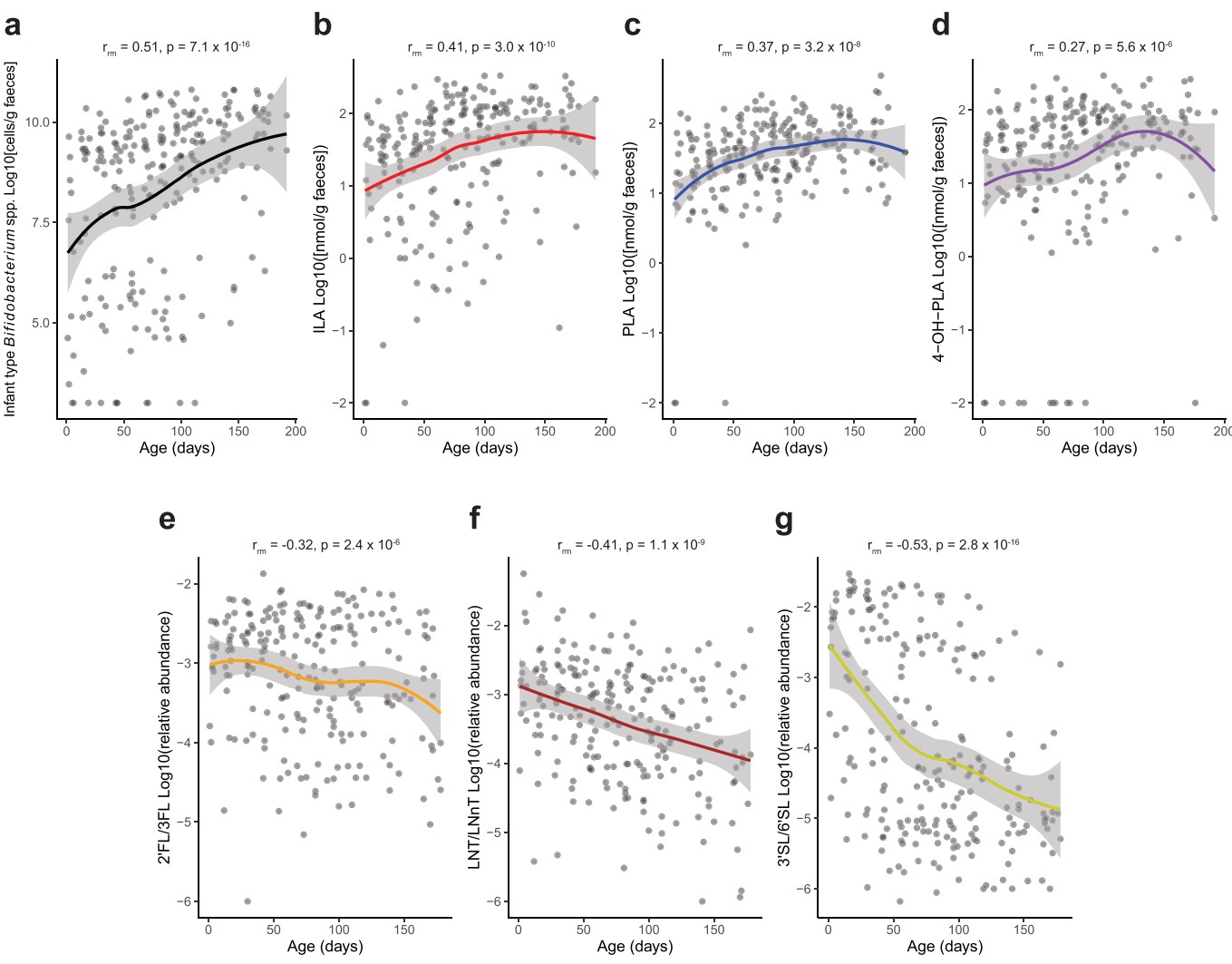

**Extended Data Fig. 5 | Temporal development in abundance of infant-type *Bifidobacterium* species, aromatic lactic acids, and human milk oligosaccharides in faeces from infants in the CIG cohort.** Scatter plots of **a**, age against absolute abundance of infant-type *Bifidobacterium* species (defined as the sum of absolute abundances of *B. longum, B. breve, B. bifidum* and *B. scardovii*) in faeces or **b–d**, age against faecal concentrations of aromatic lactic acids (ILA, PLA and 4-OH-PLA, n = 240) or **e-g**, age against relative faecal abundance of human milk oligosaccharides (2'FL/3FL, LNT/LNnT and 3'SL/6'SL, n = 228) during the first 6 months of life in the CIG cohort. A local polynomial regression (LOESS) fit is shown with coloured mean line and 95% CI shaded in grey. Statistical significance was evaluated by two-sided repeated measures correlations ($r_{rm}$ is the repeated measures correlation coefficient).

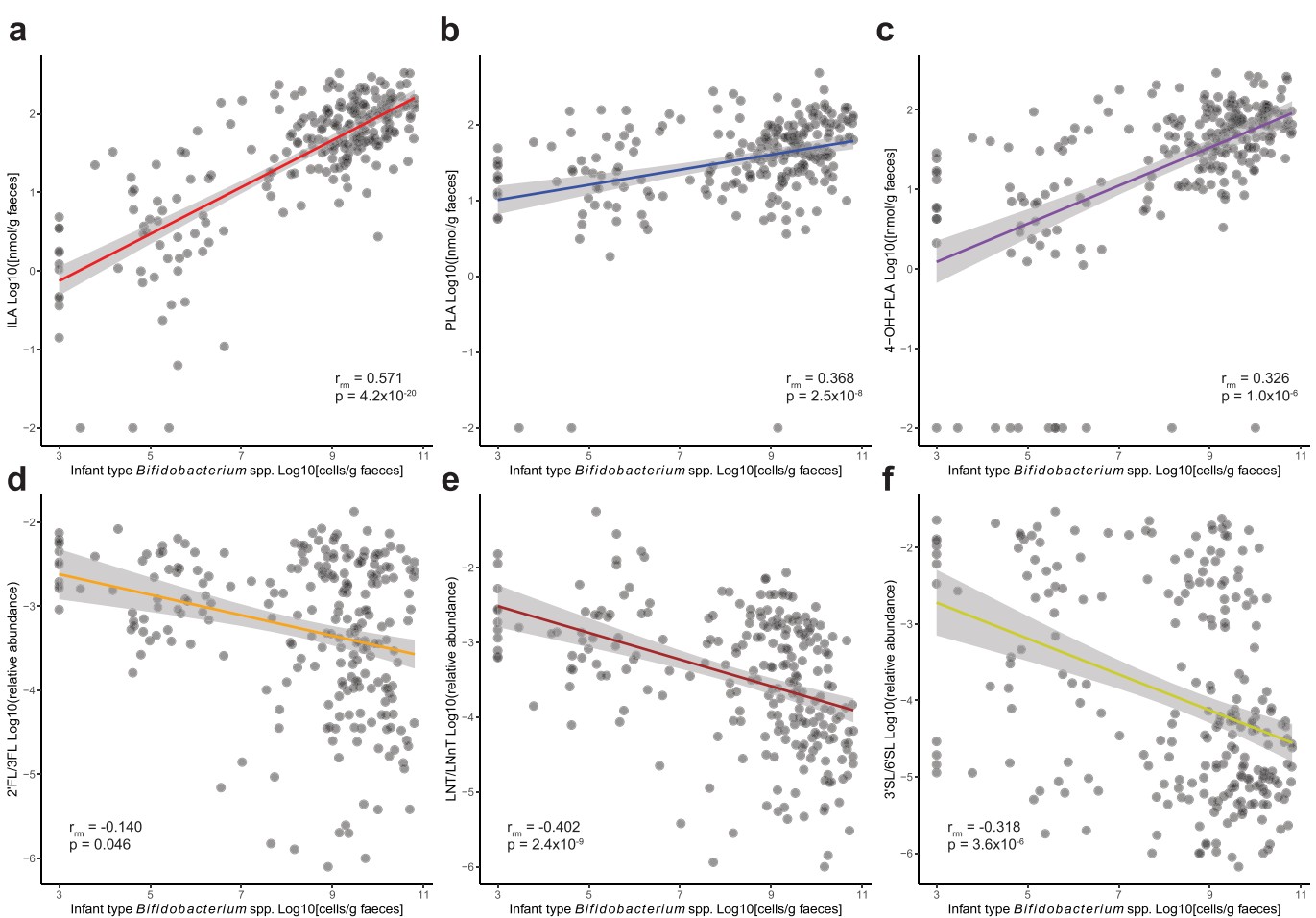

**Extended Data Fig. 6 | Associations between Infant-type *Bifidobacterium* species and aromatic lactic acids and human milk oligosaccharides in faeces from infants in the CIG cohort.** Scatter plots of the relationship between faecal absolute abundance of infant-type *Bifidobacterium* species, and **a–c**, faecal concentrations of aromatic lactic acids (ILA, PLA and 4-OH-PLA, n=240) or **d–f**, relative faecal abundance of human milk oligosaccharides (2′FL/3FL, LNT/LNnT and 3′SL/6′SL, n=228) in the CIG cohort. Statistical significance was evaluated by two-sided repeated measures correlations ($r_{rm}$ is the repeated measures correlation coefficient). Linear regression curve fits are shown with coloured mean line and 95% CI indicated in grey shading.

**a**

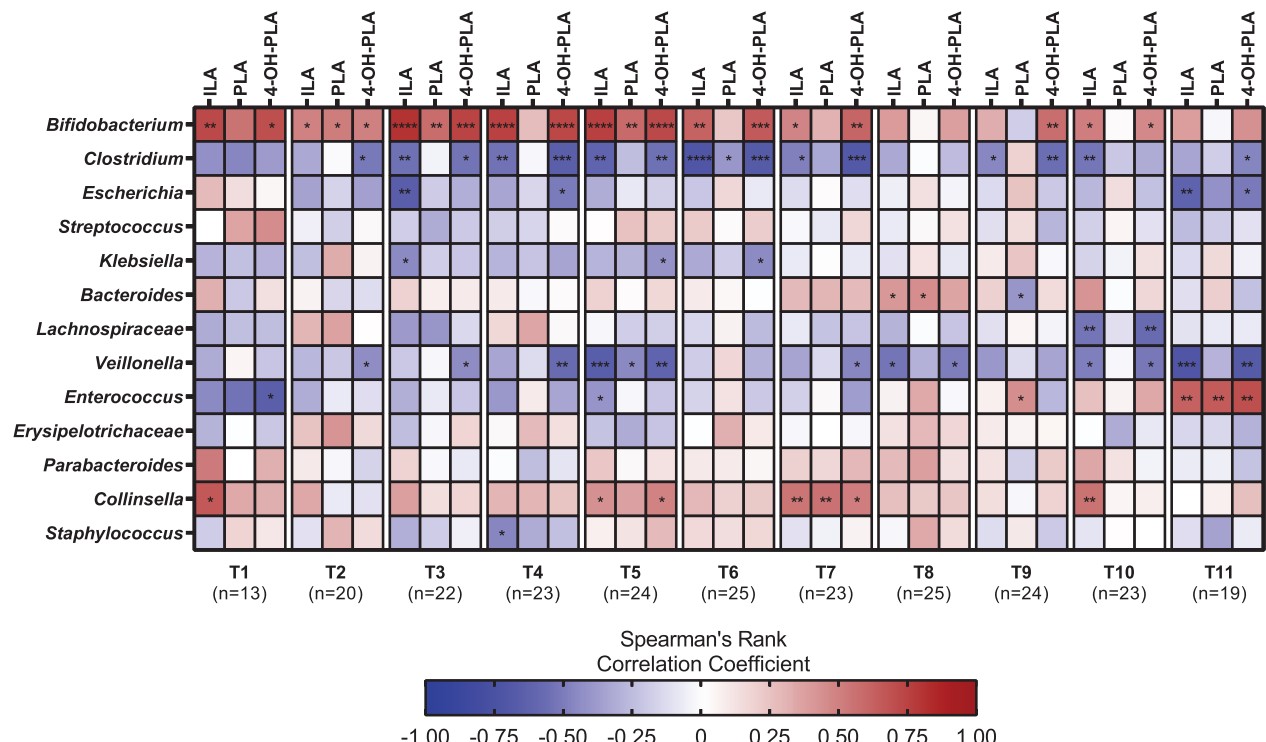

**b**

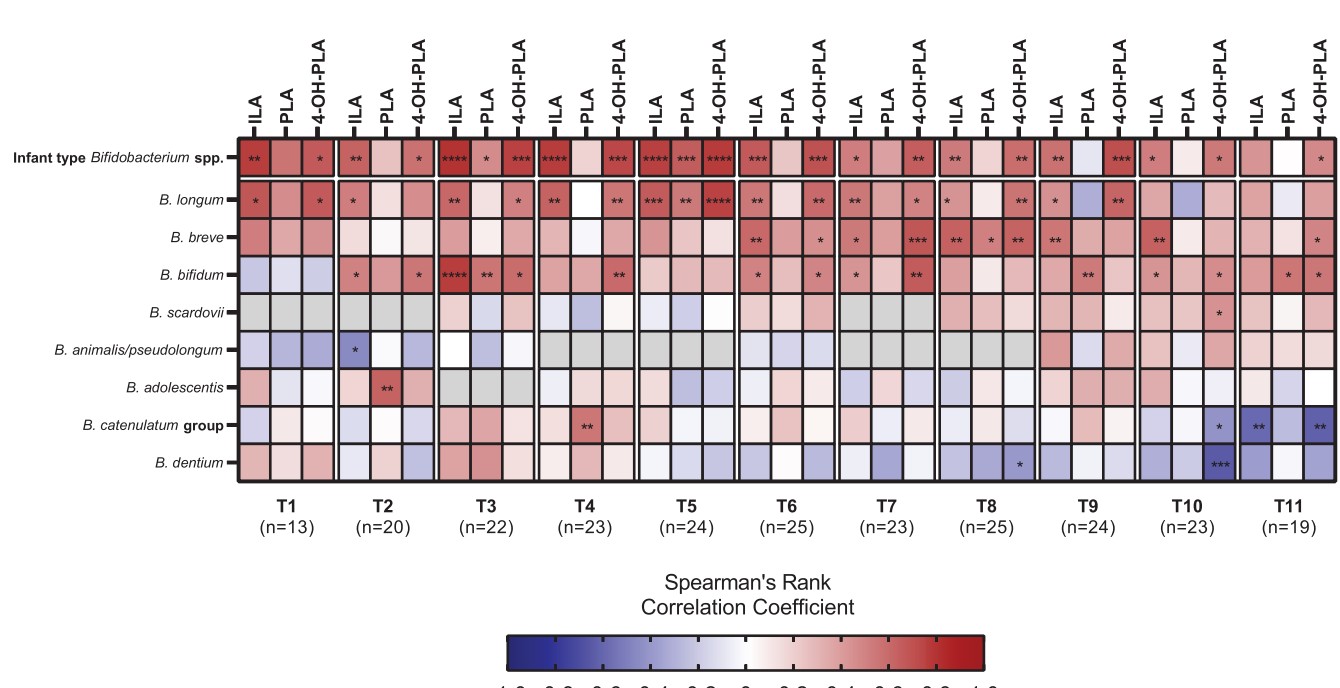

**Extended Data Fig. 7 | Correlations between relative abundance of bacterial taxa and concentrations of the aromatic lactic acids at each sampling point in the CIG cohort. a**, Spearman's rank correlations between relative abundance of faecal bacterial genera (average relative abundance > 1%) and faecal concentrations of the aromatic lactic acids (ILA, PLA and 4-OH-PLA) at each sampling point. **b**, Spearman's rank correlations between relative abundance of faecal *Bifidobacterium* species (average relative abundance > 0.1%) and faecal concentrations of the aromatic lactic acids at each sampling point. Infant-type *Bifidobacterium* species is the sum of the relative abundances of *B. longum, B. breve, B. bifidum* and *B. scardovii*. For both panels statistical significance was evaluated by uncorrected p-values (two-sided tests) indicated by asterisks with *p < 0.05, **p < 0.01, ***p < 0.001 and ****p < 0.0001.

**a**     **16S rRNA ASV analysis**     **b**     **Species/subspecies specific qPCR**

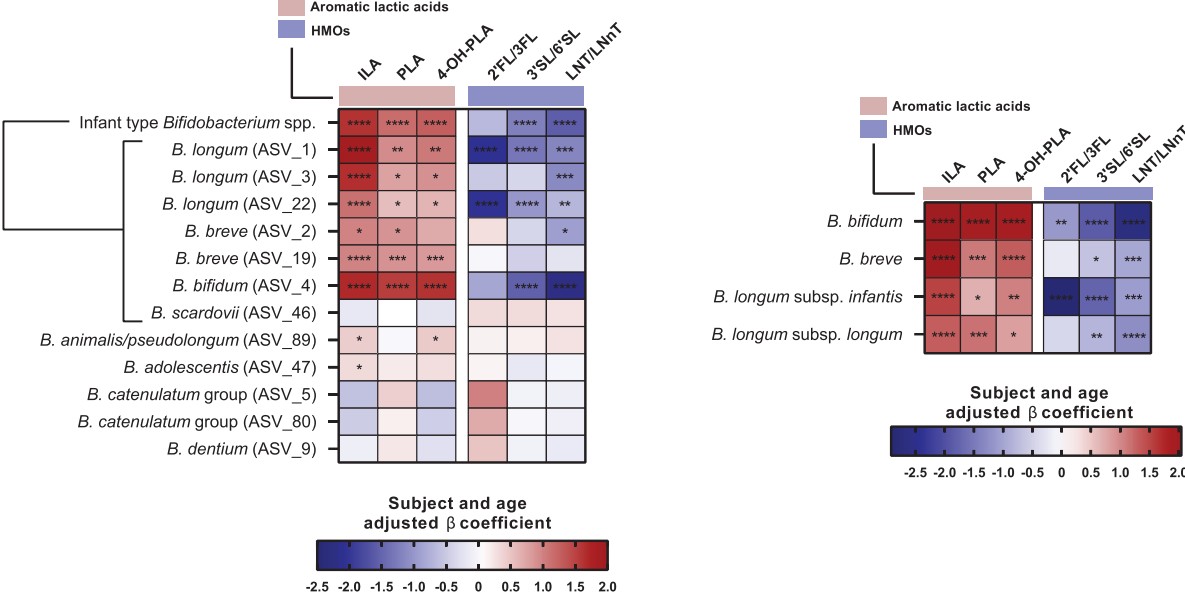

**Extended Data Fig. 8 | Validation of associations between absolute abundance of *Bifidobacterium* species and faecal abundances of the aromatic lactic acids and human milk oligosaccharides in the CIG cohort.** Heatmaps illustrating linear mixed model coefficients (adjusted for subject and age) between the absolute abundance of *Bifidobacterium* species estimated by **a**, 16 S rRNA amplicon sequence variant analysis or **b**, species/subspecies-specific qPCR and faecal concentrations of aromatic lactic acids (ILA, PLA and 4-OH-PLA, n = 240) or relative faecal abundances of human milk oligosaccharides (2′FL/3FL, 3′SL/6′SL and LNT/LNnT, n = 228) in the CIG cohort. For both panels statistical significance was evaluated by false discovery rate-corrected p-values (two-sided tests) indicated by asterisks with *p < 0.05, **p < 0.01, ***p < 0.001 and ****p < 0.0001.

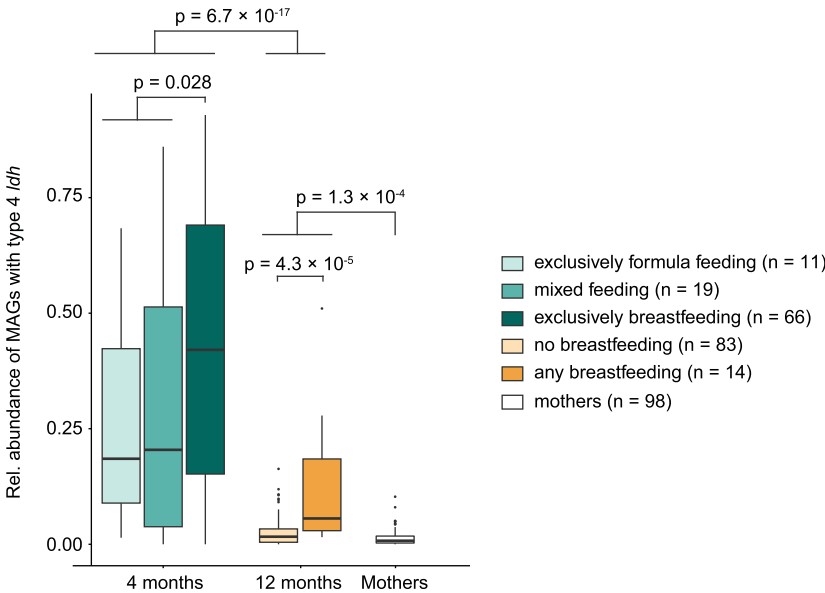

**Extended Data Fig. 9 | Relative abundance of high quality metagenome-assembled genomes (MAGs) with _aldh_ (type 4 _ldh_) in 4 months and 12 months old infants and mothers.** Metagenome data were retrieved from Bäckhed et al [reference[4]]. Line, boxes and whiskers indicate median, IQR and ±1.5*IQR and statistical significance was evaluated by two-sided Mann–Whitney U-tests.

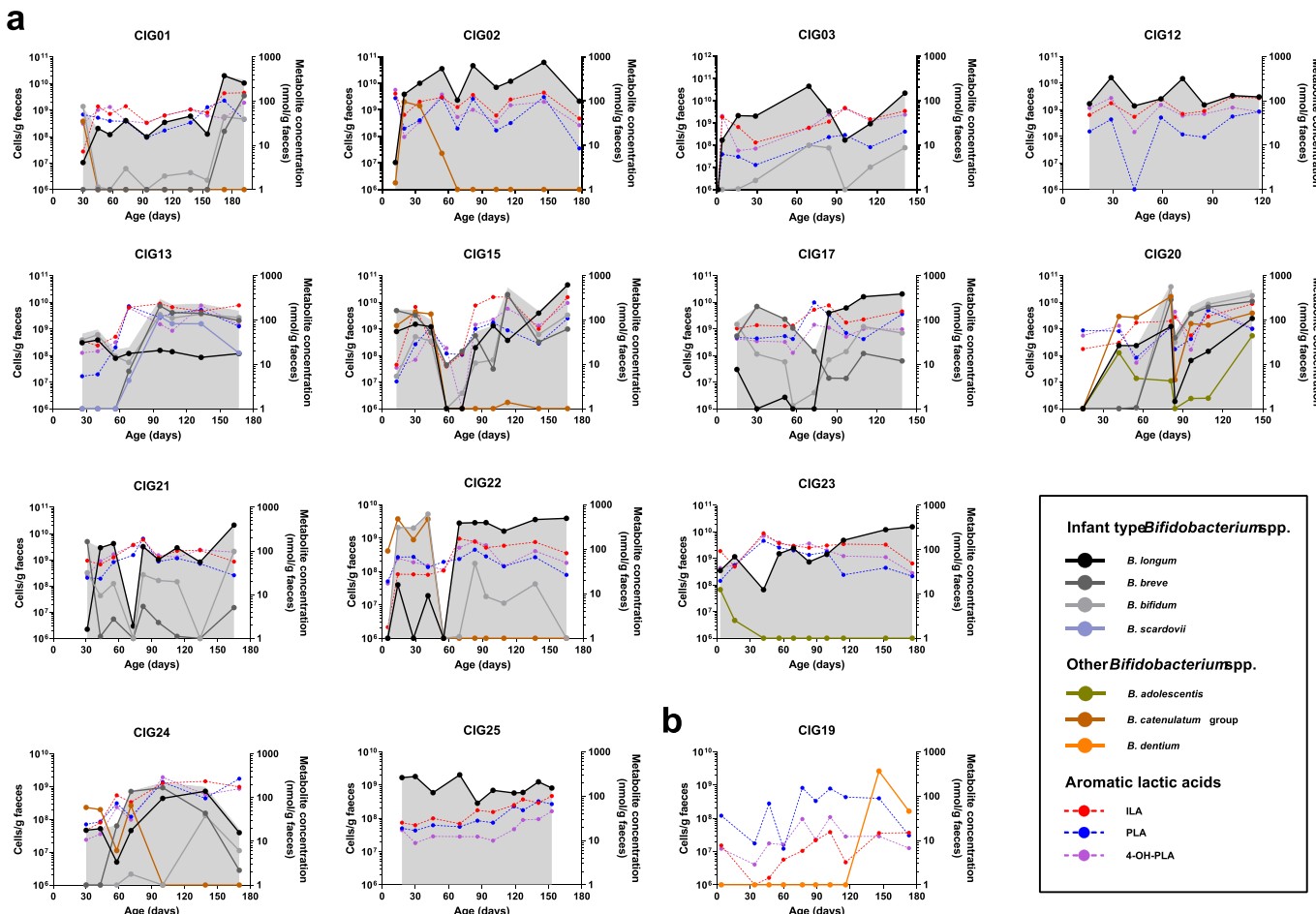

**Extended Data Fig. 10 | Absolute abundances of *Bifidobacterium* species and concentrations of aromatic lactic acids in faeces of infants in the CIG cohort. a,b**, Absolute abundance of *Bifidobacterium* species (average relative abundance >1% of total community) and concentrations of indolelactic acid (ILA), phenyllactic acid (PLA) and 4-hydroxyphenyllactic acid (4-OH-PLA) in faeces of individuals from the Copenhagen Infant Gut (CIG) cohort. Values of bacterial counts below $10^6$ cells/g faeces and metabolite concentrations below 1 nmol/g faeces are not shown. Infant-type *Bifidobacterium* species is the sum of the absolute abundances of *B. longum, B. breve, B. bifidum* and *B. scardovii* and is indicated with grey background shading. **a**, Infants early colonized with infant-type *Bifidobacterium* species (colonized within first month reaching average relative abundance >40% during first 6 months), **b**, Infants with late colonization of infant-type *Bifidobacterium* species (not detectable or on average <0.5% of total community within the first 3 months of life).

Tine Rask Licht

# Reporting Summary

Nature Research wishes to improve the reproducibility of the work that we publish. This form provides structure for consistency and transparency in reporting. For further information on Nature Research policies, see Authors & Referees and the Editorial Policy Checklist.

## Statistics

For all statistical analyses, confirm that the following items are present in the figure legend, table legend, main text, or Methods section.

| n/a | Confirmed | |
|---|---|---|
| ☐ | ☒ | The exact sample size (*n*) for each experimental group/condition, given as a discrete number and unit of measurement |
| ☐ | ☒ | A statement on whether measurements were taken from distinct samples or whether the same sample was measured repeatedly |
| ☐ | ☒ | The statistical test(s) used AND whether they are one- or two-sided<br>*Only common tests should be described solely by name; describe more complex techniques in the Methods section.* |
| ☐ | ☒ | A description of all covariates tested |
| ☐ | ☒ | A description of any assumptions or corrections, such as tests of normality and adjustment for multiple comparisons |
| ☐ | ☒ | A full description of the statistical parameters including central tendency (e.g. means) or other basic estimates (e.g. regression coefficient) AND variation (e.g. standard deviation) or associated estimates of uncertainty (e.g. confidence intervals) |
| ☐ | ☒ | For null hypothesis testing, the test statistic (e.g. *F*, *t*, *r*) with confidence intervals, effect sizes, degrees of freedom and *P* value noted<br>*Give P values as exact values whenever suitable.* |
| ☒ | ☐ | For Bayesian analysis, information on the choice of priors and Markov chain Monte Carlo settings |
| ☒ | ☐ | For hierarchical and complex designs, identification of the appropriate level for tests and full reporting of outcomes |
| ☐ | ☒ | Estimates of effect sizes (e.g. Cohen's *d*, Pearson's *r*), indicating how they were calculated |

*Our web collection on statistics for biologists contains articles on many of the points above.*

## Software and code

Policy information about availability of computer code

| Data collection | No software was used |
|---|---|
| Data analysis | Quantification of aromatic amino acids in faeces and in vitro fermentation samples was performed using QuantAnalysis version 2.2 (Bruker Daltonics, Bremen, Germany). The raw urine metabolome data were converted to netCDF format using DataBridge Software version 3.5 (Waters, Manchester, UK) and imported into MZmine version 2.28. The raw untargeted faecal metabolome data were converted to mzXML files using Bruker Compass DataAnalysis 4.2 software (Bruker Daltonics). Gut microbiota data were processed by CLC Genomic Workbench (v8.5. CLCbio, Qiagen, Aarhus, DK), QIIME v1.967, and DADA2 pipeline (v.1.14). qPCR data were processed by the LightCycler® 480 Software v.1.5. Alignment of lactate dehydrogenase genes was constructed in CLC Main Workbench (v7.6.3, CLCbio, Qiagen, Aarhus, DK) and the phylogenic tree was visualized by use of the FigTree software v1.4.3 (http://tree.bio.ed.ac.uk/software/figtree/). The kinetic parameters (kcat, K0.5, and Hill coefficient nH) were calculated by curve-fitting the experimental data to the Hill equation, using GraphPad Prism v8.1. Bifidobacterium metagenome assembled genomes (MAGs) were identified by IGGsearch and IGGdb (v.1.0.087). Statistical analyses were performed using QIIME (v1.967), R (v3.189), GraphPad Prism (v8.1), and Matlab R2014b or 2015b (Mathworks, Inc.). The follow R packages were used: ggplot2 (v3.3.3), rmcorr (v0.4.3), maaslin2 (v1.0.0), gplots (v3.1.1) |

For manuscripts utilizing custom algorithms or software that are central to the research but not yet described in published literature, software must be made available to editors/reviewers. We strongly encourage code deposition in a community repository (e.g. GitHub). See the Nature Research guidelines for submitting code & software for further information.

## Data

Policy information about availability of data

All manuscripts must include a data availability statement. This statement should provide the following information, where applicable:

- Accession codes, unique identifiers, or web links for publicly available datasets
- A list of figures that have associated raw data
- A description of any restrictions on data availability

All 16S rRNA gene amplicon sequencing data were deposited in the Sequence Read Archive (SRA) under the BioProjects PRJNA273694 (SKOT) and PRJNA554596 (CIG). The following databases were used: GreenGenes 16S rRNA database (https://greengenes.secondgenome.com/), RDP database (https://rdp.cme.msu.edu/), 16S rRNA gene sequence database at NCBI (https://ncbiinsights.ncbi.nlm.nih.gov/2020/02/21/rrna-databases/), non-redundant protein sequence database at NCBI (https://www.ncbi.nlm.nih.gov/protein/), Swiss-Prot database (https://www.uniprot.org/), MBGD (Microbial Genome Database for Comparative Analysis; http://mbgd.genome.ad.jp/), and the genome database at NCBI (https://www.ncbi.nlm.nih.gov/genome/). Metabolomics data (concentrations of aromatic amino acid metabolites) from SKOT and CIG cohorts are available in Supplementary Data 1e and 2d. Source data are provided with this paper.

# Field-specific reporting

Please select the one below that is the best fit for your research. If you are not sure, read the appropriate sections before making your selection.

☒ Life sciences          ☐ Behavioural & social sciences          ☐ Ecological, evolutionary & environmental sciences

For a reference copy of the document with all sections, see nature.com/documents/nr-reporting-summary-flat.pdf

# Life sciences study design

All studies must disclose on these points even when the disclosure is negative.

| | |
|---|---|
| Sample size | No sample size was estimated. Based on the original correlations observed between Bifidobacterium and aromatic lactic acids in 59 infants (SKOT), we estimated that 25 infants (CIG) with multiple time points would be sufficient to demonstrate the dynamics between Bifidobacterium species and aromatic lactic acids. |
| Data exclusions | In the CIG cohort, 6 samples were omitted from the Principal coordinate analyses due to low read counts (<8000). For correlation analyses between Bifidobacterium species and HMO residuals in faeces, 12 samples with no reported breastfeeding were excluded. One donor was excluded from the 200 μM indolelactate (ILA) stimulation of monocytes, since something went wrong during the stimulation. |
| Replication | All experiments were performed with full factorial (biological and technical) replication. |
| Randomization | No randomisation was performed as the study was observational. |
| Blinding | No blinding was performed as the study was observational. |

# Reporting for specific materials, systems and methods

We require information from authors about some types of materials, experimental systems and methods used in many studies. Here, indicate whether each material, system or method listed is relevant to your study. If you are not sure if a list item applies to your research, read the appropriate section before selecting a response.

### Materials & experimental systems

| n/a | Involved in the study |
|---|---|
| ☐ | ☒ Antibodies |
| ☐ | ☒ Eukaryotic cell lines |
| ☒ | ☐ Palaeontology |
| ☐ | ☒ Animals and other organisms |
| ☐ | ☒ Human research participants |
| ☒ | ☐ Clinical data |

### Methods

| n/a | Involved in the study |
|---|---|
| ☒ | ☐ ChIP-seq |
| ☒ | ☐ Flow cytometry |
| ☒ | ☐ MRI-based neuroimaging |

## Antibodies

| | |
|---|---|
| Antibodies used | CD14-PE-Cy7 (eBiosciences, catalog number: 25-0149, clone name: 61D3) and CD16-FITC (Biolegend, catalog number: 302006, clone name: 3G8) |
| Validation | The 61D3 monoclonal antibody reacts with human CD14, a 53-55 kDa GPI-linked glycoprotein. CD14 is expressed on monocytes, interfollicular macrophages and some dendritic cells (manufacturer's website: https://www.thermofisher.com/antibody/product/CD14-Antibody-clone-61D3-Monoclonal/25-0149-42) |

CD16 is known as low affinity IgG receptor III (FcγRIII). It is expressed as two distinct forms (CD16a and CD16b). CD16a (FcγRIIIA) is a 50-65 kD polypeptide-anchored transmembrane protein. It is expressed on the surface of NK cells, activated monocytes, macrophages, and placental trophoblasts in humans (manufacturer's website: https://www.biolegend.com/en-us/products/fitc-anti-human-cd16-antibody-567)

# Eukaryotic cell lines

Policy information about cell lines

| | |
|---|---|
| Cell line source(s) | Rat AhR Reporter Cells (H4IIE) was a personal gift from Dr. Michael S. Denison, Department of Environmental Toxicology, Meyer Hall, University of California, Davis California 95616, United States) Human AhR Reporter Cells from Indigo Biosciences (PA, USA, catalogue number: IB06001) cAMP Hunter™ eXpress GPR109B CHO-K1 GPCR Assay from DiscoverX (Fremont, CA, USA, catalogue number: 95-0141E2CP2M) |
| Authentication | None of the cells were tested for authentication. |
| Mycoplasma contamination | H4IIE cells were tested negative for mycoplasma contamination using the MycoAlert® Mycoplasma Detection Kit. The human Aryl Hydrocarbon Receptor (AhR) Reporter Assay System and the cAMP Hunter™ eXpress GPCR Assay were commercially available assays obtained from Indigo Biosciences and DiscoverX, respectively. We did not test these cells for mycoplasma contamination, but relied on the internal quality control of the manufacturers. |
| Commonly misidentified lines (See ICLAC register) | No commonly misidentified cell lines were used in the study. |

# Animals and other organisms

Policy information about studies involving animals; ARRIVE guidelines recommended for reporting animal research

| | |
|---|---|
| Laboratory animals | Germ-free Swiss Webster mice (Tac:SW, originally obtained from Taconic Biosciences, NY, USA) were bred and housed at the National Food Institute, Technical University of Denmark. In two separate experiments, pregnant GF mice were randomised to be colonised with either B. longum 105-A wild-type (n=4) or aldh (type 4 ldh) mutant (n=5) by a single gavage one week prior to giving birth. The mono-colonised offspring (wildtype=21, 12 males and 9 females; aldh=29, 18 males and 11 females) were euthanised at 4 weeks of age. |
| Wild animals | No wild animals were used in the study. |
| Field-collected samples | No field-collected samples were used in the study. |
| Ethics oversight | All mouse experiments were approved by the Danish Animal Experiments Inspectorate (License number 2015-15-0201-00553) and carried out in accordance with existing Danish guidelines for experimental animal welfare. |

Note that full information on the approval of the study protocol must also be provided in the manuscript.

# Human research participants

Policy information about studies involving human research participants

| | |
|---|---|
| Population characteristics | The discovery cohort consisted of a random subset of 59 healthy infants (30 male, 29 female) participating in the SKOT I study. Inclusion criteria were single birth and full term delivery, absence of chronic illness and age of 9 months ± 2 weeks at inclusion. The validation cohort, CIG, consisted of 25 healthy infants (12 male, 13 female), vaginally born (23/25) and full-term (23/25) delivered. |
| Recruitment | Infants in SKOT I were originally recruited by postal invitations to randomly selected parents of infants on the basis of extractions from the Danish National Civil Registration System. We used a random subset of this cohort for discovery. For validating the results, infants in CIG were recruited through social media and limited to the Copenhagen region. These recruitments might have biased the recruitment towards an over-representation of breastfed, healthy infants from families with a high socioeconomic status. According to the GDPR rules, no information were available regarding covariate-relevant population characteristics of the healthy anonymous donor buffy coat material provided by the Blood bank at Rigshospitalet, Copenhagen. The only criteria was 'no intake of pain killers' on the day of blood draw. |
| Ethics oversight | The SKOT study protocol was approved by the Committees on Biomedical Research Ethics for the Capital Region of Denmark (H-KF-2007-0003) and The Data Protection Agency (2002-54-0938, 2007-54-026) approved the study. The Committees on Biomedical Research Ethics for the Capital Region of Denmark confirmed that the CIG study was not notifiable according to the Act on Research Ethics Review of Health Research Projects (§ 1, subsection 4), as the study only concerned the faecal microbial composition and activity and not the health of the children. Informed consent was obtained from all parents of infants participating in the study. In addition, parents of CIG twins gave informed consent to publish data from the twins although the parents themselves would be able to identify their children using indirect identifiers. The Data Protection Agency (18/02459) approved the study. Use of the buffy coat material from healthy anonymous donors was approved by the Blood bank at Rigshospitalet, Copenhagen, under the jurisdiction of Region H. |

Note that full information on the approval of the study protocol must also be provided in the manuscript.

