## [Peer Review File · Nature Microbiology]

Peer Review Information

Journal: Nature Microbiology

Manuscript Title: **Bifidobacterium species associated with breastfeeding produce aromatic lactic acids in the infant gut**

Corresponding author name(s): Henrik Roager

Reviewer Comments & Decisions:

EA, duplicate for each version as needed, and then delete this instruction.

Decision Letter, initial version:

Dear Henrik,

Thank you for your patience while your manuscript "Breastmilk-promoted bifidobacteria produce aromatic amino acids in the infant gut" was under peer-review at Nature Microbiology. It has now been seen by 3 referees, whose expertise and comments you will find at the end of this email. Although they find your work of some potential interest, they have raised a number of concerns that will need to be addressed before we can consider publication of the work in Nature Microbiology.

In particular, referees #2 and #3 note that many statements and conclusions should be toned down given that the current data provides correlations rather than causal mechanistic explanations. As a result, please tone down several statements and conclusions noted by these referees, or perform additional experiments to provide mechanistic links to show that Bifidobacteria produce aromatic lactic acids in the gut. Referee #3 also has concerns with the methods used for the microbiome analyses. This referee provides several suggestions to update these analyses and provide more statistical support, especially for the longitudinal analyses. Referee #3 also questions the use of 16S rRNA sequencing to look at subspecies and suggests using metagenomics to provide this level of resolution, and questions discrepancies between datasets. More detailed statistical support for your analyses is also required (ref #1 and ref #3), as well as validation for your mass spectrometry (referee #2).

If any of these requests are not doable, please do get in contact and let me know.

Should further experimental data allow you to address these criticisms, we would be happy to look at a revised manuscript.

Please include a data availability statement as a separate section after Methods but before references, under the heading "Data Availability". This section should inform readers about the availability of the data used to support the conclusions of your study. This information includes accession codes to public repositories (data banks for protein, DNA or RNA sequences, microarray, proteomics data etc...), references to source data published alongside the paper, unique identifiers such as URLs to data repository entries, or data set DOIs, and any other statement about data availability. At a minimum, you should include the following statement: "The data that support the findings of this study are available from the corresponding author upon request", mentioning any restrictions on availability. If DOIs are provided, we also strongly encourage including these in the Reference list (authors, title, publisher (repository name), identifier, year). For more guidance on how to write this section please see: <http://www.nature.com/authors/policies/data/data-availability-statements-data-citations.pdf>

* If you have not done so already we suggest that you begin to revise your manuscript so that it conforms to our Article format instructions at <http://www.nature.com/nmicrobiol/info/final-submission>. Refer also to any guidelines provided in this letter.

{REDACTED}

Note: This url links to your confidential homepage and associated information about manuscripts you may have submitted or be reviewing for us. If you wish to forward this e-mail to co-authors, please delete this link to your homepage first.

Nature Microbiology is committed to improving transparency in authorship. As part of our efforts in this direction, we are now requesting that all authors identified as 'corresponding author' on published papers create and link their Open Researcher and Contributor Identifier (ORCID) with their account on the Manuscript Tracking System (MTS), prior to acceptance. This applies to primary research papers only. ORCID helps the scientific community achieve unambiguous attribution of all scholarly contributions. You can create and link your ORCID from the home page of the MTS by clicking on 'Modify my Springer Nature account'. For more information please visit www.springernature.com/orcid.

If you wish to submit a suitably revised manuscript we would hope to receive it within 6 months. If you cannot send it within this time, please let us know. We will be happy to consider your revision, even if a similar study has been accepted for publication at Nature Microbiology or published elsewhere (up to a maximum of 6 months).

{REDACTED}

Reviewer Expertise:

Referee #1: infant nutrition, microbiome

Referee #2: metabolomics, microbial metabolism, microbiome

Referee #3: mother-infant microbiome, probiotics

Reviewer Comments:

Reviewer #1 (Remarks to the Author):

The authors detail a mechanistic dissection of the molecular pathways behind the production of aromatic amino acids, their pathway of recognition by host cells, and the impacts on immune cells in infants. The findings represent a significant advancement in both the study of host-associated microbes and human infant gut symbionts. Through a series of exhaustive and confirmatory experiments, they describe and demonstrate a compelling story that confirms the production, recognition, and outcomes of aromatic amino acid production in the infant gut and demonstrate these observations via clinical samples obtained from infant stools.

The paper itself is very well written, logically organized, and I have very few recommendations for improving the manuscript further, as outlined below. I really enjoyed reading the paper, and the thoroughness and attention to detail is impressive. There are obvious implications of this work to intervention studies examining infant health and immune development, but I respect that this may be beyond the scope of the current manuscript. I look forward to seeing future extensions of this work from the authors.

I would like to make a few humble recommendations to the authors:

1. Line 238-243: It would be helpful to indicate the standard deviations, and ideally, the number/percent of infants which met the 'high' Bifidobacterium criterion used for the "early colonized infants" (>40% Bifidobacterium) elsewhere in the paper. I would also advise against "As expected.." to describe Bifidobacterium colonization among infants because this is not generalizable. While C-section delivery was not apparently an exclusion criterion for participation, the number of C-section delivered infants was very low in both the SKOT and CIG cohorts. C-section delivery has been shown to inhibit maternal Bifidobacterium transfer (e.g. Shao et al 2019), so these findings might be expected for *this* population, but are not necessarily generalizable (e.g. Casaburi et al 2021; Shao et al 2019; and others). Readers might interpret these findings to be true of all infants. Instead, I would recommend either removing the 'as expected' or clarifying the statement, e.g. "As expected for this cohort" and refer to a clinical demographics table, ideally a summary of participants similar to what is presented for the SKOT cohort (Supplementary Table 1b) but is not presented for the SIG cohort.

2. Similar to above, Line 419 writes, "Yet, specifically the symbiotic role of the breastmilk-promoted Bifidobacterium species, which are highly abundant in breastfed infants." Despite exclusive breastfeeding, a substantial proportion of infants do not develop Bifidobacterium populations early in life despite exclusive breastfeeding (e.g. Frese et al 2017; Casaburi et al 2021; Shao et al 2019; Nguyen et al 2021). While more infants acquire Bifidobacterium *later* in infancy, the comment lacks some specificity (i.e. while infant-associated Bifidobacterium are abundant in breastfed infants where they are present, not all infants harbor these organisms despite exclusive breastfeeding). I would recommend the authors rephrase the sentence to reflect this. One suggestion would be to write, "...which are highly abundant among [many/a

large proportion of] breastfed infants.”

3. Line 443: The authors describe the impact of ILA on IL-12p70 secretion by monocytes during LPS challenges and conclude: “...LPS derived from Enterobacteriaceae species that often co-inhabit the neonatal/infant gut.” Interestingly, Enterobacteriaceae abundance appears to be inversely related to Bifidobacterium colonization (e.g. Nguyen et al 2021; Frese et al 2017). I would recommend the sentence be rephrased to remove the ‘co-inhabit’ concept. For example, “Enterobacteriaceae species that can also inhabit the neonatal/infant gut.”

Minor comments:

Line 187: The use of “Assay” might be improved by saying “Assays performed...” to describe the enzymatic experiments performed.

Line 624: I assume that it’s known within the field that DSM20088 is equivalent to ATCC 15697, but it might be useful to note this in the text for other readers as there is substantial work done on this strain under that moniker.

Supplementary Table 2a: It would be helpful to have a summary of the participants similar to Supplementary Table 1B outlining the demographics of the participants (e.g. sex, diet, delivery mode).

Supplementary Figure 11a/b. Very minor, but similar to a comment above, it would be helpful to get a sense of the distribution beyond the mean in this figure and in the text. Even adding the standard deviations to the legend in 11a would be helpful, or if it’s simpler, adding a table of these OTUs and their mean abundance +/- standard deviation among the infant ages. For example, roughly half of the infants in the cohort appear to have low/no *B. longum* for all of their samples, but the rest had ~90%+, and only showing the mean hides this divergent colonization.

Note: Any papers referenced above need not be cited in the manuscript, they’re only provided as supporting information.

Reviewer #2 (Remarks to the Author):

This is a review of a manuscript entitled “Breastmilk-promoted bifidobacteria produce aromatic amino acids in the infant gut” authored by Laursen et al. The goal of this study was to evaluate whether the microbiota of breastmilk fed infants produce metabolites that might be implicated in health benefits associated with breastfeeding. To evaluate this, the authors performed 16S rRNA microbiota profiling and targeted metabolomics (focusing on aromatic amino acids and their metabolites) in feces from infants. These analyses recapitulated previous findings supporting the notion that bifidobacteria are increased in the feces of breastfed infants. The authors also identified a positive correlation between *Bifidobacterium* spp. and indolelactate (ILA), phenyllactate (PLA), and 4-OH-phenyllactate (4-OH-PLA), suggesting that *Bifidobacterium* in the gut may either produce or modulate the production of these molecules. The authors provide evidence for this activity in vitro, showing that certain *Bifidobacterium* strains produce these three molecules, and identify a lactate dehydrogenase homolog (type 4 *ldh*) potentially implicated in this pathway. Using a combination of genetics and biochemistry, they demonstrate that the type 4 *ldh* gene is involved in conversion of aromatic amino acids to ILA, PLA, and 4-OH-PLA. Next, the authors monitor the levels of *Bifidobacterium* strains and metabolite concentrations longitudinally in the feces of a separate cohort of infants. These findings further support the association between ILA/PLA/4-OH-PLA producing strains and the abundance of these metabolites in the feces. Finally, the authors ask whether ILA (one of the more abundant aromatic amino acid metabolites) could influence host phenotypes. To address this,

they test the capacity of infant feces to activate the arylhydrocarbon receptor in reporter cell lines and demonstrate a correlation between fecal Bifidobacterium abundance, ILA concentrations, and Ahr activity. They also show that ILA modulates IL-22 production ex-vivo in human T-cells in an Ahr-dependent manner, and modulates IL-12p70 production by stimulated human monocytes dependent on both Ahr and HCAR3. Collectively, these findings suggest that breastfeeding promotes Bifidobacterium colonization which in turn modulates ILA production in the gut that could influence immune cell phenotypes in infants.

Overall, this work provides an exceptional example of how microbiome/metabolomics studies can uncover mechanistic roles of the gut microbiota and their metabolites in human health. The manuscript is well written and follows a logical trajectory. The mechanistic insights from this work are likely to provide a solid foundation onto which future studies can evaluate how the gut microbiota in infants influences physiology and immune system development. While overall enthusiastic, I do have some concerns about the manuscript that I believe should be addressed. Please find major and minor comments below:

Major Comments:

1) The claim that bifidobacteria produce aromatic amino acid metabolites in the gut is not fully supported by the data. The authors show a strong correlation between aromatic amino acid metabolites and Bifidobacterium species abundance (especially those strains that encode type 4 Idh). However, there is no direct evidence that these organisms convert aromatic amino acids to their lactate derivatives in the infant gut. Alternative possibilities include a) Bifidobacteria modulate the metabolism of other microbes in the gut that produce aromatic lactic acids, b) Other microbes are major producers of aromatic lactic acids and either do not change in abundance or are below the limit of detection by 16S rRNA profiling. The issue is with the language the authors use – moving from trends, correlations and associations to causation in the absence of direct data. Exaggerated claims are present throughout the manuscript including the title, abstract (lines 26 and 44), introduction (lines 66-67), results (line 229, lines 289-291, line 299) and discussion (line 433-434). I would suggest changing the language throughout the manuscript to lessen the claims that bifidobacteria produce aromatic lactic acids in the gut.

2) The mass spectrometry methods for quantitation of aromatic metabolites in the feces of infants is missing a key validation. The authors do not show analyte recovery for the individual metabolites during their sample preparation. Normally this could be accounted for if a stable isotopically labeled internal standard was used for each individual metabolite, however the authors use 4 internal standards to cover 19 different analytes. If the recovery varies among analytes, this could have a large impact on the concentrations reported. The authors should determine the analyte recovery for each of the 19 compounds and report this information in the methods section and acknowledge whether differences in analyte recovery might affect data presented in the figures and tables.

Minor comments:

- 1) In the title, “aromatic amino acids” should be “aromatic lactic acids” or “aromatic amino acid metabolites”
- 2) Abstract, line 30: The implicit assumption is that metabolites mediate health effects, this may not be obvious to the reader.
- 3) Page 3, line 47: Is anything in biology “perfect”? Suggest modifying the sentence, perhaps to “Human breastmilk is a nutritional supply adapted for the infant”
- 4) Page 8, line 175: If the authors used a complex medium (this wasn’t clear from the results or figure legend), it may mask the phenotype of the Idh disruption mutant.
- 5) Page 9, line 201: A relatively small number of substrates were tested (pyruvate, indolepyruvate, phenylpyruvate, and 4-OH-phenylpyruvate). To re-classify this enzyme, I would expect a broader suite of substrates being tested.
- 6) Page 12, line 246: “individual” should be “individualized”

- 7) Figure 5: The analysis of IL-12p70 production by stimulated monocytes is interesting but superficial. If ILA signals through both HCAR3 and AhR to reduce IL-12p70 production, are the results additive? This could be tested by incubating cells with both HCAR3 siRNA and an AhR antagonist.
- 8) Page 20, lines 422-425: It would be helpful if the authors would provide a comprehensive comparison of the type 4 Idh from bifidobacteria and other homologs that have been characterized as well as from other gut bacteria. This could be in the form of a supplemental table showing the AA percent identities between characterized and uncharacterized homologs.
- 9) Page 21, lines 451-452: This sentence seems like a gross overstatement. There is no data in the manuscript showing that aromatic lactic acids play a central role in mediating host-microbiota interactions.
- 10) Figure 3 legend: How was the relative fecal abundance of HMOs calculated? It isn't clear from the legend, text, or methods sections.
- 11) Page 35, line 778: it is surprising that metabolite abundances were not normalized to urine creatinine. The lack of normalization would likely lead to large inter-individual differences in metabolite levels.
- 12) Figure 1d: The concentrations for 4-OH-PLA, PLA, and ILA are much lower here than in the Copenhagen cohort, but this is not addressed in the manuscript. Do the authors believe this reflects biological differences among the cohorts or is a technical artifact of sampling or sample processing?

Reviewer #3 (Remarks to the Author):

Laurenson et al. suggest that breastmilk-promoted Bifidobacterium species convert aromatic amino acids into their respective aromatic lactic acids via a previously unrecognised aromatic lactate dehydrogenase. The authors analyzed multiple layers of data (publically available, proprietary longitudinal) to inform mechanistic in vitro and ex vivo experiments. This is a commendable translational approach tackling a biologically important question (while breastfeeding is known to strongly influence the microbiome and infant immune development, the mechanisms are not well understood.) The in vitro and ex vivo work is compelling; however, the low microbiome data quality as well as the statistical analyses often do not support the authors' strong claims. Our main issue is the discrepancy between the low resolution of the 16S amplicon sequencing data (sequencing depth of 2000-8000) along with the unjustified statistical analyses, and the strength of the conclusions the authors are presenting. For example, we are not convinced that one can BLAST 16S data with such low coverage and report subspecies resolution. We therefore request that the authors use deeply sequenced metagenomic data and provide evidence that their conclusions hold. Other specific concerns are outlined below.

Major comments:

1. CIG cohort - 25 infants, sampled for ~10 time points from birth to 6 months of age (16S with qPCR, sequencing depth ~8,000 + targeted metabolomics).
 - a. The authors collapsed the data and ended up with 39 bacterial taxa. We do not think it is likely to find strain resolution in this dataset.
 - b. (lines 243-244) "the relative abundance of Bifido increased with time..." this is a problematic statement due to the compositional nature of the microbiome data. Other taxa can change (while bifido doesn't change) and still (artificially) affect Bifido's relative abundance.
 - c. (line 246-250) This is longitudinal data, but the analysis does not account for repeated measures. The authors include the same subject multiple times. In parallel, the authors claim that the data structure is explained by subject! The dimensionality reduction must take into account the fact you have repeated measures data.
 - d. (lines 259-262) "The gut microbiota of individuals dominated by infant type Bifidobacterium spp. was more stable over time than in individuals with a gut microbiota not dominated by infant type Bifidobacterium spp. ($p < 0.0001$, Mann-Whitney U test)" - How did the authors come up with this definition of stability?

- e. Partial Spearman's Rank correlation analyses with adjustments for age and repeated measures correlation analyses were performed in R using the ppcor and rmcrr packages.
- i. partial Spearman's Rank correlation coefficients - partial correlations are used to express the specific portion of variance explained by eliminating the effect of other variables when assessing the correlation between two variables. Why was this analysis used? What is the statistical justification?
- ii. The authors performed one analysis that accounts for repeated measures data (i.e., spearman correlation for repeated measures). Nonetheless, this analysis quantifies a correlation within an individual and not across individuals. This is different from the analysis the authors performed on the cross sectional data (SKOT I) and is therefore not comparable.
- f. (lines 270-272): "We have thus established a link between breastfeeding, degradation of HMOs, abundance of specific Bifidobacterium spp. and concentrations of aromatic lactic acids in early infancy. This is a very strong statement that is not backed by proper data and statistical analyses.
- g. Figure 3a-c: based on repeated measures data, thus can not make any conclusions across individuals (as they did in the analysis of the SKOT cohort).
- h. Figure 3e - there is a discrepancy between the qPCR and PCR results (absolute vs. relative abundance) in in terms of spearman correlation between the abundance of bifido species and ecal levels of ILA, PLA and 4-OH-PLA (positive) and HMOs (negative). Is the sample size identical?
- i. Figure 3e - why is there a discrepancy between the correlation results of the amplicon and qPCR ?
- j. (lines 273 - 277+ Figure 3 f-h) - this is descriptive and lacking statistical validation. This does not indicate anything about the "dynamics". No statistical analyses to support claims like: "breastfed infants early and consistently colonised by infant type Bifidobacterium species consistently showed high concentrations of aromatic lactic acids in faeces..." . It seems more anecdotal than anything else.
- k. (lines 277 - 291) - While it is interesting to see individual trajectories with granular details, these are ultimately descriptive data from a relatively small number of infants. Any inferences drawn from this data should be tempered. We question whether (even after reanalysis as recommended above) these data should be included in the paper at all, as they do not provide strong evidence to support the overall story. (lines 289-291) - "This indicates that bifidobacterial aromatic lactic acid production is compromised by pre-term delivery, exposure to antibiotics and formula supplementation". The authors had just 2 preterm deliveries and 3 infants receiving antibiotics! This is another unbacked claim.

2. Conceptual questions

- a. How relevant is the ex vivo assay with immune cells from blood, given that Bifidobacterium-produced ILA is located in the gut? Do the authors believe that ILA is absorbed into circulation? They show that fecal and urinary levels are correlated - is all urinary ILA necessarily derived from gut microbiota? If so, please clarify and provide citations.
- b. How relevant are the stool concentrations of ILA? Do they reflect concentrations in the gut?
- c. 19 aromatic amino acids and derivatives were measured. Does this capture all or the majority of such compounds expected in the infant gut?
- d. Short chain fatty acids are bacterial metabolites that have been studied in relation to infant immune function and development. This evidence base is relevant but not discussed.

3. General analysis issues:

- a. Why did the authors use Qiime and not Qiime 2? Why OTUs and not ASVs? How sensitive are the results to the denoising and clustering method? Taxonomic annotation is more reliable with DADA2 for example as compared with OTUs.
- b. The sequencing depth is very very low (2000 SKOT - 8000 CIG).
- c. To our knowledge, using BLAST on 16S data to find subspecies resolution is not reliable. You need metagenomic data to do that.
- d. Absolute abundances of B.longum subsp. Longum and B. longum subsp. Infantis (subspecies specific primers) were estimated by qPCR. Why didn't the authors use subspecies specific primers for other subspecies?
- e. A lot of the statements are lacking statistical justifications and analyses (e.g., lines 243-244 "on average

the microbial composition and shannon diversity did not change dramatically..."; lines 262-264 - "Using solely samples from breastfed infants, faecal abundances of HMO residuals showed a progressive decline with age, concurrent with the progressive increase in infant type Bifidobacterium species").

f. The authors would revise the methods section and add significantly more details on the statistical analysis. Just writing the packages used is not good enough.

g. (line 825) Metabolomics data - The data were normalised to the total intensity. That is wrong. The data should be normalized across samples and not within a sample.

Minor comments:

The authors claim to find an association between Bifidobacterium subspecies and aromatic amino acid catabolites during late infancy using 16S with sequencing depth of 2,000 + targeted metabolomics data.

SKOT I cohort -

59 healthy Danish infants from the SKOT I cohort (published in 2010). The infants were born full term, ~9 of age at sampling with ~40% partially breastfeed and ~60% weaned.

4. The authors claim (line 85) that "other metadata (age, gender, mode of delivery, current formula intake and age of introduction to solid foods) did not explain gut microbiota variation to the same degree as breastfeeding status (Supplementary Data 1c,d) - how was this tested?

5. (line 87) "Principal Component Analysis (PCA) of faecal aromatic amino acid metabolite concentrations also revealed..." - what data transformation was used for the microbiome data? For the metabolites data? The microbiome data is compositional and the metabolite data need to be normalized per component. In any case, this separation doesn't seem significant (also - no p-value reported).

6. (line 90) "Correlation analysis revealed that Bifidobacterium, but no other bacterial genera, were significantly associated with faecal concentrations of all three aromatic lactic acids (4-OH-PLA, PLA and ILA), in addition to indolealdehyde (IAld). How was this correlation performed? Did the authors transform the data (z-score per metabolite and some log ratio transformation for the microbiome)?

7. (line 93) 16S data with 2,000 reads does not seem sufficient resolution for bifido spp. identification (e.g., B. longum, bifidum, breve).

8. (line 109) "Together, this suggests that specific breastmilk-promoted Bifidobacterium species in the gut of infants convert aromatic amino acids to the corresponding aromatic lactic acids" - This is a mechanistic statement. Given the analyses performed and associations found, the authors can not make this claim at this stage of the paper, before the mechanistic studies are presented.

Author Rebuttal to Initial comments

Reviewer #1 (Remarks to the Author):

The authors detail a mechanistic dissection of the molecular pathways behind the production of aromatic amino acids, their pathway of recognition by host cells, and the impacts on immune cells in infants. The findings represent a significant advancement in both the study of host-associated microbes and human infant gut symbionts. Through a series of exhaustive and confirmatory experiments, they describe and demonstrate a compelling story that confirms the production, recognition, and outcomes of aromatic amino acid production in the infant gut and demonstrate these observations via clinical samples obtained from infant stools.

The paper itself is very well written, logically organized, and I have very few recommendations for improving the manuscript further, as outlined below. I really enjoyed reading the paper, and the thoroughness and

attention to detail is impressive. There are obvious implications of this work to intervention studies examining infant health and immune development, but I respect that this may be beyond the scope of the current manuscript. I look forward to seeing future extensions of this work from the authors.

Answer: We thank the reviewer for the very positive and encouraging feedback. We are indeed really looking forward to share our work with the community and see how it will stimulate further work in relation to infant health and immune development.

I would like to make a few humble recommendations to the authors:

1. Line 238-243: It would be helpful to indicate the standard deviations, and ideally, the number/percent of infants which met the 'high' Bifidobacterium criterion used for the "early colonized infants" (>40% Bifidobacterium) elsewhere in the paper. I would also advise against "As expected.." to describe Bifidobacterium colonization among infants because this is not generalizable. While C-section delivery was not apparently an exclusion criterion for participation, the number of C-section delivered infants was very low in both the SKOT and CIG cohorts. C-section delivery has been shown to inhibit maternal Bifidobacterium transfer (e.g. Shao et al 2019), so these findings might be expected for *this* population, but are not necessarily generalizable (e.g. Casaburi et al 2021; Shao et al 2019; and others). Readers might interpret these findings to be true of all infants. Instead, I would recommend either removing the 'as expected' or clarifying the statement, e.g.

"As expected for this cohort" and refer to a clinical demographics table, ideally a summary of participants similar to what is presented for the SKOT cohort (Supplementary Table 1b) but is not presented for the SIG cohort.

Answer: We agree with the reviewer and have now added standard deviations for all CIG taxa in Supplementary Data 2c. Furthermore, we have modified the sentence to:

Lines 247-249: "As expected **from a cohort mainly containing vaginally born, breastfed infants**, the gut microbiota was highly dominated by Bifidobacterium (average of 64.2 %)"

As suggested by the reviewer, we have also added the number of infants grouped into early colonized and late colonized (in addition to the three antibiotics-associated already mentioned) in the text:

Lines 287-293: "Examination of the Bifidobacterium and aromatic lactic acid dynamics in each of the 25 infants during the first 6 months of life (Fig. 3d-f and Supplementary Fig. 19-20) revealed that breastfed infants early colonized by infant type Bifidobacterium species (**n = 17**) consistently showed high concentrations of aromatic lactic acids in faeces (Fig. 3d and Supplementary Fig. 20a). In contrast, infants with delayed infant type Bifidobacterium species colonisation (**n = 5**) showed considerably lower concentrations of the aromatic lactic acids, in particular of ILA, despite breastfeeding (Fig. 3e and Supplementary Fig. 20b)."

2. Similar to above, Line 419 writes, "Yet, specifically the symbiotic role of the breastmilk-promoted Bifidobacterium species, which are highly abundant in breastfed infants." Despite exclusive breastfeeding, a substantial proportion of infants do not develop Bifidobacterium populations early in life despite exclusive breastfeeding (e.g. Frese et al 2017; Casaburi et al 2021; Shao et al 2019; Nguyen et al 2021). While more infants acquire Bifidobacterium *later* in infancy, the comment lacks some specificity (i.e. while infant-associated Bifidobacterium are abundant in breastfed infants where they are present, not all infants harbor these organisms despite exclusive breastfeeding). I would recommend the authors rephrase the sentence to reflect this. One suggestion would be to write, "...which are highly abundant among [many/a large proportion of] breastfed infants."

Answer: We agree and have followed the referee's suggestion:

Lines 408-409: "Yet, specifically the symbiotic role of the breastmilk-promoted Bifidobacterium species, which are highly abundant in **many** breastfed infants, remains largely unknown"

3. Line 443: The authors describe the impact of ILA on IL-12p70 secretion by monocytes during LPS challenges and conclude: "...LPS derived from Enterobacteriaceae species that often co-inhabit the neonatal/infant gut." Interestingly, Enterobacteriaceae abundance appears to be inversely related to Bifidobacterium colonization

(e.g. Nguyen et al 2021; Frese et al 2017). I would recommend the sentence be rephrased to remove the ‘co-inhabit’ concept. For example, “Enterobacteriaceae species that can also inhabit the neonatal/infant gut.”

Answer: We agree and have accommodated the referee’s suggestion:

Lines 430-434: “Further, the AhR and HCAR3-dependent inhibitory effect of ILA on IL-12p70 secretion by monocytes may constitute a means by which infant type Bifidobacterium species contribute to the regulation of the pro-inflammatory responses to LPS derived from Enterobacteriaceae species that also often colonise the neonatal/infant gut”

Minor comments:

Line 187: The use of “Assay” might be improved by saying “Assays performed...” to describe the enzymatic experiments performed.

Answer: We have changed according to the referee’s suggestion.

Reviewer: Line 624: I assume that it’s known within the field that DSM20088 is equivalent to ATCC 15697, but it might be useful to note this in the text for other readers as there is substantial work done on this strain under that moniker.

Answer: We have added the ATCC number (Line 619).

Supplementary Table 2a: It would be helpful to have a summary of the participants similar to Supplementary Table 1B outlining the demographics of the participants (e.g. sex, diet, delivery mode).

Answer: We agree with the reviewer and have now included a summary table in the new Supplementary data 2b.

Supplementary Figure 11a/b. Very minor, but similar to a comment above, it would be helpful to get a sense of the distribution beyond the mean in this figure and in the text. Even adding the standard deviations to the legend in 11a would be helpful, or if it’s simpler, adding a table of these OTUs and their mean abundance +/- standard deviation among the infant ages. For example, roughly half of the infants in the cohort appear to have low/no B. longum for all of their samples, but the rest had ~90%+, and only showing the mean hides this divergent colonization.

Answer: Mean abundances for all species are shown in Supplementary data 2c. Standard deviations have now been added.

Note: Any papers referenced above need not be cited in the manuscript, they’re only provided as supporting information.

Reviewer #2 (Remarks to the Author):

This is a review of a manuscript entitled “Breastmilk-promoted bifidobacteria produce aromatic amino acids in the infant gut” authored by Laursen et al. The goal of this study was to evaluate whether the microbiota of breastmilk fed infants produce metabolites that might be implicated in health benefits associated with breastfeeding. To evaluate this, the authors performed 16S rRNA microbiota profiling and targeted metabolomics (focusing on aromatic amino acids and their metabolites) in feces from infants. These analyses recapitulated previous findings supporting the notion that bifidobacteria are increased in the feces of breastfed infants. The authors also identified a positive correlation between *Bifidobacterium* spp. and indolelactate (ILA), phenyllactate (PLA), and 4-OH-phenyllactate (4-OH-PLA), suggesting that *Bifidobacterium* in the gut may either produce or modulate the production of these molecules. The authors provide evidence for this activity in vitro, showing that certain *Bifidobacterium* strains produce these three molecules, and identify a lactate dehydrogenase homolog (type 4 *ldh*) potentially implicated in this pathway. Using a combination of genetics and biochemistry, they demonstrate that the type 4 *ldh* gene is involved in conversion of aromatic amino acids to ILA, PLA, and 4-OH-PLA. Next, the authors monitor the levels of *Bifidobacterium* strains and metabolite concentrations longitudinally in the feces of a separate cohort of infants. These findings further support the association between ILA/PLA/4-OH-PLA producing strains and the abundance of these metabolites in the feces. Finally, the authors ask whether ILA (one of the more abundant aromatic amino acid metabolites) could influence host phenotypes. To address this, they test the capacity of infant feces to activate the arylhydrocarbon receptor in reporter cell lines and demonstrate a correlation between fecal *Bifidobacterium* abundance, ILA concentrations, and Ahr activity. They also show that ILA modulates IL-22 production ex-vivo in human T-cells in an Ahr-dependent manner, and modulates IL-12p70 production by stimulated human monocytes dependent on both Ahr and HCAR3. Collectively, these findings suggest that breastfeeding promotes *Bifidobacterium* colonization which in turn modulates ILA production in the gut that could influence immune cell phenotypes in infants.

Overall, this work provides an exceptional example of how microbiome/metabolomics studies can uncover mechanistic roles of the gut microbiota and their metabolites in human health. The manuscript is well written and follows a logical trajectory. The mechanistic insights from this work are likely to provide a solid foundation onto which future studies can evaluate how the gut microbiota in infants influences physiology and immune system development. While overall enthusiastic, I do have some concerns about the manuscript that I believe should be addressed. Please find major and minor comments below:

Answer: We thank the reviewer for the very positive remarks and for highlighting our study as an example of how microbiome studies can uncover mechanistic roles of the gut microbiota and their metabolites in human health.

Major Comments:

1) The claim that bifidobacteria produce aromatic amino acid metabolites in the gut is not fully supported by the data. The authors show a strong correlation between aromatic amino acid metabolites and *Bifidobacterium* species abundance (especially those strains that encode type 4 *ldh*). However, there is no direct evidence that these organisms convert aromatic amino acids to their lactate derivatives in the infant gut. Alternative possibilities include a) *Bifidobacterium* modulate the metabolism of other microbes in the gut that produce aromatic lactic acids, b) Other microbes are major producers of aromatic lactic acids and either do not change in abundance or are below the limit of detection by 16S rRNA profiling. The issue is with the language the authors use – moving from trends, correlations and associations to causation in the absence of direct data. Exaggerated claims are present throughout the manuscript including the title, abstract (lines 26 and 44), introduction (lines 66-67), results (line 229, lines 289-291, line 299) and discussion (line 433-434). I would suggest changing the

language throughout the manuscript to lessen the claims that bifidobacteria produce aromatic lactic acids in the gut.

Answer: We do think that our data convincingly demonstrate that the bifidobacteria produce aromatic lactic acids in the infant gut. We concluded this based on a series of exhaustive and confirmatory mechanistic experiments as already outlined in the manuscript.

To further strengthen our conclusion, we have added an animal experiment and performed confirmatory statistical analyses as outlined below:

To provide direct evidence that the aromatic lactic acids are produced by Bifidobacteria *in vivo* in the gut, we include a study of mono-colonized germ-free mice with the WT *B. longum* 105-A strain and the generated isogenic type 4 *ldh* KO mutant, respectively. Both strains (WT and KO) colonized the mice equally well and we confirmed the production of aromatic lactic acids via the aromatic lactate dehydrogenase *in vivo*. Consistent with our *in vitro* experiments, a 20-60 fold difference in concentrations of aromatic lactic acids in the cecum between the WT and type 4 *ldh* mutant groups was observed.

This has now been included in the manuscript (For method details see lines 753-771).

Lines 184-187: "Further, to demonstrate *in vivo* production of the indicated aromatic lactic acids, we mono-colonised germ free mice with either the WT or the type 4 *ldh* mutant strain and found a 20-60 fold increase in their concentrations in WT versus type 4 *ldh* mutant mono-colonised mice (Supplementary Fig. 8)"

Supplementary Figure 8. *In vivo* production of aromatic lactic acid in previously germ free mice colonised with either *B. longum* 105-A WT or type 4 *ldh* mutant.

a, CFU counts of *B. longum* 105-A WT or type 4 *ldh* mutant from caecal content of previously germ free mice mono-colonised with either the WT or type 4 *ldh* mutant strain. Bars and error bars indicate mean ± sd. **b-d**, caecal concentrations of the aromatic lactic acids. Line and error bars indicate mean ± sd. Statistical significance was evaluated by Mann Whitney U tests, with **** $p < 0.0001$.

Regarding the alternative possibilities suggested by the reviewer, we find these highly unlikely.

Regarding possibility a, if Bifidobacteria would modulate the metabolism of other microbes, we would still expect to see correlations between other microbes and aromatic lactic acids. In the SKOT I cohort, we did not find any other bacterial genera than *Bifidobacterium* that correlated with the aromatic lactic acids after correction for multiple testing (Supplementary Table 1c). We have now also added data from the CIG cohort showing that only the relative abundance of *Bifidobacterium* (and specifically the infant type species) is consistently positively associated with the aromatic lactic acids across the different time points (new Supplementary Fig. 16).

a

b

Supplementary Figure 16. Correlations between relative abundance of bacterial taxa and concentrations of the aromatic lactic acids at each sampling point in the CIG cohort.

a, Spearman's rank correlations between relative abundance of faecal bacterial genera (average relative abundance > 1%) and concentrations of the aromatic lactic acids at each sampling point. b, Spearman's rank correlations between relative abundance of faecal *Bifidobacterium* species (average relative abundance > 0.1%) and concentrations of the aromatic lactic acids at each sampling point. Infant type *Bifidobacterium* spp. is the sum of the relative abundances of *B. longum*, *B. breve*, *B. bifidum* and *B. scardovii*. Statistical significance was evaluated by uncorrected *p*-values indicated by asterisks with * *p* < 0.05, ** *p* < 0.01, *** *p* < 0.001 and **** *p* < 0.0001.

Furthermore, we have added linear mixed model analyses adjusted for subject and age to look for associations between relative abundance of the 39 CIG species and the aromatic lactic acids (Supplementary Data 2e), and

between the absolute abundance of specific *Bifidobacterium* species/subspecies and aromatic lactic acids (new Fig. 3c and Supplementary Fig. 17).

These analyses consistently show that *B. longum*, *B. breve* and *B. bifidum* are the main species (highest effect size, top 10 with highest positive effect size shown in table shown below) significantly associated with ILA, PLA and 4-OH-PLA (Supplementary Data 2e) and using absolute abundances of the *Bifidobacterium* species make these associations even stronger (New Fig. 3c and Supplementary Fig. 17).

Supplementary Data 2e. Associations between relative abundances of bacterial taxa and faecal concentrations of the aromatic lactic acids in the IG study assessed by MaAsLin2

Aromatic lactic acid	Bacterial taxon	Subject and age adjusted β -coefficient	Standard derivation	Number of samples	Number of samples with taxon detected	P-value	FDR-corrected P-value ^a
ILA	B. longum	0.595859251	0.096429039	240	221	2.91E-09	1.13515E-07
OHPLA	B. longum	0.485384354	0.092382466	240	221	3.44E-07	6.71323E-06
OHPLA	B. bifidum	0.370667935	0.074256254	240	181	1.18E-06	1.53192E-05
PLA	B. bifidum	0.315634605	0.07076952	240	181	1.29E-05	0.00168156
ILA	B. breve	0.310829809	0.08099187	240	132	0.00016	0.00124621
PLA	B. longum	0.236861029	0.090242016	240	221	0.009277	0.04525992
ILA	B. bifidum	0.229558219	0.081455473	240	181	0.005239	0.022702634
OHPLA	B. breve	0.186741878	0.077289244	240	132	0.016474	0.058406055
PLA	B. breve	0.156068608	0.073074196	240	132	0.033789	0.109813998
PLA	E. faecalis	0.148318493	0.028463825	240	127	4.26E-07	1.66028E-05
ILA	E. faecalis	0.123700572	0.031654733	240	127	0.000126	0.001228209
OHPLA	L. rhamnosus/casei/paracasei	0.10706962	0.028427122	240	120	0.000211	0.001642506
PLA	L. rhamnosus/casei/paracasei	0.09491794	0.026977449	240	120	0.000525	0.003414692
ILA	B. scardovii	0.070322276	0.019128201	240	12	0.000294	0.001912063
PLA	Enterococcus spp.	0.0614202	0.041825697	240	117	0.143372	0.31063962
PLA	E. ramosum	0.058773013	0.029006176	240	58	0.044096	0.132287505
ILA	Enterococcus spp.	0.0565037	0.04681862	240	117	0.228726	0.505965045
PLA	E. lenta	0.055794731	0.013872816	240	101	8.11E-05	0.000632392
PLA	B. scardovii	0.051924191	0.01705882	240	12	0.002618	0.014585844
OHPLA	B. scardovii	0.046896457	0.018206512	240	12	0.010637	0.041484757

Importantly, we do not claim that *Bifidobacterium* species are the only species in the infant gut capable of producing the aromatic lactic acids, but rather that they are the major contributors.

Regarding possibility b, we find it highly unlikely that low abundant / non-detected species should make a significant contribution to the measured concentration of the aromatic lactic acids. These low abundant species would be more than 1000-fold lower than the *Bifidobacterium* species (given the vast dominance of *Bifidobacterium* and a detection limit of 1/8000 reads), which we know from our series of experiments (discussed above) are actual producers of the aromatic lactic acids.

Altogether, we think this is sufficient to conclude that *Bifidobacterium* species produce aromatic lactic acids in the infant gut.

2) The mass spectrometry methods for quantitation of aromatic metabolites in the feces of infants is missing a key validation. The authors do not show analyte recovery for the individual metabolites during their sample preparation. Normally this could be accounted for if a stable isotopically labeled internal standard was used for each individual metabolite, however the authors use 4 internal standards to cover 19 different analytes. If the recovery varies among analytes, this could have a large impact on the concentrations reported. The authors should determine the analyte recovery for each of the 19 compounds and report this information in the methods section and acknowledge whether differences in analyte recovery might affect data presented in the figures and tables.

Answer: We agree that it would have been ideal with isotope-labeled internal standards for each single analyte in the present study. It would have provided more accurate concentrations as the recovery rates varied among the internal standards (average recovery in stool compared to water ≈72%, 51%, 52% and 30%). However, the recovery of the internal standards relative to each other were in general rather consistent as exemplified by the relative recoveries (Internal standard peak area in faecal sample divided by internal standard peak area in water) of internal standards in stool samples of six individuals (see figure below).

Consequently, while an improved quantitative LC-MS method could have provided more accurate measurements, the relative metabolite concentrations across samples would not have changed and thus the conclusions presented in the manuscript would not be affected.

Yet, to acknowledge the limitations of the quantitative LC-MS method correctly pointed out by the reviewer, we have now specified in Supplementary Table 1 (metabolite list), Supplementary Table 2 (metabolite concentrations in the SKOT cohort), and Supplementary Table 4 (metabolite concentrations in the CIG cohort), that four internal standards were used to obtain semi-quantitative metabolite concentrations.

Finally, we have specified in the method section that the metabolites were quantified using a semi-quantitative method.

Lines 815-817: “Aromatic amino acids and derivatives (Supplementary Table 1) of faecal and *in vitro* samples were quantified by a semi-quantitative ultra performance liquid chromatography mass spectrometry (UPLC-MS) method (ref 91)”

Minor comments:

1) In the title, “aromatic amino acids” should be “aromatic lactic acids” or “aromatic amino acid metabolites”
Answer: We thank the referee for noticing this – this is an error. It has now been corrected to ‘aromatic lactic acids’.

2) Abstract, line 30: The implicit assumption is that metabolites mediate health effects, this may not be obvious to the reader.

Answer: We agree. We have adjusted the abstract:

Lines 29-32: “Breastfeeding profoundly shapes the infant gut microbiota, which is critical for early life immune development. One way by which the gut microbiota can impact host physiology is through the production of metabolites. However, few breastmilk-dependent microbial metabolites mediating host-microbiota interactions are currently known.

3) Page 3, line 47: Is anything in biology “perfect”? Suggest modifying the sentence, perhaps to “Human breastmilk is a nutritional supply adapted for the infant”

Answer: We agree and have modified the sentence:

Line 49: “Human breastmilk is a well-adapted nutritional supply for the infant”

4) Page 8, line 175: If the authors used a complex medium (this wasn’t clear from the results or figure legend), it may mask the phenotype of the *ldh* disruption mutant.

Answer: We agree that we cannot know for sure how the phenotype would look in other (minimal/defined) media. We have now specified that it did not impair growth in a rich medium.

Lines 178-180: “Cultivation of the wild-type (WT), the type 4 *ldh* mutant strain and a complemented type 4 *ldh* mutant strain in a medium containing the three aromatic amino acids (Supplementary Fig. 6) confirmed that type 4 *ldh* disruption did not impair growth **in rich medium** (Fig. 2d).”

5) Page 9, line 201: A relatively small number of substrates were tested (pyruvate, indolepyruvate, phenylpyruvate, and 4-OH-phenylpyruvate). To re-classify this enzyme, I would expect a broader suite of substrates being tested.

Answer: We agree and have omitted the sentence about re-classification.

6) Page 12, line 246: “individual” should be “individualized”

Answer: Has been corrected (Line 245-247).

7) Figure 5: The analysis of IL-12p70 production by stimulated monocytes is interesting but superficial. If ILA signals through both HCAR3 and AhR to reduce IL-12p70 production, are the results additive? This could be tested by incubating cells with both HCAR3 siRNA and an AhR antagonist.

Answer: We agree it would be interesting to address the individual contribution of HCAR3 and AhR in more detail. It is however, a quite difficult experiment to conduct at present until HCAR3 specific antagonists are available. When trying to investigate individual receptor contributions, we identified that addition to human monocytes of HCAR3 siRNA as compared to scramble siRNA resulted in variable expression of the AHR gene depending on the donor; in some donor monocytes, it reduced the expression to add HCAR3 siRNA vs scramble siRNA, while in others it enhanced the expression. The average effect being 1.0185 +/- 0.2998 (HCAR3/scramble siRNA (average +/- SD)). Addition of siRNA is therefore not an AHR ‘undisturbing’ treatment in all donor monocytes, and we therefore believe it would require much larger data to deduce the actual contribution of each of the HCAR3 and AHR receptors in human beings. We find that this is beyond the scope of this paper. Based on our data, we believe it is correct to state that both human HCAR3 and AHR can bind to the aromatic lactic acids (based on receptor assay data) and that ILA can mediate immune regulation (suppression of the type 1 response) via both HCAR3 and AHR in human monocytes. We would like to refrain from emphasizing individual receptor contributions, and hope the peer reviewer can acknowledge this decision based on the above explanation.

8) Page 20, lines 422-425: It would be helpful if the authors would provide a comprehensive comparison of the type 4 *ldh* from bifidobacteria and other homologs that have been characterized as well as from other gut bacteria. This could be in the form of a supplemental table showing the AA percent identities between characterized and uncharacterized homologs.

Answer: We have already analyzed the presence of type 4 *ldh* gene in all available whole genome sequenced human gut derived *Bifidobacterium* species (Supplementary Table 3). This analysis revealed the universal presence of the type 4 *ldh* in *B. longum*, *B. breve*, *B. bifidum*, *B. scardovii* and *B. angulatum* strains (the latter was not detected in our cohorts and usually is not detected before the age of 3 years, see 10.1038/s41598-017-10711-5), but the gene was not found in other human associated *Bifidobacterium* species. In addition, Supplementary Fig. 7 show the alignment of the type 4 LDH amino acid sequences from *B. longum* 105-A and the type strains of *B. longum* subsp. *longum*, *B. longum* subsp. *infantis*, *B. breve*, *B. bifidum* and *B. scardovii*, and we have now included the pairwise AA percent identities in panel b of this figure as well.

a

b

	1	2	3	4	5	6
B. longum subsp. longum 105-A	1	0	0	0	0	0
B. longum subsp. longum DSM20219	99.68	2	0	0	0	0
B. longum subsp. infantis DSM20088	98.42	98.74	3	0	0	0
B. scardovii DSM13734	94.01	94.32	94.32	4	0	0
B. breve DSM20213	94.64	94.95	95.90	93.06	5	0
B. bifidum DSM20456	91.48	91.80	91.17	92.43	89.91	6

Supplementary Figure 7. Analysis of type 4 LDH amino acid sequences.

a, Alignment of the type 4 LDH amino acid sequences of *B. longum* subsp. *longum* 105-A and type strains of *B. longum* subsp. *longum*, *B. longum* subsp. *infantis*, *B. scardovii*, *B. breve* and *B. bifidum*. **b**, Amino acid percent identities (below diagonal) and number of gaps (above diagonal) in pairwise comparisons between the six strains.

We additionally performed BLASTp analyses (against the non-redundant protein sequence database and Swiss-Prot database) using the type 4 LDH amino acid sequence from *B. longum* 105-A as query and excluding *Bifidobacterium* from the results. These analyses resulted in no hits. Lastly, alignment of the type 4 *ldh* amino acid sequence with the amino acid sequence of the well characterized enzyme performing a similar reaction, namely the phenyllactate dehydrogenase (*FldH*) enzyme from *Clostridium sporogenes*, showed a very low amino acid identity of 19%.

To highlight that the type 4 *ldh* is confined to *Bifidobacterium* species, we have now added the following to the manuscript:

Lines 174-178: “The type 4 *ldh* amino acid sequence of the 105-A strain had >98% identity to the homologues in type strains of *B. longum* subsp. *longum* and *B. longum* subsp. *infantis* and >91% identity to *B. bifidum*, *B. breve* and *B. scardovii* (Supplementary Fig. 7), but no non-bifidobacterial homologues were found by BLAST analysis.”

And in the method section:

Line 588-595: “In addition, the ALDH amino acid sequence (translated from the *aldh* nucleotide sequence) of *B. longum* subsp. *longum* 105-A was aligned (gap cost 10, gap extension cost 1) with the ALDH amino acid sequences of the *B. longum* subsp. *longum*, *B. longum* subsp. *infantis*, *B. bifidum*, *B. breve* and *B. scardovii* type strains and pairwise amino acid identity percentages were calculated in CLC Main Workbench. Potential non-bifidobacterial ALDH homologues were searched for by BLASTp analysis of the 105-A amino acid sequence against the non-redundant protein sequence database and the swissprot database.”

9) Page 21, lines 451-452: This sentence seems like a gross overstatement. There is no data in the manuscript showing that aromatic lactic acids play a central role in mediating host-microbiota interactions.

Answer: We have rephrased the sentence to:

Lines 441-442: “Therefore, our findings provide a rationale for further investigation of the implications of aromatic lactic acids in infant health and immune development.”

10) Figure 3 legend: How was the relative fecal abundance of HMOs calculated? It isn’t clear from the legend, text, or methods sections.

Answer: The HMOs were obtained from the untargeted metabolomics faeces data, which were normalized to total intensity per sample. This post-acquisition normalization is a common approach in MS-based metabolomics (Karu et al 2018; *Analytica Chimica Acta*). This is described in the method section (Line 840-841).

11) Page 35, line 778: it is surprising that metabolite abundances were not normalized to urine creatinine. The lack of normalization would likely lead to large inter-individual differences in metabolite levels.

Answer: Normalization to creatinine was previously extensively applied in urinary metabolomics, with the underlying assumption that creatinine concentration reflects the urine sample concentration. However, it has been reported that creatinine excretion is highly dependent on many factors such as age, sex, ethnicity, physical activity, muscle mass, hydration, diet, diurnal rhythms, emotional stress, disease state, body mass, and may therefore not be reliable for sample normalization (Nam et al 2020; *Metabolites*; 10.3390/metabo10090376). Here we used a post-acquisition normalization technique (normalization for each feature across samples to correct the variation in urine concentration), which does not require additional experimental procedures. We have successfully applied this method many times before (Bejder et al 2021; *Medicine & Science in Sports & Exercise*; Zhou et al 2020; *J. Agric. Food Chem.*; Cuparencu et al. 2019; *Molecular Nutrition and Food Research*)

12) Figure 1d: The concentrations for 4-OH-PLA, PLA, and ILA are much lower here than in the Copenhagen cohort, but this is not addressed in the manuscript. Do the authors believe this reflects biological differences among the cohorts or is a technical artifact of sampling or sample processing?

Answer: The differences in concentrations of aromatic lactic acids between the two cohorts are likely to be explained by differences in abundance of infant type *Bifidobacterium* species in the two cohorts (CIG average: 56%, SKOT 1 average: 25%). The differences in these *Bifidobacterium* species are explained by differences in breastfeeding: The stool samples of SKOT 1 cohort infants were collected at 9 months of age where the infants were mixed fed or weaned (lower in *Bifidobacterium*) in contrast to the stool samples of the CIG cohort, which were collected from 0-6 months of infants who were mainly breastfed (higher in *Bifidobacterium*).

Reviewer #3 (Remarks to the Author):

Laursen et al. suggest that breastmilk-promoted *Bifidobacterium* species convert aromatic amino acids into their respective aromatic lactic acids via a previously unrecognised aromatic lactate dehydrogenase. The authors analyzed multiple layers of data (publically available, proprietary longitudinal) to inform mechanistic in vitro and ex vivo experiments. This is a commendable translational approach tackling a biologically important question (while breastfeeding is known to strongly influence the microbiome and infant immune development, the mechanisms are not well understood.) The in vitro and ex vivo work is compelling; however, the low microbiome data quality as well as the statistical analyses often do not support the authors' strong claims. Our main issue is the discrepancy between the low resolution of the 16S amplicon sequencing data (sequencing depth of 2000-8000) along with the unjustified statistical analyses, and the strength of the conclusions the authors are presenting. For example, we are not convinced that one can BLAST 16S data with such low coverage and report subspecies resolution. We therefore request that the authors use deeply sequenced metagenomic data and provide evidence that their conclusions hold. Other specific concerns are outlined below.

Answer: We thank the reviewer for a very thorough review.

First of all, we agree with the reviewer that the statistical analyses dealing with longitudinal data should be updated. We have done so as outlined further below.

Secondly, we acknowledge that 16S rRNA amplicon sequences does not provide the same resolution as metagenomic sequencing. It is however in our view not necessary to obtain strain (or subspecies) level resolution for the primary conclusions presented here, as our results show that the ability to produce the aromatic lactic acids is (*Bifidobacterium*) species dependent. This conclusion holds true when analyzing genomes of 100+ whole genome sequenced human gut derived *Bifidobacterium* strains (Supplementary Table 3) as the type 4 *ldh* gene is found universally in *B. longum*, *B. breve* and *B. bifidum* strains. To clarify, we do not at any point suggest to obtain subspecies level classification based in the 16S rRNA amplicon data. We do that only for qPCR using subspecies specific primers (targeting functional genes only present in the individual subspecies) to distinguish between *B. longum* subsp. *longum* and *B. longum* subsp. *infants*. While even species level resolution is not always possible/reliable with short 16S rRNA fragments, we find that the V3 region is useful for distinguishing between specific *Bifidobacterium* species (e.g. longum, breve and bifidum, See Supplementary data 2d). To substantiate our findings and to confirm the results on species level, we have now performed qPCR (using validated specific primers, see Supplementary Table 5) on the key *Bifidobacterium* species (*B. breve* and *B. bifidum*) in addition to the two *B. longum* subspecies already presented in the manuscript (See new Supplementary Fig. 17). These qPCR results confirm our 16S rRNA amplicon-based species level classification and we argue that qPCR, due to its high sensitivity and specificity is even superior to metagenomics sequencing for our purposes.

Thirdly, regarding the sequencing depth: In general, sequencing depth is unlikely to affect taxonomical classification of OTUs, but it does affect the detection limit meaning that we could miss some low abundant *Bifidobacterium* species in the SKOT data due to the lower depth. This appears however not to be the case. We see no discrepancy in *Bifidobacterium* species identification between the cohorts despite differences in sequencing depth between SKOT and CIG cohort. In other words, we observe the same species classifications within the *Bifidobacterium* OTUs (*B. longum*, *B. breve*, *B. bifidum*, *B. dentium*, *B. scardovii*, *B. adolescentis*, *B. catenulatum* group and *B. pseudolongum/animalis* group) in both cohorts.

Major comments:

1. CIG cohort - 25 infants, sampled for ~10 time points from birth to 6 months of age(16S with qPCR, sequencing depth ~8,000 + targeted metabolomics).
 - a. The authors collapsed the data and ended up with 39 bacterial taxa. We do not think it is likely to find strain

resolution in this dataset.

Answer: Our goal was not to obtain strain level resolution. Our data show that the presence of the type 4 *ldh* and production of the aromatic lactic acids is (*Bifidobacterium*) species dependent. We therefore focus at species level in our 16S data and we now provide additional data showing consistent correlations between OTUs taxonomically classified to *B. longum* (OTU_1), *B. breve* (OTU_2) and *B. bifidum* (OTU_3) and (specific and sensitive) qPCR estimated absolute abundances of *B. breve*, *B. bifidum* in addition to the two *B. longum* subspecies (See below).

b. (lines 243-244) “the relative abundance of Bifido increased with time...” this is a problematic statement due to the compositional nature of the microbiome data. Other taxa can change (while bifido doesn’t change) and still (artificially) affect Bifido’s relative abundance.

Answer: We agree that relative abundance does not necessarily equal absolute abundance. Nonetheless, the increase we observed over time is in fact true in absolute terms also as shown for the infant type *Bifidobacterium* species in new Supplementary Figure 14.

Supplementary Figure 14. Temporal development in abundance of infant type *Bifidobacterium* spp., aromatic lactic acids and human milk oligosaccharides in faeces from infants in the CIG cohort.

Scatter dot plots of age against **a**, absolute abundance of infant type *Bifidobacterium* spp. (defined as the sum of absolute abundances of *B. longum*, *B. breve*, *B. bifidum* and *B. scardovii*), **b-d**, faecal concentrations of aromatic lactic acids (ILA, PLA and 4-OH-PLA, $n=240$) and **e-g**, relative faecal abundance of HMOs (2'FL/3FL, 2'/3'-O-fucosyllactose; 3'SL/6'SL, 3'/6'-O-sialyllactose; LNT/LNnT, lacto-N-tetraose/ lacto-N-neotetraose, $n=228$) during the first 6 months of life in the CIG cohort. A local polynomial regression (LOESS) fit is shown with 95% CI shaded in grey. Statistical significance is evaluated by repeated measures correlations (r_{fm}).

c. (line 246-250) This is longitudinal data, but the analysis does not account for repeated measures. The authors include the same subject multiple times. In parallel, the authors claim that the data structure is explained by subject! The dimensionality reduction must take into account the fact you have repeated measures data.

Answer: We agree with the reviewer. We included the PCoA plots in Fig. 3 merely to illustrate differences in *Bifidobacterium* genus and species abundances between samples. However, we have now deleted the misleading Adonis analyses, since such analysis would require that repeated measures were accounted for as pointed out by the reviewer.

d. (lines 259-262) "The gut microbiota of individuals dominated by infant type *Bifidobacterium* spp. was more stable over time than in individuals with a gut microbiota not dominated by infant type *Bifidobacterium* spp. ($p < 0.0001$, Mann-Whitney U test)" - How did the authors come up with this definition of stability?

Answer: Infants were stratified based on dominance of infant type *Bifidobacterium* species (above or below 50% of the community). Stability was based on the Jaccard similarity index (1- Abundance weighted Jaccard distance). This was calculated by averaging the Jaccard similarity between adjacent time points within each individual of the CIG cohort. For references see (PMID: 27306663, PMID: 23828941 and PMID: 29242832). Given that the stability aspect is not important for our story, we have decided to omit this part, to make the

manuscript more focused.

e. Partial Spearman's Rank correlation analyses with adjustments for age and repeated measures correlation analyses were performed in R using the ppcor and rmcorr packages.

i. partial Spearman's Rank correlation coefficients - partial correlations are used to express the specific portion of variance explained by eliminating the effect of other variables when assessing the correlation between two variables. Why was this analysis used? What is the statistical justification?

ii. The authors performed one analysis that accounts for repeated measures data (i.e., spearman correlation for repeated measures). Nonetheless, this analysis quantifies a correlation within an individual and not across individuals. This is different from the analysis the authors performed on the cross sectional data (SKOT I) and is therefore not comparable.

Answer: We agree that Partial Spearman's Rank correlation analyses are not optimal when dealing with repeated measures. We have therefore omitted these in the revised manuscript and instead use linear mixed models adjusting for subject and age (new Fig. 3c, Supplementary Fig. 17 and Supplementary data 2e) as well as Spearman's rank correlation analyses performed at each sampling point separately (Supplementary Fig. 16), for better comparison with SKOT I data analyses. See methods section lines 997-1003.

f. (lines 270-272): "We have thus established a link between breastfeeding, degradation of HMOs, abundance of specific Bifidobacterium spp. and concentrations of aromatic lactic acids in early infancy. This is a very strong statement that is not backed by proper data and statistical analyses.

Answer: We believe that our updated statistical analyses on the human data, combined with the series of exhaustive and confirmatory mechanistic *in vitro* and *in vivo* (mono-colonised mice) experiments are adequate to support this statement.

g. Figure 3a-c: based on repeated measures data, thus can not make any conclusions across individuals (as they did in the analysis of the SKOT cohort).

Answer: As indicated above, we have now omitted the statistical analyses (Adonis tests) and use the PCoA plots merely to illustrate differences in *Bifidobacterium* abundance between samples.

h. Figure 3e - there is a discrepancy between the qPCR and PCR results (absolute vs. relative abundance) in terms of spearman correlation between the abundance of bifido species and fecal levels of ILA, PLA and 4-OH-PLA (positive) and HMOs (negative). Is the sample size identical?

i. Figure 3e - why is there a discrepancy between the correlation results of the amplicon and qPCR ?

Answer: We are not sure what is meant here – data used are only absolute species abundances (obtained by normalization of the 16S rRNA seq taxa compositional data with qPCR estimated total bacterial abundance, as described in methods section under "Quantitative PCR") as mentioned in the figure legend. We cannot see any discrepancy between 16S data and qPCR data? The analyses are now updated with linear mixed models (see below). 240 samples were used for the associations between Bifidobacterium and aromatic lactic acids, but 12 samples were omitted in the associations between bifidobacteria and HMOs as stool samples were collected when the infant was no longer breastfed (n=228). The 16S rRNA data is presented in the new Fig. 3c and the qPCR data validating this is presented in Supplementary Fig. 17. The data are in our view highly consistent.

Fig.3c, Heatmap illustrating linear mixed model coefficients (adjusted for subject and age) between the absolute abundance of *Bifidobacterium* species and absolute faecal concentrations of aromatic lactic acids (n=240) or relative abundances of HMOs (n=228) in the CIG cohort. Infant type *Bifidobacterium* spp. is the sum of absolute abundances of *B. longum*, *B. breve*, *B. bifidum* and *B. scardovii*. Statistical significance was evaluated by FDR-corrected p-values indicated by asterisks with * $p < 0.05$, ** $p < 0.01$, *** $p < 0.001$ and **** $p < 0.0001$.

Supplementary Figure 17. Associations between qPCR estimated absolute abundance of *Bifidobacterium* species and faecal abundances of the aromatic lactic acids and human milk oligosaccharides in the CIG cohort.

Heatmap illustrating linear mixed model coefficients (adjusted for subject and age) between the absolute abundance of *Bifidobacterium* species estimated by qPCR and absolute faecal concentrations of aromatic lactic

acids (n=240) or relative abundances of HMOs (n=228) in the CIG cohort. Statistical significance was evaluated by p-values indicated by asterisks with * p<0.05, **p<0.01, ***p<0.001 and ****p<0.0001.

j. (lines 273 - 277+ Figure 3 f-h) - this is descriptive and lacking statistical validation. This does not indicate anything about the “dynamics”. No statistical analyses to support claims like: “breastfed infants early and consistently colonised by infant type Bifidobacterium species consistently showed high concentrations of aromatic lactic acids in faeces....” . It seems more anecdotal than anything else.

Answer: We agree and acknowledge that we cannot make firm conclusions based on individual observations. However, we still think these observations are worth mentioning to stimulate further research down these lines. This is now clearly phrased in the revised manuscript.

Lines 304-308: *“Together, these results demonstrate that HMO-utilizing infant type Bifidobacterium species determine the abundance of aromatic lactic acids in the infant gut. Yet, the impact of early/late Bifidobacterium colonization, pre-term delivery, exposure to antibiotics and formula supplementation with respect to bifidobacterial aromatic lactic acid production warrants further investigation.”*

k. (lines 277 - 291) - While it is interesting to see individual trajectories with granular details, these are ultimately descriptive data from a relatively small number of infants. Any inferences drawn from this data should be tempered. We question whether (even after reanalysis as recommended above) these data should be included in the paper at all, as they do not provide strong evidence to support the overall story. (lines 289-291) - “This indicates that bifidobacterial aromatic lactic acid production is compromised by pre-term delivery, exposure to antibiotics and formula supplementation”. The authors had just 2 preterm deliveries and 3 infants receiving antibiotics! This is another unbacked claim.

Answer: We agree that these are descriptive data. We have tempered and rephrased the paragraph as outlined in the answer above.

2. Conceptual questions

a. How relevant is the ex vivo assay with immune cells from blood, given that Bifidobacterium-produced ILA is located in the gut? Do the authors believe that ILA is absorbed into circulation? They show that fecal and urinary levels are correlated - is all urinary ILA necessarily derived from gut microbiota? If so, please clarify and provide citations.

Answer: Indoles including ILA are microbially-derived metabolites that the mammalian host normally cannot generate (Roager and Licht, 2018; Nature Communications Wikoff et al 2009 PNAS; Dodd et al., 2017 Nature). Animal studies have shown that bacterially produced ILA is absorbed into the circulation (Dodd et al., 2017 Nature). This is also evident from the present study where we measure aromatic lactic acids in the urine of SKOT 1 infants and show that faecal ILA and urine ILA are positively correlated (Spearman rho=0.68) (Supplementary Figure 1b,c). Also, we know from adult cohort studies that ILA is measured in blood where it correlates with PLA and 4-OH-PLA (rho=0.35) (Shin et al 2014; Nature Genetics).

We have added the following sentence to the manuscript to justify the ex vivo assay with immune cells from blood.

Lines 364-365: *“Since ILA upon absorption in the gut is circulated in the body (Dodd et al., 2017; Nature), we next asked whether ILA affects immune function via AhR and HCAR3.”*

b. How relevant are the stool concentrations of ILA? Do they reflect concentrations in the gut?

Answer: Stool concentrations are the net result of metabolite production and absorption. Therefore, concentrations of metabolites are likely to be higher locally in specific sites in the gut.

c. 19 aromatic amino acids and derivatives were measured. Does this capture all or the majority of such compounds expected in the infant gut?

Answer: Our targeted analysis of aromatic amino acids derivatives covered all the most relevant known downstream products of aaa catabolism. But they probably represent only a subset of all the many metabolites that are potentially present in a stool sample.

d. Short chain fatty acids are bacterial metabolites that have been studied in relation to infant immune function and development. This evidence base is relevant but not discussed.

Answer: We agree that the short-chain fatty acids (SCFAs) are probably the most studied group of microbial metabolites in relation to infant immune development. However, this group of metabolites were not of focus for the present study, which is why we did not discuss them here. Yet, to acknowledge that this group of microbial metabolites is relevant in the context of early life, we have now added a sentence in the introduction:

Lines 59-63: “Despite *Bifidobacterium* dominating the gut of breastfed infants and being widely acknowledged as beneficial, mechanistic insights on the contribution of these bacteria and their metabolites to immune function and development during infancy are limited and have mainly focused on short-chain fatty acids (ref 11,12)”

3. General analysis issues:

a. Why did the authors use Qiime and not Qiime 2? Why OTUs and not ASVs? How sensitive are the results to the denoising and clustering method? Taxonomic annotation is more reliable with DADA2 for example as compared with OTUs.

Answer: Microbiome analyses were performed before QIIME2 was launched. We have now also performed ASV analyses, but this produced highly similar results (in terms of relative abundances and taxonomical classifications, see table below comparing the top 10 most abundant taxa). Therefore, a DADA2 analysis would not change our conclusions.

#ASV ID	RA	BLAST	#OTU ID	RA	BLAST
ASV_1	30.5794637	B. longum 100%	OTU_1	38.46906	B. longum 100%
ASV_2	8.60306278	B. breve 100%	OTU_2	9.118381	B. breve 100%
ASV_4	7.91936806	B. bifidum 100%	OTU_3	7.947536	B. bifidum 100%
ASV_3	7.54332899	B. longum 100%	OTU_4	6.37007	B. catenulatum group 100%
ASV_5	6.33839503	B. catenulatum group 100%	OTU_5	5.476749	C. neonatale 100%
ASV_6	5.28743819	C. neonatale 100%	OTU_9	3.69594	E. coli 100%
ASV_7	4.18377482	E. coli 100%	OTU_8	2.034769	R. gnavus 100%
ASV_10	1.99790864	R. gnavus 100%	OTU_10	1.904726	S. salivarius 100%
ASV_11	1.83952587	S. salivarius 100%	OTU_7	1.645503	B. dentium 99%
ASV_9	1.63074607	B. dentium 99%	OTU_25	1.503672	Enterococcus spp. 99%
ASV_8	1.43358531	E. ramosum 100%	OTU_6	1.434386	E. ramosum 100%

b. The sequencing depth is very very low (2000 SKOT - 8000 CIG).

Answer: We acknowledge that the minimal sequencing depths are low. However, the rarefaction to these minimal sequencing depths were only used for beta diversity and alpha diversity measures (fig. 1a-b and fig. 3a-b), and it is our experience that the measures we use (e.g. Bray Curtis and Shannon index) are robust to sequencing depths as low as a few thousand reads. To clarify, for calculation of relative abundances of each taxon total sum scaling of the un-rarefied OTU tables were used. Therefore, for the taxon abundance data we do in fact have 21781± 3110 (mean±SD) quality filtered reads for SKOT I and 40156±17614 (mean±SD) quality filtered reads for CIG. This is now more clearly stated in the methods section:

Lines 494-497: “In QIIME (ref 80), OTU tables (nOTUs[SKOT] = 545, nOTUs[CIG] = 478) was filtered to include only OTUs with abundance across all samples above 0.005% of the total OTU counts (nOTUs[SKOT]= 258, nOTUs[CIG]= 145), ending up with 21781±13110 (mean±SD) reads for SKOT I and 40156±17614 (mean±SD)

reads for CIG. OTU relative abundances within samples were then estimated by total sum scaling.”

While the differences in seq depth between samples (and cohorts) affects detection limit and thus the risk of missing some low abundant *Bifidobacterium* species, we do observe the same species classifications within the *Bifidobacterium* OTUs (*B. longum*, *B. breve*, *B. bifidum*, *B. dentium*, *B. scardovii*, *B. adolescentis*, *B. catenulatum* group and *B. pseudolongum/animalis* group) in both cohorts. In addition, our specific qPCR validates that the key taxonomical annotations are OK. (See answer above)

c. To our knowledge, using BLAST on 16S data to find subspecies resolution is not reliable. You need metagenomic data to do that.

Answer: We go to subspecies level only with qPCR using specific and validated primers (Supplementary Table 5). Please note that qPCR is more sensitive than metagenomics for this type of analysis.

d. Absolute abundances of *B. longum* subsp. *Longum* and *B. longum* subsp. *Infantis* (subspecies specific primers) were estimated by qPCR. Why didn't the authors use subspecies specific primers for other subspecies?

Answer: We have now included qPCR data using species specific primers targeting *B. bifidum* and *B. breve* in the CIG cohort, in addition to the two *longum* subspecies (see answer above). This validates the associations between *Bifidobacterium* OTUs (based on 16S rRNA amplicons) and aromatic lactic acids and HMOs (Supplementary Fig. 17).

e. A lot of the statements are lacking statistical justifications and analyses (e.g., lines 243-244 “on average the microbial composition and shannon diversity did not change dramatically...”; lines 262-264 - “Using solely samples from breastfed infants, faecal abundances of HMO residuals showed a progressive decline with age, concurrent with the progressive increase in infant type *Bifidobacterium* species”).

Answer: We thank the reviewer for pointing out this. Since this was not our focus, we have left out the part about diversity. Our updated analyses (mixed linear models adjusted for age and subject) validates the association between faecal HMOs and infant type *Bifidobacterium* species (Fig. 3c and Supplementary Fig. 17).

f. The authors would revise the methods section and add significantly more details on the statistical analysis. Just writing the packages used is not good enough.

Answer: We are not sure what level of detail the referee ask for here, but we have added extra details on the exact functions used within each R-package including the version numbers (Lines 991-1015). We are of course also willing to share R-scripts upon request, but we feel that a comprehensive description of all graphical and statistical details in this section is unnecessary as details are readily available elsewhere and since our analyses uses “default” parameters for the functions used - unless otherwise stated.

g. (line 825) Metabolomics data - The data were normalised to the total intensity. That is wrong. The data should be normalized across samples and not within a sample.

Answer: Normalization to total intensity is a common post-acquisition normalization method in metabolomics research to account for water content differences (Karu et al 2018; Analytica Chimica Acta). When comparing normalization to total intensity (SUM-normalized) with normalization for each feature across samples (Unit-normalized), we obtained very similar results. Therefore, we have kept the original metabolomics data obtained by normalizing to total intensity.

Minor comments:

The authors claim to find an association between Bifidobacterium subspecies and aromatic amino acid catabolites during late infancy using 16S with sequencing depth of 2,000 + targeted metabolomics data.

SKOT I cohort -

59 healthy Danish infants from the SKOT I cohort (published in 2010). The infants were born full term, ~9 of age at sampling with ~40% partially breastfed and ~60% weaned.

4. The authors claim (line 85) that “other metadata (age, gender, mode of delivery, current formula intake and age of introduction to solid foods) did not explain gut microbiota variation to the same degree as breastfeeding status (Supplementary Data 1c,d) - how was this tested?

Answer: We have now specified in the text that this was tested using Adonis as well as Mann-Whitney U tests. Lines 88-92: “Other metadata (age, gender, mode of delivery, current formula intake and age of introduction to solid foods) did not explain gut microbiota variation to the same degree as breastfeeding status ($r^2 < 0.05$, $p > 0.03$, Adonis tests; Supplementary Data 1c) and no bacterial genera differed significantly according to these parameters (FDR p-value > 0.1 , Mann-Whitney U tests; Supplementary Data 1d).”

5. (line 87) “Principal Component Analysis (PCA) of faecal aromatic amino acid metabolite concentrations also revealed...” - what data transformation was used for the microbiome data? For the metabolites data? The microbiome data is compositional and the metabolite data need to be normalized per component. In any case, this separation doesn’t seem significant (also - no p-value reported).

Answer: In the QIIME pipeline, based on the rarefied OTU counts a weighted Unifrac distance matrix was computed before ordination (PCoA). The absolute faecal concentrations were used for the PCA. The data were normalized per component since it were auto-scaled before PCA. We did not perform statistical test on the PCA clustering, as this is not central to the point, since mainly the three aromatic lactic acid stands out. Accordingly we have modified the wording:

Lines 92-95: “Principal Component Analysis (PCA) of faecal aromatic amino acid metabolite concentrations also **suggested a minor** separation by breastfeeding status, which was largely driven by three aromatic lactic acids, 4-hydroxyphenyllactic acid (4-OH-PLA), phenyllactic acid (PLA) and indolelactic acid (ILA) (Fig. 1c)”

6. (line 90) “Correlation analysis revealed that Bifidobacterium, but no other bacterial genera, were

significantly associated with faecal concentrations of all three aromatic lactic acids (4-OH-PLA, PLA and ILA), in addition to indolealdehyde (IAld). How was this correlation performed? Did the authors transform the data (z-score per metabolite and some log ratio transformation for the microbiome)?

Answer: No transformation was done, since we used a non-parametric Spearman Rank correlation this does not affect the results.

7. (line 93) 16S data with 2,000 reads does not seem sufficient resolution for bifido spp. identification (e.g., *B. longum*, *bifidum*, *breve*).

Answer: We do not agree, as the sequencing depth is unlikely to affect taxonomical classification of OTUs, but only the detection limit as discussed above.

8. (line 109) “Together, this suggests that specific breastmilk-promoted *Bifidobacterium* species in the gut of infants convert aromatic amino acids to the corresponding aromatic lactic acids” - This is a mechanistic statement. Given the analyses performed and associations found, the authors can not make this claim at this stage of the paper, before the mechanistic studies are presented.

Answer: We agree that at this stage in the paper we should only suggest that specific *Bifidobacterium* species produce aromatic lactic acids. We have adjusted the manuscript accordingly:

Lines 114-115: “Together, this suggests that specific *Bifidobacterium* species produce aromatic lactic acids in the infant gut”

Decision Letter, first revision:

Dear Henrik,

Thank you for submitting your revised manuscript "Breastmilk-promoted bifidobacteria produce aromatic lactic acids in the infant gut" (NMICROBIOL-21020336A). It has now been seen by the original referees and their comments are below. The reviewers find that the paper has improved in revision, and therefore we'll be happy in principle to publish it in Nature Microbiology, pending minor revisions to satisfy the referees' final requests and to comply with our editorial and formatting guidelines.

Thank you again for your interest in Nature Microbiology Please do not hesitate to contact me if you have any questions.
{REDACTED}

Reviewer #1 (Remarks to the Author):

The authors have addressed all comments.

Reviewer #2 (Remarks to the Author):

This is a review of a resubmitted manuscript entitled "Breastmilk-promoted bifidobacteria produce aromatic lactic acids in the infant gut" authored by Laursen et al. In response to queries by the referees, the authors present a revised manuscript which addresses many of the referee comments. Overall, the authors have adequately addressed my previous comments, please find point by point responses below:

Major comments:

- 1) The data provided in supplementary figure 8 provide direct evidence that *B. longum* produces aromatic lactic acids in the gut and this involves the type 4 Idh gene. This adds stronger weight to the conclusion that bifidobacteria produce aromatic lactic acids in the gut, a claim which previously was based on statistical associations.
- 2) The authors provide a figure that demonstrates differences in recovery of internal standards used. This information should be included in the supplementary material and should be clearly mentioned in the text. As the authors point out (and I agree), conclusions based on the trends of individual metabolites are unlikely to be changed. However, the claim that ILA is the most abundant aromatic lactic acid (Page 16, line 334) is not supported. I would suggest amending the text to avoid claims comparing absolute values of different metabolites.

Minor comments:

- 1) OK.
- 2) OK.
- 3) OK.
- 4) OK.
- 5) OK.
- 6) OK.
- 7) OK.
- 8) I would suggest including the % amino acid identity cutoff used to conclude “no non-bifidobacterial homologues were found” in the text.
- 9) OK.
- 10) OK.
- 11) OK.
- 12) OK.

Reviewer #3 (Remarks to the Author):

The revised manuscript is significantly improved, and the authors have addressed our main concerns. We are satisfied with the revised longitudinal data analysis and the addition of the qPCR component. The new mouse experiment and in silico analysis of the enzyme across species (in response to the other reviewers) are excellent additions. Nonetheless, we still have reservations regarding the standardization of metabolomics data and the microbiome analysis, which is not aligned with the state-of-the-art in microbiome science. We leave it to the Editor’s discretion whether to insist on rerunning the computational pipeline using state-of-the-art methods - as these are now standards in the field.

Specific comments:

1. As stated in our original review, the authors are using old microbiome analysis frameworks and methods which is problematic (QIIME rather than QIIME 2 that was published in 2019, OTUs rather than ASVs).

2. The normalization of the metabolomic data seems inappropriate. Unlike microbial samples, metabolomics data are not compositional and should not be normalized to the sum of the sample. Metabolite X should be compared across samples, not to other metabolites in the same sample. When normalizing to the sum, you enforce a negative correlation between metabolites that does not exist. We would advise normalizing across metabolites using a z-score (and not within a sample). The authors indicated they got similar results with both normalization methods. Perhaps an expert statistical review is warranted.

3. The authors did not revise the statistical analysis section in the methods. To clarify, our request was to describe the data transformations and statistical tests performed as well as the assumptions made (e.g., use of non-parametric test as the data is not Gaussian). Currently, this section is composed of a list of R packages and commands which are the implementation of the statistical analyses and not their description. The manuscript, and research following it, would benefit greatly from an elaborative “statistical analysis section” (in the supplement if necessary). This would allow reproducibility as the authors don’t always follow the state-of-the-art microbiome analysis (for

example, transformations are typically used to deal with compositionality and other data characteristics, regardless of the assumed underlying distribution - but this has not been done.)

Decision Letter, final checks:

Dear Henrik,

Thank you for your patience as we've prepared the guidelines for final submission of your Nature Microbiology manuscript, "Breastmilk-promoted bifidobacteria produce aromatic lactic acids in the infant gut" (NMICROBIOL-21020336A). Please carefully follow the step-by-step instructions provided in the attached file, and add a response in each row of the table to indicate the changes that you have made. Please also check and comment on any additional marked-up edits we have proposed within the text. Ensuring that each point is addressed will help to ensure that your revised manuscript can be swiftly handed over to our production team.

In recognition of the time and expertise our reviewers provide to Nature Microbiology's editorial process, we would like to formally acknowledge their contribution to the external peer review of your manuscript entitled "Breastmilk-promoted bifidobacteria produce aromatic lactic acids in the infant gut". For those reviewers who give their assent, we will be publishing their names alongside the published article.

Nature Microbiology offers a Transparent Peer Review option for new original research manuscripts submitted after December 1st, 2019. As part of this initiative, we encourage our authors to support increased transparency into the peer review process by agreeing to have the reviewer comments, author rebuttal letters, and editorial decision letters published as a Supplementary item. When you submit your final files please clearly state in your cover letter whether or not you would like to participate in this initiative. Please note that failure to state your preference will result in delays in accepting your manuscript for publication.

Cover suggestions

As you prepare your final files we encourage you to consider whether you have any images or illustrations that may be appropriate for use on the cover of Nature Microbiology.

Nature Microbiology has now transitioned to a unified Rights Collection system which will allow our Author Services team to quickly and easily collect the rights and permissions required to publish your work. Approximately 10 days after your paper is formally accepted, you will receive an email in providing you with a link to complete the grant of rights. If your paper is eligible for Open Access, our Author Services team will also be in touch regarding any additional information that may be required to arrange payment for your article.

Please note that *Nature Microbiology* is a Transformative Journal (TJ). Authors may publish their research with us through the traditional subscription access route or make their paper immediately open access through payment of an article-processing charge (APC). Authors will not be required to make a final decision about access to their article until it has been accepted. [Find out more about Transformative Journals](https://www.springernature.com/gp/open-research/transformative-journals)

Authors may need to take specific actions to achieve compliance with funder and institutional open access mandates. For submissions from January 2021, if your research is supported by a funder that requires immediate open access (e.g. according to [Plan S principles](https://www.springernature.com/gp/open-research/plan-s-compliance)) then you should select the gold OA route, and we will direct you to the compliant route where possible. For authors selecting the subscription publication route our standard licensing terms will need to be accepted, including our [self-archiving policies](https://www.springernature.com/gp/open-research/policies/journal-policies). Those standard licensing terms will supersede any other terms that the author or any third party may assert apply to any version of the manuscript.

For information regarding our different publishing models please see our page

[href="https://www.springernature.com/gp/open-research/transformative-journals">](https://www.springernature.com/gp/open-research/transformative-journals) Transformative Journals page. If you have any questions about costs, Open Access requirements, or our legal forms, please contact ASJournals@springernature.com.

Please use the following link for uploading these materials:
{REDACTED}

{REDACTED}

Reviewer #1:

Remarks to the Author:

The authors have addressed all comments.

Reviewer #2:

Remarks to the Author:

This is a review of a resubmitted manuscript entitled "Breastmilk-promoted bifidobacteria produce aromatic lactic acids in the infant gut" authored by Laursen et al. In response to queries by the referees, the authors present a revised manuscript which addresses many of the referee comments. Overall, the authors have adequately addressed my previous comments, please find point by point responses below:

Major comments:

1) The data provided in supplementary figure 8 provide direct evidence that *B. longum* produces aromatic lactic acids in the gut and this involves the type 4 Idh gene. This adds stronger weight to the conclusion that bifidobacteria produce aromatic lactic acids in the gut, a claim which previously was based on statistical associations.

2) The authors provide a figure that demonstrates differences in recovery of internal standards used. This information should be included in the supplementary material and should be clearly mentioned in the text. As the authors point out (and I agree), conclusions based on the trends of individual metabolites are unlikely to be changed. However, the claim that ILA is the most abundant aromatic lactic acid (Page 16, line 334) is not supported. I would suggest amending the text to avoid claims comparing absolute values of different metabolites.

Minor comments:

- 1) OK.
- 2) OK.
- 3) OK.
- 4) OK.
- 5) OK.
- 6) OK.
- 7) OK.

8) I would suggest including the % amino acid identity cutoff used to conclude “no non-bifidobacterial homologues were found” in the text.

9) OK.

10) OK.

11) OK.

12) OK.

Reviewer #3:

Remarks to the Author:

The revised manuscript is significantly improved, and the authors have addressed our main concerns. We are satisfied with the revised longitudinal data analysis and the addition of the qPCR component. The new mouse experiment and in silico analysis of the enzyme across species (in response to the other reviewers) are excellent additions. Nonetheless, we still have reservations regarding the standardization of metabolomics data and the microbiome analysis, which is not aligned with the state-of-the-art in microbiome science. We leave it to the Editor’s discretion whether to insist on rerunning the computational pipeline using state-of-the-art methods - as these are now standards in the field.

Specific comments:

1. As stated in our original review, the authors are using old microbiome analysis frameworks and methods which is problematic (QIIME rather than QIIME 2 that was published in 2019, OTUs rather than ASVs).

2. The normalization of the metabolomic data seems inappropriate. Unlike microbial samples, metabolomics data are not compositional and should not be normalized to the sum of the sample. Metabolite X should be compared across samples, not to other metabolites in the same sample. When normalizing to the sum, you enforce a negative correlation between metabolites that does not exist. We would advise normalizing across metabolites using a z-score (and not within a sample). The authors indicated they got similar results with both normalization methods. Perhaps an expert statistical review is warranted.

3. The authors did not revise the statistical analysis section in the methods. To clarify, our request was to describe the data transformations and statistical tests performed as well as the assumptions made (e.g., use of non-parametric test as the data is not Gaussian). Currently, this section is composed of a list of R packages and commands which are the implementation of the statistical analyses and not their description. The manuscript, and research following it, would benefit greatly from an elaborative “statistical analysis section” (in the supplement if necessary). This would allow reproducibility as the authors don’t always follow the state-of-the-art microbiome analysis (for example, transformations are typically used to deal with compositionality and other data characteristics, regardless of the assumed underlying distribution - but this has not been done.)

Response letter to reviewers

We sincerely thank the reviewers for their constructive critiques and suggestions, which have helped us to improve the clarity and quality of our work. Please find our point-by-point responses to each of the remaining comments below.

Reviewer #1 (Remarks to the Author):

The authors have addressed all comments.

Reviewer #2 (Remarks to the Author):

This is a review of a resubmitted manuscript entitled “Breastmilk-promoted bifidobacteria produce aromatic lactic acids in the infant gut” authored by Laursen et al. In response to queries by the referees, the authors present a revised manuscript which addresses many of the referee comments. Overall, the authors have adequately addressed my previous comments, please find point by point responses below:

Answer: Thank you!

Major comments:

1) The data provided in supplementary figure 8 provide direct evidence that *B. longum* produces aromatic lactic acids in the gut and this involves the type 4 *ldh* gene. This adds stronger weight to the conclusion that bifidobacteria produce aromatic lactic acids in the gut, a claim which previously was based on statistical associations.

Answer: We agree.

2) The authors provide a figure that demonstrates differences in recovery of internal standards used. This information should be included in the supplementary material and should be clearly mentioned in the text. As the authors point out (and I agree), conclusions based on the trends of individual metabolites are unlikely to be changed. However, the claim that ILA is the most abundant aromatic lactic acid (Page 16, line 334) is not supported. I would suggest amending the text to avoid claims comparing absolute values of different metabolites.

Answer: We agree and have removed the sentences stating that ILA is the most abundant aromatic lactic acids. In addition, we have as suggested by the reviewer included a supplementary figure that demonstrates differences in recovery of the internal standards used and stated this in the text:

Line 744-748: *“The recoveries of the internal standards varied but were relative to each other in general rather consistent (Supplementary Fig. 14) emphasizing that while the absolute concentrations may not be accurate due to lack of isotope-labeled internal standards for each single analyte, the relative metabolite concentrations across samples were robust with the applied LC-MS method.”*

8) I would suggest including the % amino acid identity cutoff used to conclude “no non-bifidobacterial homologues were found” in the text.

Answer: Thank you for this suggestions. We agree and have included the amino acid sequence identity cutoff of 60%.

Lines 154-158: *“The type 4 LDH amino acid sequence of the 105-A strain had >98% identity to the homologues in type strains of *B. longum* subsp. *longum* and *B. longum* subsp. *infantis* and >91% identity to *B. bifidum*, *B. breve* and *B. scardovii* (Supplementary Fig. 6b), but no non-bifidobacterial homologues were found by BLAST analysis (amino acid sequence identity cutoff 60%).”*

Reviewer #3 (Remarks to the Author):

The revised manuscript is significantly improved, and the authors have addressed our main concerns. We are satisfied with the revised longitudinal data analysis and the addition of the qPCR component. The new mouse experiment and in silico analysis of the enzyme across species (in response to the other reviewers) are excellent additions. Nonetheless, we still have reservations regarding the standardization of metabolomics data and the microbiome analysis, which is not aligned with the state-of-the-art in microbiome science. We leave it to the Editor's discretion whether to insist on rerunning the computational pipeline using state-of-the-art methods - as these are now standards in the field.

Answer: Thank you!

Specific comments:

1. As stated in our original review, the authors are using old microbiome analysis frameworks and methods which is problematic (QIIME rather than QIIME 2 that was published in 2019, OTUs rather than ASVs).

Answer: We have now verified our main results based on OTU analyses by additionally performing ASV analyses now shown in **Extended Data Fig. 8a**. The results are highly similar regarding *Bifidobacterium* ASVs (*B. longum*, *B. breve* and *B. bifidum*) correlating positively with the aromatic lactic acids and negatively with the HMO residuals in feces. This has been included in the manuscript.

Main text:

Line 225-226: *"Furthermore, re-analysing the microbiome data at the ASV level showed very similar results (Extended Data Fig. 8a)."*

Methods:

Line 419-425: *"In addition, to validate the findings from the OTU analysis, we performed amplicon sequence variant (ASV) analysis on the CIG cohort samples using the DADA2 pipeline v1.14 (ref 83) with the demultiplexed and trimmed reads and the same cutoffs as for the OTU analysis (MAX_EE=1, ASVs filtered to include only those with abundance across all samples above 0.005% of the total ASV counts), resulting in a total of 211 ASVs and 12 ASVs assigned to Bifidobacterium (using the RDP database v18) above the abundance cutoff of 0.1% of the total Bifidobacterium population (Supplementary Data 2f)."*

2. The normalization of the metabolomic data seems inappropriate. Unlike microbial samples, metabolomics data are not compositional and should not be normalized to the sum of the sample. Metabolite X should be compared across samples, not to other metabolites in the same sample. When normalizing to the sum, you enforce a negative correlation between metabolites that does not exist. We would advise normalizing across metabolites using a z-score (and not within a sample). The authors indicated they got similar results with both normalization methods. Perhaps an expert statistical review is warranted.

Answer: We refer to our previous answer on this matter.

3. The authors did not revise the statistical analysis section in the methods. To clarify, our request was to describe the data transformations and statistical tests performed as well as the assumptions made (e.g., use of non-parametric test as the data is not Gaussian). Currently, this section is composed of a list of R packages and commands which are the implementation of the statistical analyses and not their description. The manuscript, and research following it, would benefit greatly from an elaborative “statistical analysis section” (in the supplement if necessary). This would allow reproducibility as the authors don’t always follow the state-of-the-art microbiome analysis (for example, transformations are typically used to deal with compositionality and other data characteristics, regardless of the assumed underlying distribution - but this has not been done.)

Answer: We have described in the methods section all statistical tests used and included a statement on

handling of normally/non-normally distributed data. We have further described how distance/dissimilarity matrices were generated (no transformations) and that ADONIS and PERMDISP tests were run directly on these. We use mainly premade and available scripts within QIIME or R (rmcorr, Maaslin2) and all deviations from defaults settings are mentioned in the statistics section. No custom code was used for the statistical analyses. We therefore believe that the information provided in the statistics section is adequate to reproduce results, but we are of course willing to share R-scripts upon request and have provided this statement under “code availability”. Regarding compositionality of the microbiome data, we deal with this by multiplying relative abundance of taxa with total bacterial abundance (bacterial load) based on qPCR using universal primers.

Final Decision Letter

Dear Henrik,

I am pleased to accept your Article "Bifidobacterium species associated with breastfeeding produce aromatic lactic acids in the infant gut" for publication in Nature Microbiology. Thank you for having chosen to submit your work to us and many congratulations.

Before your manuscript is typeset, we will edit the text to ensure it is intelligible to our wide readership and conforms to house style. We look particularly carefully at the titles of all papers to ensure that they are relatively brief and understandable.

Acceptance of your manuscript is conditional on all authors' agreement with our publication policies (see www.nature.com/nmicrobiolate/authors/gta/content-type/index.html). In particular your manuscript must not be published elsewhere and there must be no announcement of the work to any media outlet until the publication date (the day on which it is uploaded onto our website).

Please note that *Nature Microbiology* is a Transformative Journal (TJ). Authors may publish their research with us through the traditional subscription access route or make their paper immediately open access through payment of an article-processing charge (APC). Authors will not be required to make a final decision about access to their article until it has been accepted. [Find out more about Transformative Journals](https://www.springernature.com/gp/open-research/transformative-journals)

Authors may need to take specific actions to achieve [compliance](https://www.springernature.com/gp/open-research/funding/policy-compliance-faqs) with funder and institutional open access mandates. For submissions from January 2021, if your research is supported by a funder that requires immediate open access (e.g. according to [Plan S principles](https://www.springernature.com/gp/open-research/plan-s-compliance)) then you should select the gold OA route, and we will direct you to the compliant route where possible. For authors selecting the subscription publication route our standard licensing terms will need to be accepted, including our [self-archiving policies](https://www.springernature.com/gp/open-research/policies/journal-policies). Those standard licensing terms will supersede any other terms that the author or any third party may assert apply to any version of the manuscript.

We welcome the submission of potential cover material (including a short caption of around 40 words) related to your manuscript; suggestions should be sent to Nature Microbiology as electronic

files (the image should be 300 dpi at 210 x 297 mm in either TIFF or JPEG format). Please note that such pictures should be selected more for their aesthetic appeal than for their scientific content, and that colour images work better than black and white or grayscale images. Please do not try to design a cover with the Nature Microbiology logo etc., and please do not submit composites of images related to your work. I am sure you will understand that we cannot make any promise as to whether any of your suggestions might be selected for the cover of the journal.
